


# Temporal variation of [129]I and [127]I in aerosols from Xi'an, China: influence of East Asian monsoon and heavy haze events

Luyuan Zhang [1,2*], Xiaolin Hou [1,2,3], Sheng Xu [4], Tian Feng [1], Peng Cheng [1], Yunchong Fu [1], Ning Chen [1]

[1]State Key Laboratory of Loess and Quaternary Geology, Shaanxi Key Laboratory of Accelerator Mass Spectrometry Technology and Application, Xi'an AMS Center, Institute of Earth Environment CAS, Xi'an 710061, China

[2]Center for Excellence in Quaternary Science and Global Change, Chinese Academy of Sciences, Xian 710061, China

[3]Center for Nuclear Technologies, Technical University of Denmark, Risø Campus, Roskilde 4000, Denmark

[4]Institute of Surface-Earth System Science, Tianjin University, Tianjin 300072, China

*Correspondence to*: Luyuan Zhang (zhangluyuan.118@163.com)

**Abstract.** Aerosol iodine isotopes are pivotal links in atmospheric circulation of iodine in both atmospheric and nuclear sciences, while their sources, temporal change and transport are still not well understood. This work presents the day-resolution temporal variation of iodine-129 ([129]I) and iodine-127 ([127]I) in aerosols from Xi'an, northwest China during 2017/2018. Both iodine isotopes have significant fluctuations with time, showing highest levels in winter, approximately two to three times

higher than in other seasons, but the correlation between [129]I and [127]I reflects they have different sources. Aerosol [127]I is found to be noticeably positively correlated with air quality index and five air pollutants. Enhanced fossil fuel combustion and inverse weather conditions can explain the increased concentrations and peaks of [127]I in winter. The change of [129]I confirms that source and level of [129]I in the monsoonal region were alternatively dominated by the [129]I-enriched East Asian winter monsoon and [129]I-poor East Asian summer monsoon. The mean [129]I/[127]I of $(101 \pm 124) \times 10^{-10}$ provides an atmospheric background level for

the purpose of nuclear environmental safety monitoring. This study suggests that locally discharged stable [127]I and externally input [129]I are likely involved into fine particles formation in urban air, shedding insights into long-range transport of air pollutants and iodine's role in particulate formation in urban atmosphere.

## 1 Introduction

Iodine is one of active halogen elements, and involved into plenty of atmospheric chemical reactions (i.e. ozone depletion and

new particles formation from condensable iodine-containing vapours), drawing increasing attention in not only atmospheric science, but also environmental fields in recent years (Saiz-Lopez et al., 2012). A number of studies on atmospheric iodine just focus on the processes and mechanisms in marine boundary layer since over 99.8% of iodine derives from ocean (McFiggans et al., 2000). Other sources of iodine in air comprise volatile iodine and resuspended particles from soil, as well as combustion of fossil fuel (Fuge and Johnson, 1986). Whitehead et al. estimated annual release of iodine from fossil fuel

combustion is about 400 ton, accounting for only 0.1% of total iodine in air (Whitehead, 1984). Whereas, anthropogenic iodine





in Chinese megacities is believed to be significantly underestimated due to coal combustion (Wu et al., 2014). Few studies have found high iodine concentrations in air and particles in China (Gao et al., 2010; Xu et al., 2010). Although marine atmospheric iodine has been proven to form fine particles, little is known about terrestrial atmospheric iodine, particularly in urban sites with severe air pollution.

Along with atmospheric circulation of stable $^{127}$I, long-lived radioactive $^{129}$I with half-life of 15.7 million years is also of importance in global transport since it is a major fission product with yield of 0.7% in nuclear industry. China is in transition phase of energy structure to solve the environmental pollution issues, and has put great emphasis on developing nuclear power (World Nuclear Association, 2017). Furthermore, nuclear waste reprocessing is in the process of construction in China, which may be a key source of $^{129}$I in the future. Investigation on level, sources, temporal changes are extremely necessary for nuclear

environmental safety assessment and nuclear emergency preparedness. Environmental $^{129}$I/$^{127}$I atomic ratios have been increased from natural $^{129}$I level of $10^{-12}$ to anthropogenic level beyond $10^{-10}$ in modern environment due to the atmospheric nuclear weapon testing, nuclear accidents, nuclear fuel reprocessing process (Snyder et al., 2010). More than 95% of the environmental $^{129}$I was discharged by the two European nuclear fuel reprocessing plants (NFRP), Sellafield in United Kingdom and La Hague in France to the seas and air in liquid and gaseous forms. As a consequence of these point sources of $^{129}$I, the

distribution of $^{129}$I is rather uneven (Snyder et al., 2010). Atmospheric $^{129}$I investigations have been conducted in Europe, Japan, USA and Canada, but aerosol $^{129}$I studies are still rare, and no aerosol $^{129}$I data is available in China at present (Hasegawa et al., 2017; Hou et al., 2009; Jabbar et al., 2013; Moran et al., 1999; Toyama et al., 2013; Xu et al., 2013). The previous studies present the time series of $^{129}$I in aerosols in monthly resolution for the purpose of nuclear environmental monitoring, while the low time-resolution is not sufficient to understand the source, transport and temporal variation pattern and its influencing factor

of $^{129}$I.

Here, we present a day-resolution temporal variation of $^{129}$I and $^{127}$I in aerosols during 2017/2018 from a typical monsoonal zone, Xi'an city in the Guanzhong Basin of northwest of China, to make attempts to investigate the level, sources and temporal change characteristics of $^{127}$I and $^{129}$I, to establish a background value of $^{129}$I/$^{127}$I ratio serving the nuclear environmental safety monitoring, as well as to make clear key influencing factors including meteorological parameters, East Asian monsoon (EAM)

and heavy haze events.

## 2 Materials and methods

The aerosol samples were collected by a high-volume sampler on the roof of the Xi'an AMS Centre in Xi'an, China (34°13'25"N, 109°0'0"E) with an elevation of 440 m above mean sea level (Fig.1). Xi'an, located in the Guanzhong basin, is the largest city in northwest China with a population of 9.9 million. The basin is nestled between the Qin Ling in the south and

the Loess Plateau in the north, and is warm temperate zone with semi-humid continental monsoon climate (Fig.1b).

Sixty eight aerosol samples were selected for measurement of iodine isotopes using the pyrolysis combing with AgI-AgCl coprecipitation for separation and accelerator mass spectrometry (AMS, 3MV, HVEE, the Netherland) and inductively coupled





plasma mass spectrometry (ICP-MS, Agilent 8800, USA) for determination of $^{129}I/^{127}I$ ratios and $^{127}I$ concentrations, respectively, as previously reported (Zhang et al., 2018b). The sample collection and preparation procedure are described in

detail in the supplementary information (SI-1). $^{129}I/^{127}I$ ratios of the iodine carrier are determined to be less than $2\times10^{-13}$, and the analytical precision was less than 5% for all the samples.

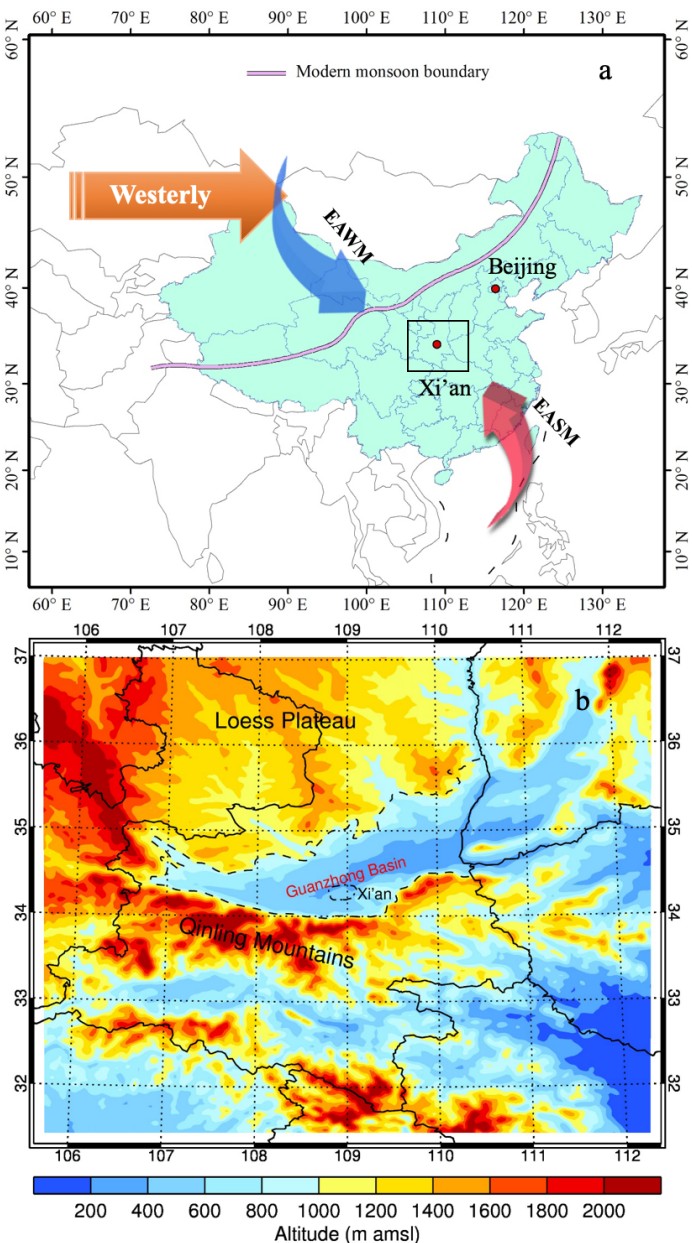

**Fig.1** Mapping shown the sampling location, East Asian monsoon (EAM) system and topography (a) China, (b) Xi'an city in the Guanzhong Basin between the Loess Plateau to the north and Qinling Mountains to the south. East Asian monsoon (EAM),
constituted by East Asian summer monsoon (EASM) and East Asian winter monsoon (EAWM), is one of vital components of the





global atmospheric circulation system. The pink line in the upper panel shows the modern monsoon boundary, and the arrows indicate the westerly (orange), the EAWM (blue) and the EASM (red).

## 3 Results

Results of $^{127}$I and $^{129}$I concentrations, $^{129}$I/$^{127}$I atomic ration in aerosol samples in Xi'an, China from March 2017 to March 2018, are shown in Fig.2 and Table S1 in Supporting Information. Concentrations of $^{127}$I and $^{129}$I and $^{129}$I/$^{127}$I atomic ratios in aerosol samples from Xi'an fell within 1.21-21.4 µg m$^{-3}$, (0.13-7.53) ×10$^5$ atoms m$^{-3}$, and (10.6-743) ×10$^{-10}$, respectively. The mean values were 6.22±4.48 µg m$^{-3}$, (2.22±1.87) ×10$^5$ atoms m$^{-3}$, and (101±124) ×10$^{-10}$ for $^{127}$I, $^{129}$I concentrations and $^{129}$I/$^{127}$I atomic ratios, respectively.

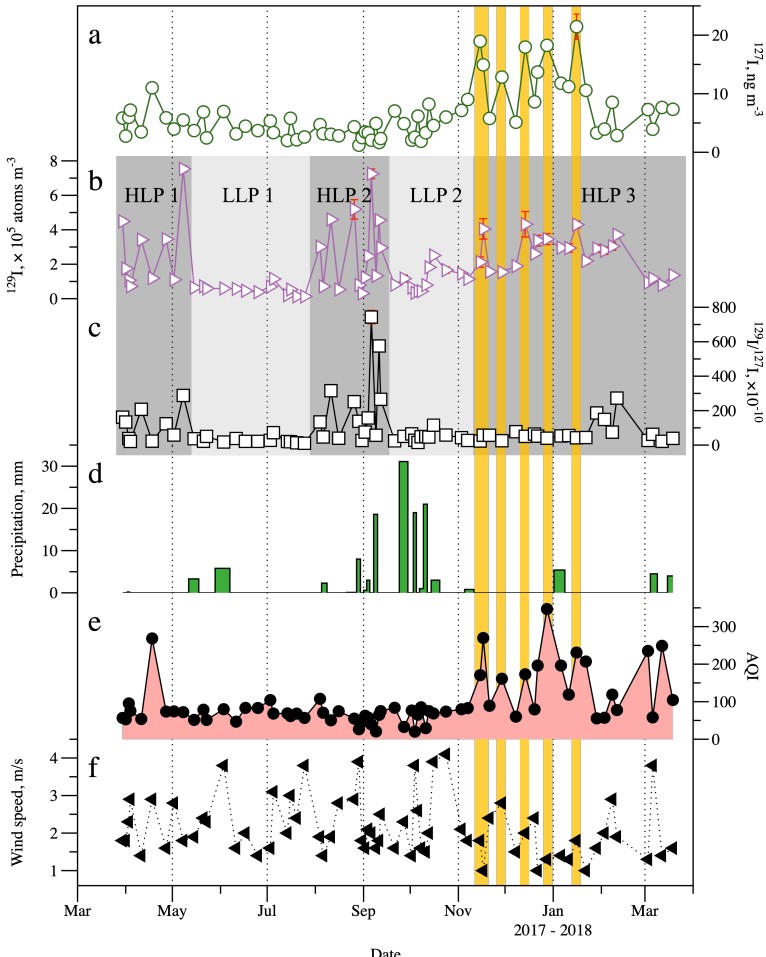

**Fig.2 Temporal variation of $^{127}$I (a), $^{129}$I (b) and $^{129}$I/$^{127}$I ratios (c) in aerosol samples collected in Xi'an, China from March 2017 to March 2018. The meteorological and air quality data includes precipitation (d), Air quality index (AQI, e) and wind speed (f). Orange**


**bands indicate five heavy haze episodes corresponding with five $^{127}$I peaks. Three dark and two light grey shades in b and c**
**demonstrate the high-level and low-level periods (HLP and LLP), respectively, for $^{129}$I and $^{129}$I/$^{127}$I ratios, alternatively dominated**
**by the EAWM and EASM, respectively.**

$^{127}$I and $^{129}$I in aerosols are characterized with the apparent monthly and seasonal variations, which are described in SI-1 and
SI-2 in detail (Fig.S1 and S2). In general, the concentrations of $^{127}$I and $^{129}$I are found highest in winter with maximum in
December and September, respectively, and lowest in summer with minimum in August and July, respectively, and in between
for spring and fall.

A weak correlation between $^{129}$I and $^{127}$I was found with a Pearson correlation coefficient of 0.34 (p=0.01) for the whole year
data, while no significant correlation between the two iodine isotopes in each season at the level of 0.05 (Table 1 and Fig. S3).
The correlation analysis between iodine isotopes and total suspended particle (TSP) indicate that there was a strong correlation
between $^{127}$I and TSP, while no correlation between radioactive $^{129}$I and TSP (Fig. S4).

**4 Discussion**

**4.1 Level and sources of $^{127}$I and $^{129}$I**

Although a weak correlation was observed between $^{127}$I and $^{129}$I in the whole year sampling, there were no correlations between
the two isotopes in each season, indicating $^{127}$I and $^{129}$I have different sources and influence factors.

**4.1.1 $^{127}$I**

The level of $^{127}$I concentrations, in particular in winter, is much higher than those in terrestrial air (1 ng m$^{-3}$), and also slightly
higher than the marine air (< 10 ng m$^{-3}$) (Saiz-Lopez et al., 2012). Whereas, a similar range of $^{127}$I was observed to be 4.5-22
ng m$^{-3}$ at coastal urban and Shengsi Island of Shanghai, China (Cheng et al., 2017; Gao et al., 2010). This suggests that a
relatively higher $^{127}$I level in aerosols in both inland and coastal cities in China.

Iodine in urban air generally origins from natural and anthropogenic sources. Natural iodine is from marine emission through
sea spray, weathering of base rock and continental release through vegetation and suspended soil particles (Fuge and Johnson,
1986). Due to the influence of southeasterly EASM, moisture from the Pacific Ocean and the Chinese seas might bring oceanic
iodine. Whereas, the mean $^{127}$I concentration in summer aerosol is 3.61±1.49 µg m$^{-3}$, about three-fold lower than that in winter.
The sampling location, Xi'an, is an inland city about 900 km away from the nearest coastline. The contribution of oceanic
iodine to terrestrial surface system in winter is considered to be negligible when the site is over 400 km away from the ocean
(Cohen, 1985). Taking sodium and calcium as reference elements for sea spray and direct volatilization of iodine from the
ocean and weathering of soil and rock, respectively, He et al. (2012) has been estimated that less than 0.04% and 5.2% of
iodine were from the direct contribution of ocean and weathering of soil and rock to the precipitation at Zhouzhi county, Xi'an
city (He, 2012).



Iodine is also emitted from volatility of terrestrial soil and respiration of vegetation, which was estimated to be 2.27 μg m$^{-2}$ d$^{-1}$ in the form of $CH_3I$ (Sive et al., 2007). Dry deposition of iodine, however, can be calculated to be 8.78-39.6 μg m$^{-2}$ d$^{-1}$ based on aerosol $^{127}I$ concentrations in this study and an average dust fall flux of 13.2 t (km$^{-2}$ 30 d$^{-1}$) from the "2017 Xi'an Environmental bulletin" (Xi'an Bureau of Statistics, 2018). The iodine deposition was far beyond terrestrial sources of soil and vegetations, indicates they might be major iodine sources in summer, but not in winter.

The significant increase of $^{127}I$ from summer to winter suggests that anthropogenic discharge of iodine is the dominant source of $^{127}I$ in Xi'an aerosol samples, mainly including combustion of biomass and fossil fuel (Wu et al., 2014). Biomass combustion generally occurs in summer harvest time, normally in later May and early June. In order to improve air quality, Xi'an government has banned biomass combustion since 2009. Additionally, no obvious change in $^{127}I$ concentrations was found in May and June, indicating the biomass combustion is not the major source.

A recent study has confirmed that particulate iodine around two coal plants in Nanchang city, China, was greatly increased up to 36 ng m$^{-3}$, and iodine concentrations within 9 km from the coal plants were much higher than that in non-coal sites (Duan, 2018). Coal consumption accounts for 72.7% of total energy consumption in Shaanxi province in 2013. Coal is dominant in energy consumption structure. In 2017, the coal consumption in Guanzhong basin is 67.4 million tons (Shaanxi Provincial Bureau of Statistics, 2018). $^{127}I$ concentration in coal produced in Shaanxi province ranges from 0.39 to 6.53 μg g$^{-1}$ with a mean value of 1.47 μg g$^{-1}$ (Wu et al., 2014). An atmospheric iodine emission factor that equals to the ratio of the iodine released into the atmospheric from the coal is from 78.8% to 99.4%, depending on the coal combustion technology and emission control devices (Wu et al., 2014). If simply assuming anthropogenic iodine is solely from combustion of coal in our study area and the atmospheric iodine emission factor is 92%, about 91 tons of $^{127}I$ can be released to the atmosphere in the Guanzhong Basin in 2017. The area of the Guanzhong Basin is 3.6×10$^4$ m$^2$, and the height of troposphere is taking as 10 km. Then, $^{127}I$ concentration in the air is about 250 ng m$^{-3}$. The particle-associated iodine accounts for approximately 10%-20% (Hasegawa et al., 2017). Thus, $^{127}I$ in aerosols can be estimated to be about 25-50 ng m$^{-3}$. The estimated value is comparable with the $^{127}I$ peak values in winter, but about ten times higher than the less polluted aerosol $^{127}I$ concentrations (1.21-9.01 ng m$^{-3}$). Xi'an, a northern city in China, consumes more coals in the heating period from November 15 to March 15, which aggravates the iodine release from coal combustion. Thus, we suggested that coal combustion is the major source of $^{127}I$ in Xi'an urban aerosols in particular during the heating period of winter, and more than 60% of coal-derived iodine has been dispersed out of the Guanzhong Basin. This also suggests that $^{127}I$ was regionally or locally input, and can be treated as internal release.

### 4.2.2 $^{129}I$

The aerosol $^{129}I$ levels reported in the previous studies and this work could be categorized into three groups (Fig.3). 1) Compared to other investigating sites, aerosol $^{129}I$ concentrations were less than 10×10$^5$ atoms m$^{-3}$ in Xi'an, northwest China. This low level is also found at those sites remote from the nuclear facilities in southern and central Europe, as well as Japan before the Fukushima accident (Hasegawa et al., 2017; Jabbar et al., 2013; Santos et al., 2005). The lowest $^{129}I$ (< 0.1×10$^5$ atoms m$^{-3}$) in aerosols have been found at two high altitude sites of Alps mountains (about 3000 m above the sea level). 2) The





high values beyond $1000 \times 10^5$ atoms m$^{-3}$ have been reported at the sites directly contaminated either by nuclear reprocessing plants, such as Hanford, Sellafield and WAK at Karlsruhe, or by Fukushima nuclear accident in 2011 (Brauer et al., 1973; Jackson et al., 2002; Wershofen and Aumann, 1989; Xu et al., 2015). 3) In between, aerosol $^{129}$I within the range from $10 \times 10^5$

atoms m$^{-3}$ to $1000 \times 10^5$ atoms m$^{-3}$, are mainly found in the sites and periods with global fallout from atmospheric nuclear weapon testing, and indirectly contaminations from nuclear fuel reprocessing plants (Brauer et al., 1973; Englund et al., 2010; Kadowaki et al., 2018; Tsukada et al., 1991; Zhang et al., 2016).

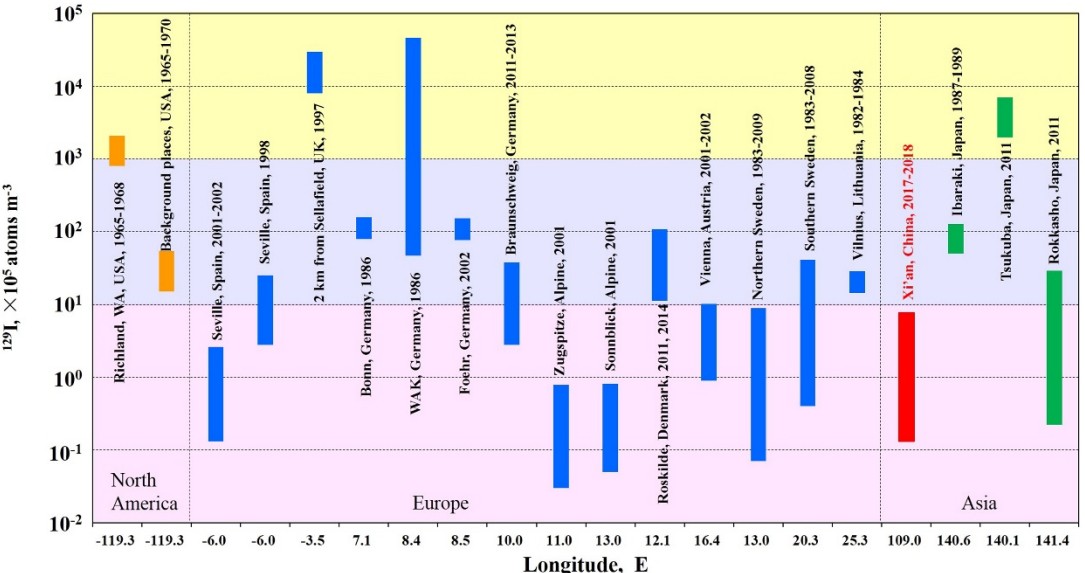

**Fig.3 Comparison of aerosol $^{129}$I level in Xi'an, China (red bars) with other investigations in North America (orange), Europe (blue)**
**and East Asia (Green) distributed by longitude.**

The source term of $^{129}$I is crucial for spatial and temporal distributions of $^{129}$I in global scale. In the Xi'an aerosols, $^{129}$I/$^{127}$I atomic ratios range from $10.6 \times 10^{-10}$ to $743 \times 10^{-10}$, which is at least three orders of magnitude higher than the level of naturally produced $^{129}$I ($1.5 \times 10^{-12}$) (Fehn et al., 2005), indicating human nuclear activities are dominant contributor for the increase of

$^{129}$I level in the environment. The level and source of $^{129}$I in soil, vegetation, rain and rivers water samples have been previously investigated in Xi'an region, where $^{129}$I/$^{127}$I varied from $1.1 \times 10^{-10}$ to $43.5 \times 10^{-10}$ with a mean value of $20.6 \times 10^{-10}$ (Zhang et al., 2011). $^{129}$I/$^{127}$I ratios in aerosols were about one order of magnitude higher than those in other environmental samples, indicating $^{129}$I in Xi'an aerosols was not released by local soil suspension and vegetation release. Coal combustion contributes a large proportion of stable $^{127}$I in winter, while $^{129}$I amount in coals is almost negligible because coal was formed in Tertiary

(2.58-66 million years) at the latest so that $^{129}$I has been decayed out or in an extremely low value of $10^{-13} \sim 10^{-10}$ for $^{129}$I/$^{127}$I. Thus, coal combustion is not a major source of atmospheric $^{129}$I.





Nuclear activities including the historic nuclear weapon testing sites, nuclear reactors, nuclear spent fuel reprocessing plant (NFRP) in China and Europe, as well as the underground nuclear weapon testing are considered. Two nuclear weapon testing sites, Semipalatinsk and Lop Nor, locating upwind, may input $^{129}$I into Xi'an region through soil resuspension and gaseous re-emission. However, evidence from $^{129}$I distribution in surface soils from upwind regions reveals that the two nuclear weapon testing sites has limit impact on the atmospheric $^{129}$I level in the remote regions farther than 1000 km from these test sites (Fan, 2013). This is also supported by the back-trajectory analysis that $^{129}$I concentration did not significantly raised when abundant air masses from Xinjiang passing through the Lop Nor test site on December 28, 2018 (Fig.S5g). Five nuclear power plants are in operation along the southeast coastal areas in China. $^{129}$I data in sea water collected within 10 km from a Chinese nuclear power plant suggests that normal operation of reactors does not have significant increase in $^{129}$I concentrations (He et al., 2011). Although information on gaseous release of $^{129}$I from these reactors is unknown, the low $^{129}$I/$^{127}$I (about $7\times10^{-10}$) in the surface soil of southern China (Guangxi, Jiangxi and Fujian Provinces) close to the reactors can confirm that there is no marked deposition from the gaseous release (Fan, 2013). Toyama et al. (2012) have shown a direct close-in influence of a pilot plant in Tokaimura (Ibaraki Prefecture), Japan on the $^{129}$I deposition in Tokyo (Toyama et al., 2013). Similarly, a pilot nuclear spent fuel reprocessing plant (NFRP) has been established and operated in Gansu province, China since 2010. This NFRP is locating in an upwind area and about 1200 km northeast of Xi'an. During the sampling period in 2017/2018, no abnormally high $^{129}$I was observed, while this contribution cannot be neglected in the future operation, and should be continuously monitored. In addition, the possible influence of the sixth underground nuclear weapon test conducted by North Korea on September 3, 2017 has been excluded based on the back and forward trajectories and the nuclear environmental monitoring around the Chinese northeast border by the government (Ministry of Environmental Protection of the People's Republic of China, 2017).

It is well documented that gaseous and liquid discharges from the nuclear fuel reprocessing plants (NFRPs) in Sellafield, United Kingdom and La Hague, France, as well as the secondary emission from the contaminated seas and land, are the predominant source of $^{129}$I in the modern atmosphere, in particular in European environment (Jabbar et al., 2013). The two NFRPs are located in the 50-55°N, the westerly belt. The prevailing westerly winds throughout the year in the mid-latitude act as a crucial pathway of $^{129}$I transport from its source to the whole mid-latitude regions of the northern hemisphere, as observed in the sediment core from Jiaozhou Bay, east coast of China (Fan et al., 2016). The 60-year record of $^{129}$I in a lacustrine sediment from Philippines further shows that the EAWM plays an important role in transporting the mid-latitude $^{129}$I to the low-latitude regions (Zhang et al., 2018a). The feature of $^{129}$I variation also shows that $^{129}$I was in high level in spring and winter when EAWM prevailing and low level in summer when EASM prevailing, supporting that the $^{129}$I is dominantly sourced from the long-range transport of European NFRPs discharges. In this case, $^{129}$I is externally input relative to locally released stable $^{127}$I.

**4.2 Factors influencing temporal variation of iodine isotopes**



As discussed above, even though variation pattern of $^{127}I$ and $^{129}I$ were similar, they were considerably influenced by many factors owing to their different sources. In this work, meteorological factors including precipitation, wind speed, temperature and dust storm events, atmospheric circulation (in particular EAM), heavy air pollution periods are discussed.

### 4.2.1 Meteorological factors

**Precipitation and wind speed.** As discussed in supplementary information (SI-3), the influences of precipitation and wind speed on temporal changes of iodine isotopes are not significant (Fig.2e and 2f). However, the winter days with absence of wet precipitation and lower wind speed well corresponded to the heavy haze episodes when iodine concentrations, in particular stable $^{127}I$, were greatly increased, indicative of less dispersion. The details about haze influence on iodine will also be discussed in the other following section.

**Temperature.** Temperature and its associated physiochemical processes and biological release of iodine from source regions might be a reason for the variation patterns. In summer, the temperature is from 20-40°C in the north hemisphere, which is favourable for direct volatilization of iodine from the surfaces of land and seas. Ozone in air-sea boundary layer is suggested to act as an oxidants to transform iodide in seawater to volatile molecular iodine that enters into the air, which is believed more significant than the biological process (Carpenter et al., 2013). Ozone concentrations in summer is around 30 pptv, roughly two times higher than winter (Ayers et al., 1996), which may increase the re-emission rate of iodine from the ocean and $^{129}I$-contaminated sea surface into the air. Additionally, the bloom of phytoplankton and algae in summer, can release biogenic organic iodine into the air through a mechanism of anti-oxidation (Küpper et al., 2008). The temperature, ozone concentration and marine biomass greatly reduces in winter, which will result in less iodine released from the source regions, and can be used to explain the relatively weak peaks in winter than in summer. As discussed above, $^{127}I$ and $^{129}I$ in Xi'an aerosols were mainly derived from coal combustion and long-range transport from Europe. The change in release amount of $^{127}I$ and $^{129}I$ at the source regions is obviously not the determining factor for the changes of iodine isotopes since Xi'an is far from the oceans and the $^{129}I$ source regions. Furthermore, the seasonal variation of $^{127}I$ and $^{129}I$ with low level in summer can also easily exclude the possibility of temperature influence.

**Dust storm.** Two severe dust storm events occurred in Xi'an in 17-18 April and 4-6 May, 2017, as shown by the peaks of air quality index (AQI) of 268 and 306, respectively (Fig. 2e). A $^{127}I$ peak, 11.0 ng m$^{-3}$, was observed on 18 April, 2017, while $^{127}I$ levels in other samples were almost below 6 ng m$^{-3}$ in spring and summer time. Dust storms frequently occur in winter and spring in north China, and normally originate from the arid and semi-arid desert regions mainly locating in Mongolia and northwest China. The first dust storm arrived the Guanzhong basin on 17 April 2017, and lasted until 19 April (China Meteororological Administration, 2017). The small peak of $^{127}I$ is likely attributed to the suspended particulate matter from the soil surface in the dust storm source. In contrast, variation of $^{129}I$ level did not reflect the dust storm influence. The fact that $^{129}I$ was not correlated with particulate concentrations (Fig.S4), indicates that the extrinsic $^{129}I$ is not related to the heavy particulate events, since the major dust source areas include Taklimakan desert, the Gobi Desert in Inner Mongolia, and the Loess Plateau, where the $^{129}I/^{127}I$ ratios in surface soil fell below $60\times10^{-10}$, apparently much lower than those in aerosols (Zhang





et al., 2011). Meantime the back trajectory analysis also showed that the low $^{129}$I level on April 18 can be partially attributed to a low-altitude air mass (< 900m) (Fig.S5a).

The second dust storm has started from the south-central Mongolia and the west-central Inner Mongolia autonomous region since 3 May, arrived at Xi'an on 5 May and retreated on 6 May. It is pity that no sample was analysed in this event, but

a significant $^{129}$I peak with value of $7.53\times10^5$ atoms m$^{-3}$ was found after three days of this event (Fig. 2b). The back trajectory analysis suggests the $^{129}$I peak on May 8, 2017 is found to relate to the downdraft originated from high altitude (2000-6000 m) to low altitude (500 m) (Fig.S5b). This elevation of $^{129}$I after the dust storm events is likely attributed that the intensified winter monsoon and strong cold high pressure transporting greater $^{129}$I from Europe to China.

### 4.2.2 Heavy haze episodes during 2017/2018 winter

A significantly positive correlation between $^{127}$I and air quality index (AQI) was found with a high Pearson correlation coefficient of 0.79 (p<<0.05) for the whole-year sampling period, and an increased coefficient of 0.84 in winter (Table 1). The $^{127}$I concentration in winter can reach to 10 times as much as in summer (Fig. 2a). Furthermore, five $^{127}$I peaks from 12.8 to 21.4 ng m$^{-3}$ were clearly identified on 15 and 29 November, 14 and 28 December, and 16 January, respectively, which well coincided with the heavy haze episodes with AQI mostly over 200, namely heavily polluted air (Fig. 2e). As discussed in

section 4.1, the irrelevance between $^{127}$I and $^{129}$I in aerosols attributed to their different sources, also demonstrates that locally discharged iodine and externally input iodine are not contemporaneously subjected to formation of iodine-containing particles. Typically, new particle formation occurs in two distinct stages, i.e., nucleation to form a critical nucleus and subsequent growth of the freshly nucleated particle to a larger size (Zhang et al., 2015). It is widely accepted that iodine is involved into the formation of fine particles, and increasing investigations have been carried out in coastal and open sea areas (Saiz-Lopez et

al., 2012). However, in megacities with severe air pollution, the role of iodine on formation and development of heavy haze events is far not understood. Iodine-mediated particles were suggested to be formed from highly concentrated, localized pockets of iodine oxides as primary nucleation, and to rapidly grow by uptake of $H_2SO_4$, $H_2O$, $NO_2$, short chain dicarboxylic acids, gaseous iodine and other gaseous species (Saiz-Lopez et al., 2012). Winter urban air in Xi'an provides two requirements of sufficiently high iodine concentrations and the presence of high levels of aerosol nucleation precursors, such as $SO_2$, $NH_3$,

amines, and anthropogenic VOCs.

Further analysis showed close relationship between $^{127}$I and six air pollutants, including PM 10, PM 2.5, CO, $SO_2$, $NO_2$ and $O_3$ (Table 1 and Fig. S6). In spring and summer, the high correlation between $^{127}$I and AQI can be attributed to the high correlation between $^{127}$I with PM10 and PM2.5. In fall and winter, $^{127}$I, is significantly positively correlated with PM 10, PM 2.5, CO, $SO_2$ and $NO_2$, and negatively correlated with $O_3$ (Pearson correlation coefficient =-0.60, p=0.02). In contrast, there is

no such good agreement between $^{129}$I and these gaseous pollutants. Despite that, three $^{129}$I peaks were found on 15 November, 14 December, 2017 and 16 January 2018, respectively, which well corresponded with high $^{127}$I concentrations (Fig. 2a and 2b) during the haze episodes. This reflects that the formation mechanism of iodine-containing aerosols might be seasonally different. In spring and summer, iodine is probably associated with primary matters and secondary organic aerosols due to low




level of air iodine and greatly increased artificial and biogenic VOCs (Feng et al., 2016). In fall and winter when the key aerosol nucleation precursors are noticeably elevated, the significantly positive correlation between [127]I and these precursors indicates that locally emitted iodine is likely involved into formation of secondary inorganic aerosols, while externally input [129]I may not occur in the nucleation of secondary inorganic aerosols. However, the three peaks of [129]I in aerosols during the heavy haze episodes suggest that local and external iodine are subjected to subsequent growth of particles due to a longer residence time in stagnant weather conditions. The minimum in ozone concentrations on 15 November and 14 December, 2017 may support iodine-containing aerosol nucleation process, in which ozone acted as oxidant and reactant to form iodine oxidizes, and aggregated into high valence iodine oxidizes (Saiz-Lopez et al., 2012). This study suggests iodine is closely related to aerosol formations, and high level of iodine likely facilitates the growth of fine particles along with major aerosol precursors particularly during haze episodes.

**Table 1. Pearson correlation coefficients between iodine isotopes and atmospheric pollutants and weather conditions** *.

| Correlation | Whole year [127]I | | [129]I | | Spring (3-5) [127]I | | [129]I | | Summer (6-8) [127]I | | [129]I | | Fall (9-11) [127]I | | [129]I | | Winter (12-2) [127]I | | [129]I | |
|---|---|---|---|---|---|---|---|---|---|---|---|---|---|---|---|---|---|---|---|---|
| | Pears. | Sig. | Pears. | Sig. | Pears. | Sig. | Pears. | Sig. | Pears. | Sig. | Pears. | Sig. | Pears. | Sig. | Pears. | Sig. | Pears. | Sig. | Pears. | Sig. |
| [129]I | **0.34** | **0.01** | | | -0.05 | 0.86 | | | 0.15 | 0.56 | | | -0.01 | 0.97 | | | 0.31 | 0.30 | | |
| [129]I/[127]I | -0.29 | 0.02 | **0.68** | **0.00** | -0.35 | 0.19 | **0.92** | **0.00** | -0.08 | 0.76 | **0.94** | **0.00** | -0.39 | 0.07 | **0.87** | **0.00** | **-0.69** | **0.01** | 0.33 | 0.27 |
| Temp | **-0.54** | **0.00** | **-0.46** | **0.00** | 0.10 | 0.70 | -0.02 | 0.93 | -0.06 | 0.82 | -0.18 | 0.50 | **-0.61** | **0.00** | 0.22 | 0.33 | 0.31 | 0.30 | -0.28 | 0.36 |
| Humidity | -0.04 | 0.73 | -0.17 | 0.17 | 0.14 | 0.60 | -0.42 | 0.10 | -0.13 | 0.62 | 0.17 | 0.52 | -0.43 | 0.05 | -0.20 | 0.38 | **0.60** | **0.03** | 0.31 | 0.31 |
| Wind speed | -0.24 | 0.05 | **-0.26** | **0.04** | 0.03 | 0.93 | -0.35 | 0.18 | 0.05 | 0.85 | -0.03 | 0.90 | -0.15 | 0.51 | -0.11 | 0.63 | -0.23 | 0.46 | 0.21 | 0.49 |
| Precipitation | -0.14 | 0.24 | -0.20 | 0.10 | -0.14 | 0.60 | -0.17 | 0.53 | -0.01 | 0.96 | -0.13 | 0.62 | -0.23 | 0.30 | -0.28 | 0.20 | 0.06 | 0.84 | 0.02 | 0.94 |
| AQI | **0.79** | **0.00** | **0.24** | **0.05** | **0.77** | **0.00** | -0.21 | 0.44 | **0.58** | **0.02** | -0.06 | 0.82 | **0.80** | **0.00** | 0.19 | 0.41 | **0.84** | **0.00** | 0.16 | 0.60 |
| CO | **0.69** | **0.00** | 0.19 | 0.12 | 0.49 | 0.06 | 0.01 | 0.96 | 0.16 | 0.53 | **0.50** | **0.04** | **0.60** | **0.00** | -0.11 | 0.61 | **0.84** | **0.00** | 0.06 | 0.85 |
| SO₂ | **0.72** | **0.00** | **0.45** | **0.00** | 0.09 | 0.74 | -0.14 | 0.59 | 0.29 | 0.26 | 0.13 | 0.62 | **0.84** | **0.00** | 0.17 | 0.45 | 0.51 | 0.08 | 0.04 | 0.89 |
| NO₂ | **0.71** | **0.00** | **0.35** | **0.00** | 0.37 | 0.16 | -0.10 | 0.70 | 0.05 | 0.85 | 0.26 | 0.31 | **0.69** | **0.00** | -0.03 | 0.89 | **0.63** | **0.02** | -0.02 | 0.96 |
| O₃ | **-0.42** | **0.00** | **-0.28** | **0.02** | -0.21 | 0.45 | -0.01 | 0.96 | 0.39 | 0.12 | -0.39 | 0.12 | -0.34 | 0.13 | 0.36 | 0.10 | **-0.66** | **0.02** | 0.37 | 0.21 |
| PM10 | **0.73** | **0.00** | 0.19 | 0.12 | **0.82** | **0.00** | -0.20 | 0.47 | **0.67** | **0.00** | 0.05 | 0.86 | **0.75** | **0.00** | 0.12 | 0.59 | **0.80** | **0.00** | 0.20 | 0.52 |
| PM2.5 | **0.81** | **0.00** | **0.27** | **0.03** | **0.63** | **0.01** | -0.19 | 0.48 | **0.74** | **0.00** | 0.04 | 0.88 | **0.78** | **0.00** | 0.14 | 0.53 | **0.84** | **0.00** | 0.14 | 0.65 |

* Pearson correlation coefficient. Correlation significant at the 0.05 level is in bold.

### 4.2.3 Impact of EAM for long-range transport of [129]I

Increasing evidence have suggested that the prevailing westerly and EAM system act as crucial driving forces and pathways for transport of the European NFRPs derived [129]I from Europe to East Asia and even to low-latitude southeast Asia (Fan et al., 2016; Zhang et al., 2018a). Monthly variations of atmospheric [129]I in Japan also showed a clear pattern with low [129]I deposition in summer and high in winter, which is also attributed to the impact of EAM (Hasegawa et al., 2017; Kadowaki et al., 2018; Toyama et al., 2013). In this work, seasonal variation of [129]I was identical to the observation in the previous studies (Toyama et al., 2013). However, the day-resolution variation patterns of [129]I and [129]I/[127]I in Xi'an, distinct from monthly variation in Japan, showed three periods with high levels and two periods with low levels, indicating more complex influence of EAM in the typically continental monsoon climate city, Xi'an.





The whole-year time series can be divided into five periods with three high-level periods (HLP), a) from late March to early May (HLP 1), b) from middle August to early September (HLP 2), and c) from middle November, 2017 to late February, 2018

(HLP 3); as well as two low-level periods (LLP), d) from early May to middle August (LLP 1), and e) from middle September to early November, 2017 (LLP 2) (Fig.2c and 2c). $^{129}$I levels in the three HLPs fell within the range of $(1.98\text{-}2.41) \times 10^5$ atoms m$^{-3}$, which is 3-5 times higher than those during the two LLPs with $(0.49\text{-}0.66) \times 10^5$ atoms m$^{-3}$ (Table S2). The relative standard deviation shows much higher variability during HLP 1 and 2 from 91% to 109% in contrast to the variability in other clusters less than 60%.

The significant difference between the HLPs and LLPs suggests the transportation process of $^{129}$I is obviously distinct. The westerly is a crucial driving force of $^{129}$I from the NFRPs point sources and their contaminated seas, and labelled by a high $^{129}$I level up to $10^{-6}$ for $^{129}$I/$^{127}$I ratio (Michel et al., 2012; Zhang et al., 2016) (Fig.1a). Due to interplay between westerly and EAWM (An et al., 2012), EAWM inherits the high $^{129}$I feature of $10^{-7}\text{-}10^{-9}$ for $^{129}$I/$^{127}$I ratio in the long-distance transport process. Therefore, the HLP 1 and 3 was strongly affected by the EAWM prevailing from early September to early may in

2017. Compared to the violent fluctuation of $^{129}$I in spring (HLP1), the weak fluctuations of HLP 3 in winter might be attributed to a relatively stable interaction process between the strengthened westerly and the EAWM. In addition, the $^{129}$I level in March 2018 was much less than that in March 2017, seems to be consequences of EAWM in March 2018 that was weaker than in March 2017. This is in good agreement with the EAWM index of 2.04 in 2017 and -1.86 in 2018 (MODES forecast motor (NCEP I), 2019). The HLP 2 was not the case as HLPs 1 and 3, since the period was under the control of EASM.

The EASM origins from the Pacific and Indian tropical under the role of subtropical highs, and transports moisture from the ocean to East Asia since early summer. $^{129}$I/$^{127}$I ratios in the Pacific Ocean, the East China Seas, and the Indian Ocean are as low as $10^{-10}$ (Liu et al., 2016; Povinec et al., 2011). Even after the Fukushima accident, $^{129}$I/$^{127}$I ratios are still less than $40 \times 10^{-10}$ in the western Pacific Ocean (Guilderson et al., 2014). Thus, EASM is poor in $^{129}$I in comparison to the winter monsoon. This is well in agreement with the low $^{129}$I level during the two LLPs (Fig. 2b). The 850 hPa water vapor transmission flow

field showed that the southeast wind moisture moving northward to the north of 35°N May 2, followed by another two outbreaks of on May 21 and June 3 (Fig.S7), indicative of EAWM retreat and EASM advance. During this period, $^{129}$I dropped abruptly from $3.45 \times 10^5$ atoms m$^{-3}$ on 27th April to $1.10 \times 10^5$ atoms m$^{-3}$ on 2nd May, followed by a maximum on 8th May, then have a sudden decline to $0.64 \times 10^5$ atoms m$^{-3}$ on 15th May. The violet fluctuation of $^{129}$I is likely caused by the onset of EASM is quite violent in a way of stepwise northward jumps, which is fully supported by the previous metrological

observations (Ding and Chan, 2005). As the EASM turned into the active stage since mid-May, $^{129}$I level was low and in a relatively stable state, as showed in the LLP 1.

After the active stage of EASM, however, it is out of the expectation that increased and variable $^{129}$I levels were observed from middle August to early September (HLP 2). The $^{129}$I peak on September 6, 2017 was the highest throughout the sampling year. The back-trajectory model shows that five low-altitude air masses (< 1000 m above ground level) from the Baltic Sea moved

fast eastward and arrived at the Guanzhong Basin within five days (Fig. S5e). The Baltic Sea contains high $^{129}$I concentration due to the water exchange with the North Sea that receives over 100 kg year$^{-1}$ $^{129}$I from La Hague and Sellafield NFRPs (Snyder




et al., 2010). Therefore, a $^{129}$I peak observed here indicates the $^{129}$I-enriched westerly has interplayed with the EASM, the latter of which was retreating to the south. It is reported that Xi'an enters into the EASM break stage during this time based on the rainfall data (Ding and Chan, 2005). The intensive interaction between westerly and EASM facilitates the formation of rainfall

at their confluence area, resulting in the drastically fluctuating $^{129}$I levels. Therefore, the elevated and variable $^{129}$I levels in HLP 2 can be attributed to the EASM break stage.

After the break stage with significant $^{129}$I fluctuation, the second LLP of $^{129}$I from 21st September to 11th October (LLP 2) occurred when the summer monsoon turns into the revival stage (Fig.2b). Despite lower than the break period, the $^{129}$I level in this period has slightly increased from $0.49 \times 10^5$ atoms m$^{-3}$ in the active stage to $0.66 \times 10^5$ atoms m$^{-3}$ in the revival stage. After

the active-break-revival cycle of summer monsoon reflected by low-high-low $^{129}$I level, the $^{129}$I level has stepwise increased since mid-October, suggesting the EAWM has taken the place of the EASM in the Guanzhong Basin, and last until march next year.

The influence of EAM on variation of $^{129}$I has been quantitatively characterized using z-score normalized values in supplementary information (SI-4 and Fig.S8), which has clearly confirms that the EAM plays a decisive role on the temporal

variation and long-range transport of not only $^{129}$I, but also other air pollutants (i.e. persisting organic pollutants, inorganic air pollutants) in Chinese monsoon-affected regions.

### 4.3 Atmospheric background level of $^{129}$I/$^{127}$I ratios

Both two iodine isotopes show apparently temporal changes in northwest China, while $^{129}$I/$^{127}$I ratios show relatively weak fluctuation (Fig.2c). The mean $^{129}$I/$^{127}$I ratio of $(101 \pm 124) \times 10^{-10}$ can be simply regarded as the atmospheric background level

of $^{129}$I in northwest China. The previous studies on $^{129}$I environmental baseline have never carefully investigate the influence of climate on time variation of $^{129}$I. Here our day-resolution $^{129}$I dataset in this monsoon climate city showed that time variation of the atmospheric baseline level related to metrological conditions, heavy haze events and atmospheric circulation, has to be carefully considered and used for better evaluation of the impact of possible nuclear incidents in a practical way. Particularly, a pilot nuclear reprocessing plants locating upwind to Xi'an, might be extended and will be a source of radionuclides in the

future. The baseline established in this work is, therefore, of significance to long-term monitor nuclear environmental safety, sensitive assess the impact of nuclear incidents and apply on environmental process tracing.

### 5 Conclusions

The study firstly presents a high-resolution temporal variation of atmospheric $^{127}$I and $^{129}$I in northwest China, showing the vivid seasonal characteristics of iodine isotopes and an $^{129}$I/$^{127}$I baseline ratio of $(101 \pm 129) \times 10^{-10}$. Variation of $^{127}$I strongly

linking with atmospheric pollutions and heavy haze episodes, in particular in winter, indicates that $^{127}$I in Xi'an aerosols mainly derives from combustion of fossil fuel. Aerosol $^{129}$I mainly originates from European nuclear reprocessing plants through long-range transport, and its temporal variation is strongly dominated by the interplay of East Asian winter and summer monsoon.



Previous studies on temporal changes of atmospheric $^{129}$I in other monsoonal regions showed a simple pattern with lowest level in summer and highest in winter, while our day-resolution dataset showed that high $^{129}$I level could be found in summer

time due to the break of East Asian summer monsoon. The locally input $^{127}$I and exogenous $^{129}$I were greatly increased during haze events, reflecting the possible role of iodine in the formation of urban fine particles, therefore, further investigations are expected to focus on the speciation of iodine isotopes for mechanism study of iodine's impact on air pollution.

## Supplement

Supplementary information accompanies this paper in a separate file.

**Author contribution**

LZ, XH and SX designed and optimized the experiment. LZ, and NC performed the experiment, with the help of PC and YF. TF collected the air pollutant data. LZ, TF, PC, and YF draw the figures. The data analysis and interpretation were carried out by LZ, XH, SX, TF and NC.  LZ prepared the paper, with contributions from all co-authors.

## Competing interests

The authors declare that they have no conflict of interests.

## Acknowledgement

This work was supported by the National Natural Science Foundation of China (No. 11605207 and 41603125), the Youth Innovation Promotion Association of CAS, the Ministry of Science and Technology basic project (No. 2015FY110800), the Bureau of International Co-operation, CAS (132B61KYSB20180003) and National Research Program for Key Issues in Air

Pollution Control (DQGG0105-02). L.Z. also gratefully acknowledges Dr. Q. Liu, Ms. M. Fang, Mr. L. Wang and Mr. J. Zhou in IEECAS for their kind help on AMS measurement, sample collection and figure drawing.

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
