# Peer review of "Temporal variation of 129I and 127I in aerosols from Xi'an, China: influence of East Asian monsoon and heavy haze events"

_Atmospheric Chemistry and Physics, 2019_

## Referee Comment (RC1) · Anonymous Referee #1 · 19 Oct 2019

The paper acp-2019-818-manuscript-version1 entitled " Temporal variation of 129I and 127I in aerosols from Xi'an, China: influence of East Asian monsoon and heavy haze events" provides interesting data for the distribution of iodine isotopes in the aerosols of part of China which are missing from the international data base. The paper has the potential for publication after revision as given below. 1. The paper needs some linguistic revision as there are many grammatic mistakes. 2. Lines 44-45 "As a consequence of these point sources of 129I, the distribution of 129I is rather uneven (Snyder et al., 2010)? Where? 3. The paragraph between lines 50 and 55 is long and difficult to follow and could be rewritten to focus on aims of the study. 4. It is not clear how much time is the "a day-resolution" sampling reflects in term of iodine residence time

in the atmosphere? 5. Figures 1 a and b can be combined in one figure. 6. The results part needs further additions from the supplementary data including Figs. S1 and S2. 7. Connection of iodine chemical forms (I-127 and I-129) from the sources and in the atmosphere may elucidate some of the inconclusive correlations and relationship to spatial and temporal atmospheric transport on short and long distances. 8. More elaboration of weathering of basement rocks as a source of I-129 will be interesting. 9. Addition of Figure S7 and S8 to the discussion section will enhance the understanding of the atmospheric transport pathways of the isotopes. 10. More details on the paragraph in lines 100-104 can add clarity to general statement with respect to 127-distribution in China. 11. The anthropogenic source for I-127 is mainly related to coal consumption (local source) whereas the I-129 source is mainly related to far away transport. It will be good to provide some details of how these isotopes are associated in the atmosphere with respect to airmasses altitude, chemistry and residence time of the isotope. 12. May be good to make the text in Figure 3 in larger font. 13. Although the authors pointed out the possible use of air masses transport to predict iodine sources and impact on future iodine distribution, it is still not clear how the iodine data enhance our understanding of the climate or atmospheric circulation.

---

## Referee Comment (RC2) · Anonymous Referee #2 · 24 Oct 2019

This manuscript reports the concentrations and ratios of 129-I and 127-I in aerosol samples collected over a period of approximately one year at Xi'an in China. The data are interpreted in terms of the dominant sources and transport pathways of these isotopes to the site, and the discussion considers the influence of the fluctuating modes of the East Asian Monsoon on the observed record. The subject matter is highly relevant for Atmospheric Chemistry and Physics and the authors have a track record of producing high quality data from the demanding measurements employed. However, the manuscript suffers from a number of shortcomings, including factual errors and subjective and unsupported interpretations. I think that this will be an excellent contribution once these have been addressed. There are many minor errors in the English used,

but the meaning of the manuscript is still clear.

Major comments There are numerous instances of inconsistent units being used for iodine concentrations for the Xi'an site in the Results section (line 77 onwards). In the text, the units are frequently given as micrograms per cubic metre, while in the figures and supplementary material the units are nanograms per cubic metre. Since the numbers in both cases are the same, one of the units must be incorrect. I assume that the units should actually be nanograms per cubic metre, but please check and correct.

On line 91 the authors state that "A weak correlation between 129I and 127I was found with a Pearson correlation coefficient of 0.34 (p=0.01) for the whole year data, while no significant correlation between the two iodine isotopes in each season at the level of 0.05 (Table 1 and Fig. S3)." This does not agree with the statement made in the caption to Fig S3: "Relationship between 127I and 129I, showing no significant correlation (R=0.265) between the two iodine isotopes". Why do these statements not agree? Since it is apparent from Fig S3 that the dataset is not normally distributed, I would suggest the authors use a non-parametric regression method (such as Spearman's Rank Correlation) instead of Pearson's for all regression analysis in the manuscript. This will give far more robust results. Perhaps Figure S3 might be more informative if plotted with different symbols for the time periods of interest.

I am not quite sure how the authors have used the values published in Saiz-Lopez et al., 2012 to compare to the results obtained at Xi'an. In Table 5 of Saiz-Lopez, aerosol iodine concentrations of up to 25 ng m-3 and >3.3 ng m-3 are quoted for open ocean and continental sites respectively. These do not seem to relate to the values for "terrestrial" (1 ng m-3) and "marine" (<10 ng m-3) air quoted in lines 100 & 101. The higher values in Saiz-Lopez et al. also do not give strong support to the statement in the last sentence of this paragraph (lines 102-103).

The statement about natural sources of iodine (lines 104-105) comes from a rather

old source (Fuge & Johnson, 1986). While it is true that sea spray contributes iodine to the atmosphere, we now know that gas-phase emissions of iodine from the ocean are a much stronger source (see, for example, Carpenter et al., 2013 – which the authors cite later in the manuscript). Thus the study of He (2012) which apparently used sodium concentrations to estimate the seaspray contribution of iodine to precipitation at Zhouzhi county almost certainly greatly underestimated the "direct contribution of ocean". (The citation of He 2012 in the reference list does not give sufficient information for the source to be found).

In lines 114 - 118, the authors attempt to balance estimated emissions of iodine from terrestrial soil and vegetation (from Sive et al., 2007) against an estimate of iodine deposition flux. There is insufficient detail given of how this deposition flux calculation was done, but it appears to be based on "dust fall". Better explanation is required if this calculation is to be understood. Does "dust" here refer to mineral dust? If so, why should its deposition be specifically associated with the deposition of iodine? How exactly was the calculation done? The value given for the terrestrial emission flux (2.27 ug m-2 d-1) does not seem to agree with the value given by Sive et al. (2.7 ug m-2 d-1). How reliable is the comparison likely to be when the emission flux estimate is derived entirely from observations in North America, where vegetation types and land surfaces are different from the study region here?

Have the authors considered the influence of seasonal changes in boundary layer height on aerosol iodine concentrations? These could potentially be significant, and could cause changes in surface level concentrations even when emission fluxes are constant.

While I understand that the authors' estimate of the potential contribution of coal combustion to aerosol iodine loading at their study site is only intended as a first-order estimate, I do not think that they have sufficient information to attempt it. The assumption that surface iodine emissions are mixed through the entire troposphere (i.e. to 10 km) is certainly not realistic, since only a small proportion of emissions are likely to

leave the boundary layer ($\sim$1 km). This implies an order of magnitude greater iodine concentration derived from coal, which does not appear to be plausible.

Lines 221 – 222: "Two severe dust storm events occured in Xi'an in 17-18 April and 4-6 May, 2017, as shown by the peaks of air quality index (AQI) of 268 and 306, respectively (Fig. 2e)." There is only one peak in AQI visible in Fig 2e in this time period. Please explain or amend.

Please give further explanation of the significance of the "low-altitude air mass" mentioned on line 232.

On lines 250 – 254 (and later in the manuscript) the authors discuss the possibility that the aerosol iodine they observed might have formed through primary nucleation. While there are relationships between iodine concentration and those of other species associated with nucleation (e.g. $SO_2$, Fig S6), it is also apparent that the concentration of $SO_2$ is three orders of magnitude greater than that of aerosol iodine. There is no evidence available in this dataset that would make it possible to determine whether iodine is incorporated into aerosol in Xi'an via primary formation or secondary uptake onto existing particles. I therefore suggest that discussion of the iodine aerosol formation mechanism can only be speculation, and it would be better to remove it entirely.

In section 4.2.3 the authors make a convincing case for the influence of interactions with the East Asian Monsoon on long-range 129-I transport to the study site. I am not familiar with the EAWM index mentioned on line 303, but I wonder whether it is possible to make more use of this when exploring the variations in iodine isotope concentrations and their ratios during the study. Can it be plotted on Fig 2? The "z-score" approach discussed on lines 333 – 336 would be more convincing if it could be combined with some quantitative indicator of EAM strength.

On lines 301 – 302 it is stated that "In addition, the 129I level in March 2018 was much less than that in March 2017". This is certainly true, and in fact the 129-I concentration in March 2018 is very similar to that in the two LLP periods. Why did the authors

choose to include these samples in the HLP period?

The statement on line 338 that the iodine isotope ratio shows "relatively weak fluctuation" seems rather subjective, and quite surprising given the relative standard deviation quoted for the parameter of >120%. There are strong variations in the ratio during the HLP 2 period, which do not appear to be consistent with the statement on line 339 about background levels.

Minor comments Line 61: replace "combing" with "combined"? Line 75: replace "ration" with "ratio" Line 178-179: Toyama et al. is cited both at the beginning and end of this sentence, but with different years. Please correct. Line 211: I think the correct units for ozone concentration here should be ppbv, not pptv.

---

## Author Response (AR3)

**Response to Editor**

Comments to the Author:

Many thanks for revising the paper so quickly.

I am happy with the corrections made except formatting of axis labels and the AQI data in Fig. 2e. Please follow the links to SI brochure and IUPAC Green Book given in the manuscript preparation guidelines:

https://www.atmospheric-chemistry-and-physics.net/for_authors/manuscript_preparation.html

Specifically, section 5.4.1 of the SI Brochure and section 1.1 pf the IUPAC Green Book give examples for the correct formating of axis labels. The unit is separated from the quantity symbol by a division symbol, e.g. $N/(10^5 \text{ m}{-3})$. There should be no commas after the quantity symbol or multiplication symbols before the unit.

Table 1 and Fig. S4 still need quantity symbols for the concentrations, e.g. $\gamma(O3)/(\mu g \text{ m-3})$, and other variables.

Forgive me if I am wrong, but the AQI data in Fig. 2e still seem to have much lower time resolution than the data in the figure provided in the response to the reviewers (cyan shaded area).

Dear Editor Dr. Jan Kaiser,

We sincerely thank for your great efforts with the strong responsibility. The recommended files help us a lot on the usage of SI.

According to your comments and the SI related documents, we carefully revised the manuscript and Supplementary Information, mainly including axis labels and table labels in Figs. 2-5 and Table 1 in MS, as well as Figs. 1,2,4 and Table S1 in SI.

As you mentioned, the time resolution of AQI data in Fig. 2e is not so high as that in our reply because we originally would like to show the data on those days when iodine isotopes were measured. Now in this version, we use the daily resolution of AQI data to reproduce Fig. 2e to give a clearer view of air pollution states during our analysis period.

Best regards and happy Chinese New Year!

Luyuan Zhang

Jan 24, 2020

[revised manuscript text omitted]
 $c(^{127}I)/(ng\ m^{-3})$ | | $N(^{129}I)/(10^5\ m^{-3})$ | | Spring (3-5) $c(^{127}I)/(ng\ m^{-3})$ | | $N(^{129}I)/(10^5\ m^{-3})$ | | Summer (6-8) $c(^{127}I)/(ng\ m^{-3})$ | | $N(^{129}I)/(10^5\ m^{-3})$ | | Fall (9-11) $c(^{127}I)/(ng\ m^{-3})$ | | $N(^{129}I)/(10^5\ m^{-3})$ | | Winter (12-2) $c(^{127}I)/(ng\ m^{-3})$ | | $N(^{129}I)/(10^5\ m^{-3})$ | |
|---|---|---|---|---|---|---|---|---|---|---|---|---|---|---|---|---|---|---|---|---|
| | Spear.r | Sig. | Spear.r | Sig. | Spear.r | Sig. | Spear.r | Sig. | Spear.r | Sig. | Spear.r | Sig. | Spear.r | Sig. | Spear.r | Sig. | Spear.r | Sig. | Spear.r | Sig. |
| $N(^{129}I)/(10^5\ m^{-3})$ | **0.33** | **0.01** | | | -0.05 | 0.86 | | | 0.35 | 0.17 | | | 0.04 | 0.86 | | | 0.51 | 0.08 | | |
| $^{129}I/^{127}I$ number ratio/$(10^{10})$ | -**0.28** | **0.02** | **0.74** | **0.00** | -**0.64** | **0.01** | **0.77** | **0.00** | -0.03 | 0.90 | **0.87** | **0.00** | -**0.65** | **0.00** | **0.60** | **0.00** | -**0.94** | **0.00** | -0.35 | 0.25 |
| Temperature/°C | -**0.53** | **0.00** | -**0.47** | **0.00** | -0.09 | 0.75 | 0.21 | 0.44 | -0.15 | 0.55 | 0.37 | 0.14 | -**0.54** | **0.01** | 0.07 | 0.75 | 0.13 | 0.67 | -0.19 | 0.53 |
| Humidity | 0.06 | 0.64 | -0.13 | 0.27 | 0.24 | 0.37 | -**0.53** | **0.04** | -0.06 | 0.82 | 0.29 | 0.26 | -**0.45** | **0.04** | -0.30 | 0.18 | **0.69** | **0.01** | 0.34 | 0.26 |
| Wind speed/(m s⁻¹) | -0.19 | 0.13 | -0.21 | 0.09 | 0.08 | 0.76 | -0.31 | 0.24 | -0.09 | 0.73 | 0.02 | 0.95 | 0.05 | 0.81 | 0.14 | 0.54 | -0.34 | 0.26 | 0.14 | 0.65 |
| Precipitation/mm | -0.14 | 0.25 | -0.18 | 0.15 | -0.04 | 0.88 | -0.13 | 0.64 | 0.04 | 0.87 | 0.42 | 0.09 | -0.29 | 0.19 | -0.39 | 0.07 | 0.15 | 0.61 | 0.00 | 1.00 |
| AQI | **0.72** | **0.00** | 0.17 | 0.17 | **0.95** | **0.00** | -0.05 | 0.85 | **0.59** | **0.01** | -0.01 | 0.97 | **0.56** | **0.01** | 0.13 | 0.55 | **0.87** | **0.00** | 0.45 | 0.12 |
| $c(CO)/(mg\ m^{-3})$ | **0.54** | **0.00** | 0.20 | 0.11 | **0.66** | **0.01** | 0.19 | 0.49 | 0.17 | 0.52 | **0.56** | **0.02** | **0.46** | **0.03** | -0.18 | 0.42 | **0.85** | **0.00** | 0.40 | 0.18 |
| $c(SO_2)/(\mu g\ m^{-3})$ | **0.60** | **0.00** | **0.47** | **0.00** | 0.24 | 0.38 | -0.14 | 0.59 | 0.37 | 0.15 | 0.22 | 0.40 | **0.53** | **0.01** | 0.26 | 0.25 | 0.48 | 0.10 | 0.26 | 0.38 |
| $c(NO_2)/(\mu g\ m^{-3})$ | **0.63** | **0.00** | **0.42** | **0.00** | 0.45 | 0.08 | 0.00 | 0.99 | 0.13 | 0.61 | 0.27 | 0.30 | **0.61** | **0.00** | 0.08 | 0.74 | **0.73** | **0.00** | 0.14 | 0.64 |
| $c(O_3)/(\mu g\ m^{-3})$ | -**0.44** | **0.00** | -**0.33** | **0.01** | -0.32 | 0.23 | 0.09 | 0.75 | 0.39 | 0.12 | -0.37 | 0.15 | -0.20 | 0.37 | 0.28 | 0.21 | -**0.64** | **0.02** | -0.11 | 0.72 |
| $c(PM10)/(\mu g\ m^{-3})$ | **0.71** | **0.00** | **0.24** | **0.05** | **0.84** | **0.00** | 0.00 | 0.99 | **0.74** | **0.00** | 0.21 | 0.42 | **0.51** | **0.02** | 0.14 | 0.54 | **0.84** | **0.00** | 0.50 | 0.08 |
| $c(PM2.5)/(\mu g\ m^{-3})$ | **0.75** | **0.00** | **0.24** | **0.05** | **0.94** | **0.00** | 0.00 | 1.00 | **0.76** | **0.00** | 0.19 | 0.47 | **0.45** | **0.03** | 0.10 | 0.67 | **0.85** | **0.00** | 0.47 | 0.11 |

\* Spearman correlation coefficient is used. 2-tailed test of significance is used. Correlation significant at the 0.05 level is in bold.

**4.2.3 Impact of EAM for long-range transport of [129]I**

Increasing evidence have suggested that the prevailing westerly and EAM system act as crucial driving forces and pathways for transport of the European NFRPs derived [129]I from Europe to East Asia and even to low-latitude southeast Asia (Fan et al., 2016; Zhang et al., 2018a). Monthly variations of atmospheric [129]I in Japan also showed a clear pattern with low [129]I deposition in summer and high in winter, which is also attributed to the impact of EAM (Hasegawa et al., 2017; Kadowaki et al., 2018; Toyama et al., 2013). In this work, seasonal variation of [129]I was identical to the observation in the previous studies (Toyama et al., 2013). However, the day-resolution variation patterns of [129]I and [129]I/[127]I in Xi'an, distinct from monthly variation in Japan, showed three periods with high levels and two periods with low levels, indicating more complex influence of EAM in the typically continental monsoon climate city, Xi'an.

Formatted ... [1]
Formatted ... [2]
Formatted ... [3]
Formatted ... [4]
Formatted ... [5]
Formatted ... [6]
Formatted ... [7]
Formatted ... [8]
Formatted ... [9]
Formatted ... [10]
Formatted ... [11]
Formatted ... [12]
Formatted ... [13]
Formatted ... [14]
Formatted ... [15]
Formatted ... [16]
Formatted ... [17]
Formatted ... [18]
Formatted ... [19]
Formatted ... [20]
Formatted ... [21]
Formatted ... [23]
Formatted ... [24]
Formatted ... [25]
Formatted ... [26]
Formatted ... [27]
Formatted ... [28]
Formatted ... [29]
Formatted ... [30]
Formatted ... [31]
Formatted ... [32]
Formatted ... [33]
Formatted ... [34]
Formatted ... [35]
Formatted ... [36]
Formatted ... [37]
Formatted ... [38]
Formatted ... [39]
Formatted ... [40]
Formatted ... [41]
Formatted ... [42]
Formatted ... [22]
Formatted ... [43]
Formatted ... [44]
Formatted ... [45]

[revised manuscript text omitted]

Not Superscript/ Subscript

| Page 15: [1] Formatted | Luyuan Zhang | 1/23/20 11:49:00 PM |

Not Superscript/ Subscript

| Page 15: [1] Formatted | Luyuan Zhang | 1/23/20 11:49:00 PM |

Not Superscript/ Subscript

| Page 15: [1] Formatted | Luyuan Zhang | 1/23/20 11:49:00 PM |

Not Superscript/ Subscript

| Page 15: [2] Formatted | Luyuan Zhang | 1/23/20 11:49:00 PM |

Formatted

| Page 15: [2] Formatted | Luyuan Zhang | 1/23/20 11:49:00 PM |

Formatted

| Page 15: [2] Formatted | Luyuan Zhang | 1/23/20 11:49:00 PM |

Formatted

| Page 15: [2] Formatted | Luyuan Zhang | 1/23/20 11:49:00 PM |

Formatted

| Page 15: [3] Formatted | Luyuan Zhang | 1/23/20 11:49:00 PM |

Formatted

| Page 15: [4] Formatted | Luyuan Zhang | 1/23/20 11:49:00 PM |

Formatted

| Page 15: [4] Formatted | Luyuan Zhang | 1/23/20 11:49:00 PM |

Formatted

| Page 15: [5] Formatted | Luyuan Zhang | 1/23/20 11:49:00 PM |

Formatted

| Page 15: [6] Formatted | Luyuan Zhang | 1/23/20 11:49:00 PM |

Formatted

| Page 15: [6] Formatted | Luyuan Zhang | 1/23/20 11:49:00 PM |

Formatted

| Page 15: [7] Formatted | Luyuan Zhang | 1/23/20 11:49:00 PM |

Formatted

| Page 15: [8] Formatted | Luyuan Zhang | 1/23/20 11:49:00 PM |

| Page 15: [8] Formatted | Luyuan Zhang | 1/23/20 11:49:00 PM |
|---|---|---|

Formatted

| Page 15: [9] Formatted | Luyuan Zhang | 1/23/20 11:49:00 PM |
|---|---|---|

Formatted

| Page 15: [10] Formatted | Luyuan Zhang | 1/23/20 11:49:00 PM |
|---|---|---|

Formatted

| Page 15: [10] Formatted | Luyuan Zhang | 1/23/20 11:49:00 PM |
|---|---|---|

Formatted

| Page 15: [11] Formatted | Luyuan Zhang | 1/23/20 11:49:00 PM |
|---|---|---|

Formatted

| Page 15: [11] Formatted | Luyuan Zhang | 1/23/20 11:49:00 PM |
|---|---|---|

Formatted

| Page 15: [12] Formatted | Luyuan Zhang | 1/23/20 11:49:00 PM |
|---|---|---|

Font: (Default) +Body (Times New Roman)

| Page 15: [13] Formatted | Luyuan Zhang | 1/23/20 11:49:00 PM |
|---|---|---|

Font: (Default) +Body (Times New Roman)

| Page 15: [14] Formatted | Luyuan Zhang | 1/23/20 11:49:00 PM |
|---|---|---|

Font: (Default) +Body (Times New Roman)

| Page 15: [15] Formatted | Luyuan Zhang | 1/23/20 11:49:00 PM |
|---|---|---|

Font: (Default) +Body (Times New Roman)

| Page 15: [16] Formatted | Luyuan Zhang | 1/23/20 11:49:00 PM |
|---|---|---|

Font: (Default) +Body (Times New Roman)

| Page 15: [17] Formatted | Luyuan Zhang | 1/23/20 11:49:00 PM |
|---|---|---|

Font: (Default) +Body (Times New Roman)

| Page 15: [18] Formatted | Luyuan Zhang | 1/23/20 11:49:00 PM |
|---|---|---|

Font: (Default) +Body (Times New Roman)

| Page 15: [19] Formatted | Luyuan Zhang | 1/23/20 11:49:00 PM |
|---|---|---|

Font: (Default) +Body (Times New Roman)

| Page 15: [20] Formatted | Luyuan Zhang | 1/23/20 11:49:00 PM |
|---|---|---|

Font: (Default) +Body (Times New Roman)

| Page 15: [21] Formatted | Luyuan Zhang | 1/23/20 11:49:00 PM |
|---|---|---|

| Page 15: [22] Formatted | Luyuan Zhang | 1/23/20 11:49:00 PM |
|---|---|---|

Formatted

| Page 15: [22] Formatted | Luyuan Zhang | 1/23/20 11:49:00 PM |
|---|---|---|

Formatted

| Page 15: [23] Formatted | Luyuan Zhang | 1/23/20 11:49:00 PM |
|---|---|---|

Font: (Default) +Body (Times New Roman)

| Page 15: [24] Formatted | Luyuan Zhang | 1/23/20 11:49:00 PM |
|---|---|---|

Font: (Default) +Body (Times New Roman)

| Page 15: [25] Formatted | Luyuan Zhang | 1/23/20 11:49:00 PM |
|---|---|---|

Font: (Default) +Body (Times New Roman)

| Page 15: [26] Formatted | Luyuan Zhang | 1/23/20 11:49:00 PM |
|---|---|---|

Font: (Default) +Body (Times New Roman)

| Page 15: [27] Formatted | Luyuan Zhang | 1/23/20 11:49:00 PM |
|---|---|---|

Font: (Default) +Body (Times New Roman)

| Page 15: [28] Formatted | Luyuan Zhang | 1/23/20 11:49:00 PM |
|---|---|---|

Font: (Default) +Body (Times New Roman)

| Page 15: [29] Formatted | Luyuan Zhang | 1/23/20 11:49:00 PM |
|---|---|---|

Font: (Default) +Body (Times New Roman)

| Page 15: [30] Formatted | Luyuan Zhang | 1/23/20 11:49:00 PM |
|---|---|---|

Font: (Default) +Body (Times New Roman)

| Page 15: [31] Formatted | Luyuan Zhang | 1/23/20 11:49:00 PM |
|---|---|---|

Font: (Default) +Body (Times New Roman)

| Page 15: [32] Formatted | Luyuan Zhang | 1/23/20 11:49:00 PM |
|---|---|---|

Font: (Default) +Body (Times New Roman)

| Page 15: [33] Formatted | Luyuan Zhang | 1/23/20 11:49:00 PM |
|---|---|---|

Font: (Default) +Body (Times New Roman)

| Page 15: [34] Formatted | Luyuan Zhang | 1/23/20 11:49:00 PM |
|---|---|---|

Font: (Default) +Body (Times New Roman)

| Page 15: [35] Formatted | Luyuan Zhang | 1/23/20 11:49:00 PM |
|---|---|---|

Font: (Default) +Body (Times New Roman)

| Page 15: [36] Formatted | Luyuan Zhang | 1/23/20 11:49:00 PM |
|---|---|---|

| Page 15: [37] Formatted | Luyuan Zhang | 1/23/20 11:49:00 PM |
|---|---|---|

Font: (Default) +Body (Times New Roman)

| Page 15: [38] Formatted | Luyuan Zhang | 1/23/20 11:49:00 PM |
|---|---|---|

Font: (Default) +Body (Times New Roman)

| Page 15: [39] Formatted | Luyuan Zhang | 1/23/20 11:49:00 PM |
|---|---|---|

Font: (Default) +Body (Times New Roman)

| Page 15: [40] Formatted | Luyuan Zhang | 1/23/20 11:49:00 PM |
|---|---|---|

Font: (Default) +Body (Times New Roman)

| Page 15: [41] Formatted | Luyuan Zhang | 1/23/20 11:49:00 PM |
|---|---|---|

Font: (Default) +Body (Times New Roman)

| Page 15: [42] Formatted | Luyuan Zhang | 1/23/20 11:49:00 PM |
|---|---|---|

Font: (Default) +Body (Times New Roman)

| Page 15: [43] Formatted | Luyuan Zhang | 1/23/20 11:49:00 PM |
|---|---|---|

Formatted

| Page 15: [43] Formatted | Luyuan Zhang | 1/23/20 11:49:00 PM |
|---|---|---|

Formatted

| Page 15: [44] Formatted | Luyuan Zhang | 1/23/20 11:49:00 PM |
|---|---|---|

Font: (Default) +Body (Times New Roman)

| Page 15: [45] Formatted | Luyuan Zhang | 1/23/20 11:49:00 PM |
|---|---|---|

Font: (Default) +Body (Times New Roman)

| Page 15: [46] Formatted | Luyuan Zhang | 1/23/20 11:49:00 PM |
|---|---|---|

Font: (Default) +Body (Times New Roman)

| Page 15: [47] Formatted | Luyuan Zhang | 1/23/20 11:49:00 PM |
|---|---|---|

Font: (Default) +Body (Times New Roman)

| Page 15: [48] Formatted | Luyuan Zhang | 1/23/20 11:49:00 PM |
|---|---|---|

Font: (Default) +Body (Times New Roman)

| Page 15: [49] Formatted | Luyuan Zhang | 1/23/20 11:49:00 PM |
|---|---|---|

Font: (Default) +Body (Times New Roman)

| Page 15: [50] Formatted | Luyuan Zhang | 1/23/20 11:49:00 PM |
|---|---|---|

Font: (Default) +Body (Times New Roman)

| Page 15: [51] Formatted | Luyuan Zhang | 1/23/20 11:49:00 PM |
|---|---|---|

**Page 15: [52] Formatted**      **Luyuan Zhang**      **1/23/20 11:49:00 PM**

Font: (Default) +Body (Times New Roman)

**Page 15: [53] Formatted**      **Luyuan Zhang**      **1/23/20 11:49:00 PM**

Font: (Default) +Body (Times New Roman)

**Page 15: [54] Formatted**      **Luyuan Zhang**      **1/23/20 11:49:00 PM**

Font: (Default) +Body (Times New Roman)

**Page 15: [55] Formatted**      **Luyuan Zhang**      **1/23/20 11:49:00 PM**

Font: (Default) +Body (Times New Roman)

**Page 15: [56] Formatted**      **Luyuan Zhang**      **1/23/20 11:49:00 PM**

Font: (Default) +Body (Times New Roman)

**Page 15: [57] Formatted**      **Luyuan Zhang**      **1/23/20 11:49:00 PM**

Font: (Default) +Body (Times New Roman)

**Page 15: [58] Formatted**      **Luyuan Zhang**      **1/23/20 11:49:00 PM**

Font: (Default) +Body (Times New Roman)

**Page 15: [59] Formatted**      **Luyuan Zhang**      **1/23/20 11:49:00 PM**

Font: (Default) +Body (Times New Roman)

**Page 15: [60] Formatted**      **Luyuan Zhang**      **1/23/20 11:49:00 PM**

Font: (Default) +Body (Times New Roman)

**Page 15: [61] Formatted**      **Luyuan Zhang**      **1/23/20 11:49:00 PM**

Font: (Default) +Body (Times New Roman)

**Page 15: [62] Formatted**      **Luyuan Zhang**      **1/23/20 11:49:00 PM**

Font: (Default) +Body (Times New Roman)

**Page 15: [63] Formatted**      **Luyuan Zhang**      **1/23/20 11:49:00 PM**

Font: (Default) +Body (Times New Roman)

**Page 15: [64] Formatted**      **Luyuan Zhang**      **1/23/20 11:49:00 PM**

Formatted

**Page 15: [65] Formatted**      **Luyuan Zhang**      **1/23/20 11:49:00 PM**

Font: (Default) +Body (Times New Roman)

**Page 15: [66] Formatted**      **Luyuan Zhang**      **1/23/20 11:49:00 PM**

Font: (Default) +Body (Times New Roman)

**Page 15: [67] Formatted**      **Luyuan Zhang**      **1/23/20 11:49:00 PM**

| Page 15: [68] Formatted | Luyuan Zhang | 1/23/20 11:49:00 PM |
|---|---|---|

Font: (Default) +Body (Times New Roman)

| Page 15: [69] Formatted | Luyuan Zhang | 1/23/20 11:49:00 PM |
|---|---|---|

Font: (Default) +Body (Times New Roman)

| Page 15: [70] Formatted | Luyuan Zhang | 1/23/20 11:49:00 PM |
|---|---|---|

Font: (Default) +Body (Times New Roman)

| Page 15: [71] Formatted | Luyuan Zhang | 1/23/20 11:49:00 PM |
|---|---|---|

Font: (Default) +Body (Times New Roman)

| Page 15: [72] Formatted | Luyuan Zhang | 1/23/20 11:49:00 PM |
|---|---|---|

Font: (Default) +Body (Times New Roman)

| Page 15: [73] Formatted | Luyuan Zhang | 1/23/20 11:49:00 PM |
|---|---|---|

Font: (Default) +Body (Times New Roman)

| Page 15: [74] Formatted | Luyuan Zhang | 1/23/20 11:49:00 PM |
|---|---|---|

Font: (Default) +Body (Times New Roman)

| Page 15: [75] Formatted | Luyuan Zhang | 1/23/20 11:49:00 PM |
|---|---|---|

Font: (Default) +Body (Times New Roman)

| Page 15: [76] Formatted | Luyuan Zhang | 1/23/20 11:49:00 PM |
|---|---|---|

Font: (Default) +Body (Times New Roman)

| Page 15: [77] Formatted | Luyuan Zhang | 1/23/20 11:49:00 PM |
|---|---|---|

Font: (Default) +Body (Times New Roman)

| Page 15: [78] Formatted | Luyuan Zhang | 1/23/20 11:49:00 PM |
|---|---|---|

Font: (Default) +Body (Times New Roman)

| Page 15: [79] Formatted | Luyuan Zhang | 1/23/20 11:49:00 PM |
|---|---|---|

Font: (Default) +Body (Times New Roman)

| Page 15: [80] Formatted | Luyuan Zhang | 1/23/20 11:49:00 PM |
|---|---|---|

Font: (Default) +Body (Times New Roman)

| Page 15: [81] Formatted | Luyuan Zhang | 1/23/20 11:49:00 PM |
|---|---|---|

Font: (Default) +Body (Times New Roman)

| Page 15: [82] Formatted | Luyuan Zhang | 1/23/20 11:49:00 PM |
|---|---|---|

Font: (Default) +Body (Times New Roman)

| Page 15: [83] Formatted | Luyuan Zhang | 1/23/20 11:49:00 PM |
|---|---|---|

**Page 15: [84] Formatted**      **Luyuan Zhang**      **1/23/20 11:49:00 PM**

Font: (Default) +Body (Times New Roman)

**Page 15: [85] Formatted**      **Luyuan Zhang**      **1/23/20 11:49:00 PM**

Superscript

**Page 15: [86] Formatted**      **Luyuan Zhang**      **1/23/20 11:49:00 PM**

Font: (Default) +Body (Times New Roman)

**Page 15: [87] Formatted**      **Luyuan Zhang**      **1/23/20 11:49:00 PM**

Font: (Default) +Body (Times New Roman)

**Page 15: [88] Formatted**      **Luyuan Zhang**      **1/23/20 11:49:00 PM**

Font: (Default) +Body (Times New Roman)

**Page 15: [89] Formatted**      **Luyuan Zhang**      **1/23/20 11:49:00 PM**

Font: (Default) +Body (Times New Roman)

**Page 15: [90] Formatted**      **Luyuan Zhang**      **1/23/20 11:49:00 PM**

Font: (Default) +Body (Times New Roman)

**Page 15: [91] Formatted**      **Luyuan Zhang**      **1/23/20 11:49:00 PM**

Font: (Default) +Body (Times New Roman)

**Page 15: [92] Formatted**      **Luyuan Zhang**      **1/23/20 11:49:00 PM**

Font: (Default) +Body (Times New Roman)

**Page 15: [93] Formatted**      **Luyuan Zhang**      **1/23/20 11:49:00 PM**

Font: (Default) +Body (Times New Roman)

**Page 15: [94] Formatted**      **Luyuan Zhang**      **1/23/20 11:49:00 PM**

Font: (Default) +Body (Times New Roman)

**Page 15: [95] Formatted**      **Luyuan Zhang**      **1/23/20 11:49:00 PM**

Font: (Default) +Body (Times New Roman)

**Page 15: [96] Formatted**      **Luyuan Zhang**      **1/23/20 11:49:00 PM**

Font: (Default) +Body (Times New Roman)

**Page 15: [97] Formatted**      **Luyuan Zhang**      **1/23/20 11:49:00 PM**

Font: (Default) +Body (Times New Roman)

**Page 15: [98] Formatted**      **Luyuan Zhang**      **1/23/20 11:49:00 PM**

Font: (Default) +Body (Times New Roman)

**Page 15: [99] Formatted**      **Luyuan Zhang**      **1/23/20 11:49:00 PM**

**Page 15: [100] Formatted**        Luyuan Zhang        1/23/20 11:49:00 PM

Font: (Default) +Body (Times New Roman)

**Page 15: [101] Formatted**        Luyuan Zhang        1/23/20 11:49:00 PM

Font: (Default) +Body (Times New Roman)

**Page 15: [102] Formatted**        Luyuan Zhang        1/23/20 11:49:00 PM

Font: (Default) +Body (Times New Roman)

**Page 15: [103] Formatted**        Luyuan Zhang        1/23/20 11:49:00 PM

Font: (Default) +Body (Times New Roman)

**Page 15: [104] Formatted**        Luyuan Zhang        1/23/20 11:49:00 PM

Font: (Default) +Body (Times New Roman)

**Page 15: [105] Formatted**        Luyuan Zhang        1/23/20 11:49:00 PM

Font: (Default) +Body (Times New Roman)

**Page 15: [106] Formatted**        Luyuan Zhang        1/23/20 11:49:00 PM

Font: (Default) +Body (Times New Roman)

**Page 15: [107] Formatted**        Luyuan Zhang        1/23/20 11:49:00 PM

Font: (Default) +Body (Times New Roman)

**Page 15: [108] Formatted**        Luyuan Zhang        1/23/20 11:49:00 PM

Font: (Default) +Body (Times New Roman)

**Page 15: [109] Formatted**        Luyuan Zhang        1/23/20 11:49:00 PM

Font: (Default) +Body (Times New Roman)

**Page 15: [110] Formatted**        Luyuan Zhang        1/23/20 11:49:00 PM

Font: (Default) +Body (Times New Roman)

**Page 15: [111] Formatted**        Luyuan Zhang        1/23/20 11:49:00 PM

Font: (Default) +Body (Times New Roman)

**Page 15: [112] Formatted**        Luyuan Zhang        1/23/20 11:49:00 PM

Font: (Default) +Body (Times New Roman)

**Page 15: [113] Formatted**        Luyuan Zhang        1/23/20 11:49:00 PM

Font: (Default) +Body (Times New Roman)

**Page 15: [114] Formatted**        Luyuan Zhang        1/23/20 11:49:00 PM

Font: (Default) +Body (Times New Roman)

**Page 15: [115] Formatted**        Luyuan Zhang        1/23/20 11:49:00 PM

**Page 15: [116] Formatted** Luyuan Zhang 1/23/20 11:49:00 PM

Font: (Default) +Body (Times New Roman)

**Page 15: [117] Formatted** Luyuan Zhang 1/23/20 11:49:00 PM

Font: (Default) +Body (Times New Roman)

**Page 15: [118] Formatted** Luyuan Zhang 1/23/20 11:49:00 PM

Font: (Default) +Body (Times New Roman)

**Page 15: [119] Formatted** Luyuan Zhang 1/23/20 11:49:00 PM

Font: (Default) +Body (Times New Roman)

**Page 15: [120] Formatted** Luyuan Zhang 1/23/20 11:49:00 PM

Font: (Default) +Body (Times New Roman)

**Page 15: [121] Formatted** Luyuan Zhang 1/23/20 11:49:00 PM

Font: (Default) +Body (Times New Roman)

**Page 15: [122] Formatted** Luyuan Zhang 1/23/20 11:49:00 PM

Font: (Default) +Body (Times New Roman)

**Page 15: [123] Formatted** Luyuan Zhang 1/23/20 11:49:00 PM

Font: (Default) +Body (Times New Roman)

**Page 15: [124] Formatted** Luyuan Zhang 1/23/20 11:49:00 PM

Font: (Default) +Body (Times New Roman)

**Page 15: [125] Formatted** Luyuan Zhang 1/23/20 11:49:00 PM

Font: (Default) +Body (Times New Roman)

**Page 15: [126] Formatted** Luyuan Zhang 1/23/20 11:49:00 PM

Font: (Default) +Body (Times New Roman)

**Page 15: [127] Formatted** Luyuan Zhang 1/23/20 11:49:00 PM

Font: (Default) +Body (Times New Roman)

**Page 15: [128] Formatted** Luyuan Zhang 1/23/20 11:49:00 PM

Font: (Default) +Body (Times New Roman)

**Page 15: [129] Formatted** Luyuan Zhang 1/23/20 11:49:00 PM

Font: (Default) +Body (Times New Roman)

**Page 15: [130] Formatted** Luyuan Zhang 1/23/20 11:49:00 PM

Font: (Default) +Body (Times New Roman)

**Page 15: [131] Formatted** Luyuan Zhang 1/23/20 11:49:00 PM

**Page 15: [132] Formatted**      **Luyuan Zhang**      **1/23/20 11:49:00 PM**

Font: (Default) +Body (Times New Roman)

**Page 15: [133] Formatted**      **Luyuan Zhang**      **1/23/20 11:49:00 PM**

Font: (Default) +Body (Times New Roman)

**Page 15: [134] Formatted**      **Luyuan Zhang**      **1/23/20 11:49:00 PM**

Font: (Default) +Body (Times New Roman)

**Page 15: [135] Formatted**      **Luyuan Zhang**      **1/23/20 11:49:00 PM**

Font: (Default) +Body (Times New Roman)

**Page 15: [136] Formatted**      **Luyuan Zhang**      **1/23/20 11:49:00 PM**

Font: (Default) +Body (Times New Roman)

**Page 15: [137] Formatted**      **Luyuan Zhang**      **1/23/20 11:49:00 PM**

Font: (Default) +Body (Times New Roman)

**Page 15: [138] Formatted**      **Luyuan Zhang**      **1/23/20 11:49:00 PM**

Font: (Default) +Body (Times New Roman)

**Page 15: [139] Formatted**      **Luyuan Zhang**      **1/23/20 11:49:00 PM**

Font: (Default) +Body (Times New Roman)

**Page 15: [140] Formatted**      **Luyuan Zhang**      **1/23/20 11:49:00 PM**

Font: (Default) +Body (Times New Roman)

**Page 15: [141] Formatted**      **Luyuan Zhang**      **1/23/20 11:49:00 PM**

Font: (Default) +Body (Times New Roman)

**Page 15: [142] Formatted**      **Luyuan Zhang**      **1/23/20 11:49:00 PM**

Font: (Default) +Body (Times New Roman)

**Page 15: [143] Formatted**      **Luyuan Zhang**      **1/23/20 11:49:00 PM**

Font: (Default) +Body (Times New Roman)

**Page 15: [144] Formatted**      **Luyuan Zhang**      **1/23/20 11:49:00 PM**

Font: (Default) +Body (Times New Roman)

**Page 15: [145] Formatted**      **Luyuan Zhang**      **1/23/20 11:49:00 PM**

Font: (Default) +Body (Times New Roman)

**Page 15: [146] Formatted**      **Luyuan Zhang**      **1/23/20 11:49:00 PM**

Formatted

**Page 15: [147] Formatted**      **Luyuan Zhang**      **1/23/20 11:49:00 PM**

**Page 15: [148] Formatted**        **Luyuan Zhang**        **1/23/20 11:49:00 PM**

Font: (Default) +Body (Times New Roman)

**Page 15: [149] Formatted**        **Luyuan Zhang**        **1/23/20 11:49:00 PM**

Font: (Default) +Body (Times New Roman)

**Page 15: [150] Formatted**        **Luyuan Zhang**        **1/23/20 11:49:00 PM**

Font: (Default) +Body (Times New Roman)

**Page 15: [151] Formatted**        **Luyuan Zhang**        **1/23/20 11:49:00 PM**

Font: (Default) +Body (Times New Roman)

**Page 15: [152] Formatted**        **Luyuan Zhang**        **1/23/20 11:49:00 PM**

Font: (Default) +Body (Times New Roman)

**Page 15: [153] Formatted**        **Luyuan Zhang**        **1/23/20 11:49:00 PM**

Font: (Default) +Body (Times New Roman)

**Page 15: [154] Formatted**        **Luyuan Zhang**        **1/23/20 11:49:00 PM**

Font: (Default) +Body (Times New Roman)

**Page 15: [155] Formatted**        **Luyuan Zhang**        **1/23/20 11:49:00 PM**

Font: (Default) +Body (Times New Roman)

**Page 15: [156] Formatted**        **Luyuan Zhang**        **1/23/20 11:49:00 PM**

Font: (Default) +Body (Times New Roman)

**Page 15: [157] Formatted**        **Luyuan Zhang**        **1/23/20 11:49:00 PM**

Font: (Default) +Body (Times New Roman)

**Page 15: [158] Formatted**        **Luyuan Zhang**        **1/23/20 11:49:00 PM**

Font: (Default) +Body (Times New Roman)

**Page 15: [159] Formatted**        **Luyuan Zhang**        **1/23/20 11:49:00 PM**

Font: (Default) +Body (Times New Roman)

**Page 15: [160] Formatted**        **Luyuan Zhang**        **1/23/20 11:49:00 PM**

Font: (Default) +Body (Times New Roman)

**Page 15: [161] Formatted**        **Luyuan Zhang**        **1/23/20 11:49:00 PM**

Font: (Default) +Body (Times New Roman)

**Page 15: [162] Formatted**        **Luyuan Zhang**        **1/23/20 11:49:00 PM**

Font: (Default) +Body (Times New Roman)

**Page 15: [163] Formatted**        **Luyuan Zhang**        **1/23/20 11:49:00 PM**

**Page 15: [164] Formatted**        **Luyuan Zhang**        **1/23/20 11:49:00 PM**

Font: (Default) +Body (Times New Roman)

**Page 15: [165] Formatted**        **Luyuan Zhang**        **1/23/20 11:49:00 PM**

Font: (Default) +Body (Times New Roman)

**Page 15: [166] Formatted**        **Luyuan Zhang**        **1/23/20 11:49:00 PM**

Font: (Default) +Body (Times New Roman)

**Page 15: [167] Formatted**        **Luyuan Zhang**        **1/23/20 11:49:00 PM**

Formatted

**Page 15: [167] Formatted**        **Luyuan Zhang**        **1/23/20 11:49:00 PM**

Formatted

**Page 15: [168] Formatted**        **Luyuan Zhang**        **1/23/20 11:49:00 PM**

Font: (Default) +Body (Times New Roman)

**Page 15: [169] Formatted**        **Luyuan Zhang**        **1/23/20 11:49:00 PM**

Font: (Default) +Body (Times New Roman)

**Page 15: [170] Formatted**        **Luyuan Zhang**        **1/23/20 11:49:00 PM**

Font: (Default) +Body (Times New Roman)

**Page 15: [171] Formatted**        **Luyuan Zhang**        **1/23/20 11:49:00 PM**

Font: (Default) +Body (Times New Roman)

**Page 15: [172] Formatted**        **Luyuan Zhang**        **1/23/20 11:49:00 PM**

Font: (Default) +Body (Times New Roman)

**Page 15: [173] Formatted**        **Luyuan Zhang**        **1/23/20 11:49:00 PM**

Font: (Default) +Body (Times New Roman)

**Page 15: [174] Formatted**        **Luyuan Zhang**        **1/23/20 11:49:00 PM**

Font: (Default) +Body (Times New Roman)

**Page 15: [175] Formatted**        **Luyuan Zhang**        **1/23/20 11:49:00 PM**

Font: (Default) +Body (Times New Roman)

**Page 15: [176] Formatted**        **Luyuan Zhang**        **1/23/20 11:49:00 PM**

Font: (Default) +Body (Times New Roman)

**Page 15: [177] Formatted**        **Luyuan Zhang**        **1/23/20 11:49:00 PM**

Font: (Default) +Body (Times New Roman)

**Page 15: [178] Formatted**        **Luyuan Zhang**        **1/23/20 11:49:00 PM**

**Page 15: [179] Formatted** | **Luyuan Zhang** | **1/23/20 11:49:00 PM**

Font: (Default) +Body (Times New Roman)

**Page 15: [180] Formatted** | **Luyuan Zhang** | **1/23/20 11:49:00 PM**

Font: (Default) +Body (Times New Roman)

**Page 15: [181] Formatted** | **Luyuan Zhang** | **1/23/20 11:49:00 PM**

Font: (Default) +Body (Times New Roman)

**Page 15: [182] Formatted** | **Luyuan Zhang** | **1/23/20 11:49:00 PM**

Font: (Default) +Body (Times New Roman)

**Page 15: [183] Formatted** | **Luyuan Zhang** | **1/23/20 11:49:00 PM**

Font: (Default) +Body (Times New Roman)

**Page 15: [184] Formatted** | **Luyuan Zhang** | **1/23/20 11:49:00 PM**

Font: (Default) +Body (Times New Roman)

**Page 15: [185] Formatted** | **Luyuan Zhang** | **1/23/20 11:49:00 PM**

Font: (Default) +Body (Times New Roman)

**Page 15: [186] Formatted** | **Luyuan Zhang** | **1/23/20 11:49:00 PM**

Font: (Default) +Body (Times New Roman)

**Page 15: [187] Formatted** | **Luyuan Zhang** | **1/23/20 11:49:00 PM**

Font: (Default) +Body (Times New Roman)

**Page 15: [188] Formatted** | **Luyuan Zhang** | **1/23/20 11:49:00 PM**

Formatted

**Page 15: [188] Formatted** | **Luyuan Zhang** | **1/23/20 11:49:00 PM**

Formatted

**Page 15: [189] Formatted** | **Luyuan Zhang** | **1/23/20 11:49:00 PM**

Font: (Default) +Body (Times New Roman)

**Page 15: [190] Formatted** | **Luyuan Zhang** | **1/23/20 11:49:00 PM**

Font: (Default) +Body (Times New Roman)

**Page 15: [191] Formatted** | **Luyuan Zhang** | **1/23/20 11:49:00 PM**

Font: (Default) +Body (Times New Roman)

**Page 15: [192] Formatted** | **Luyuan Zhang** | **1/23/20 11:49:00 PM**

Font: (Default) +Body (Times New Roman)

**Page 15: [193] Formatted** | **Luyuan Zhang** | **1/23/20 11:49:00 PM**

**Page 15: [194] Formatted** | **Luyuan Zhang** | **1/23/20 11:49:00 PM**

Font: (Default) +Body (Times New Roman)

**Page 15: [195] Formatted** | **Luyuan Zhang** | **1/23/20 11:49:00 PM**

Font: (Default) +Body (Times New Roman)

**Page 15: [196] Formatted** | **Luyuan Zhang** | **1/23/20 11:49:00 PM**

Font: (Default) +Body (Times New Roman)

**Page 15: [197] Formatted** | **Luyuan Zhang** | **1/23/20 11:49:00 PM**

Font: (Default) +Body (Times New Roman)

**Page 15: [198] Formatted** | **Luyuan Zhang** | **1/23/20 11:49:00 PM**

Font: (Default) +Body (Times New Roman)

**Page 15: [199] Formatted** | **Luyuan Zhang** | **1/23/20 11:49:00 PM**

Font: (Default) +Body (Times New Roman)

**Page 15: [200] Formatted** | **Luyuan Zhang** | **1/23/20 11:49:00 PM**

Font: (Default) +Body (Times New Roman)

**Page 15: [201] Formatted** | **Luyuan Zhang** | **1/23/20 11:49:00 PM**

Font: (Default) +Body (Times New Roman)

**Page 15: [202] Formatted** | **Luyuan Zhang** | **1/23/20 11:49:00 PM**

Font: (Default) +Body (Times New Roman)

**Page 15: [203] Formatted** | **Luyuan Zhang** | **1/23/20 11:49:00 PM**

Font: (Default) +Body (Times New Roman)

**Page 15: [204] Formatted** | **Luyuan Zhang** | **1/23/20 11:49:00 PM**

Font: (Default) +Body (Times New Roman)

**Page 15: [205] Formatted** | **Luyuan Zhang** | **1/23/20 11:49:00 PM**

Font: (Default) +Body (Times New Roman)

**Page 15: [206] Formatted** | **Luyuan Zhang** | **1/23/20 11:49:00 PM**

Font: (Default) +Body (Times New Roman)

**Page 15: [207] Formatted** | **Luyuan Zhang** | **1/23/20 11:49:00 PM**

Font: (Default) +Body (Times New Roman)

**Page 15: [208] Formatted** | **Luyuan Zhang** | **1/23/20 11:49:00 PM**

Font: (Default) +Body (Times New Roman)

**Page 15: [209] Formatted** | **Luyuan Zhang** | **1/23/20 11:49:00 PM**

| Page 15: [209] Formatted | Luyuan Zhang | 1/23/20 11:49:00 PM |
|---|---|---|

Formatted

| Page 15: [210] Formatted | Luyuan Zhang | 1/23/20 11:49:00 PM |
|---|---|---|

Font: (Default) +Body (Times New Roman)

| Page 15: [211] Formatted | Luyuan Zhang | 1/23/20 11:49:00 PM |
|---|---|---|

Font: (Default) +Body (Times New Roman)

| Page 15: [212] Formatted | Luyuan Zhang | 1/23/20 11:49:00 PM |
|---|---|---|

Font: (Default) +Body (Times New Roman)

| Page 15: [213] Formatted | Luyuan Zhang | 1/23/20 11:49:00 PM |
|---|---|---|

Font: (Default) +Body (Times New Roman)

| Page 15: [214] Formatted | Luyuan Zhang | 1/23/20 11:49:00 PM |
|---|---|---|

Font: (Default) +Body (Times New Roman)

| Page 15: [215] Formatted | Luyuan Zhang | 1/23/20 11:49:00 PM |
|---|---|---|

Font: (Default) +Body (Times New Roman)

| Page 15: [216] Formatted | Luyuan Zhang | 1/23/20 11:49:00 PM |
|---|---|---|

Font: (Default) +Body (Times New Roman)

| Page 15: [217] Formatted | Luyuan Zhang | 1/23/20 11:49:00 PM |
|---|---|---|

Font: (Default) +Body (Times New Roman)

| Page 15: [218] Formatted | Luyuan Zhang | 1/23/20 11:49:00 PM |
|---|---|---|

Font: (Default) +Body (Times New Roman)

| Page 15: [219] Formatted | Luyuan Zhang | 1/23/20 11:49:00 PM |
|---|---|---|

Font: (Default) +Body (Times New Roman)

| Page 15: [220] Formatted | Luyuan Zhang | 1/23/20 11:49:00 PM |
|---|---|---|

Font: (Default) +Body (Times New Roman)

| Page 15: [221] Formatted | Luyuan Zhang | 1/23/20 11:49:00 PM |
|---|---|---|

Font: (Default) +Body (Times New Roman)

| Page 15: [222] Formatted | Luyuan Zhang | 1/23/20 11:49:00 PM |
|---|---|---|

Font: (Default) +Body (Times New Roman)

| Page 15: [223] Formatted | Luyuan Zhang | 1/23/20 11:49:00 PM |
|---|---|---|

Font: (Default) +Body (Times New Roman)

| Page 15: [224] Formatted | Luyuan Zhang | 1/23/20 11:49:00 PM |
|---|---|---|

| Page 15: [225] Formatted | Luyuan Zhang | 1/23/20 11:49:00 PM |
|---|---|---|

Font: (Default) +Body (Times New Roman)

| Page 15: [226] Formatted | Luyuan Zhang | 1/23/20 11:49:00 PM |
|---|---|---|

Font: (Default) +Body (Times New Roman)

| Page 15: [227] Formatted | Luyuan Zhang | 1/23/20 11:49:00 PM |
|---|---|---|

Font: (Default) +Body (Times New Roman)

| Page 15: [228] Formatted | Luyuan Zhang | 1/23/20 11:49:00 PM |
|---|---|---|

Font: (Default) +Body (Times New Roman)

| Page 15: [229] Formatted | Luyuan Zhang | 1/23/20 11:49:00 PM |
|---|---|---|

Font: (Default) +Body (Times New Roman)

| Page 15: [230] Formatted | Luyuan Zhang | 1/23/20 11:49:00 PM |
|---|---|---|

Formatted

| Page 15: [230] Formatted | Luyuan Zhang | 1/23/20 11:49:00 PM |
|---|---|---|

Formatted

| Page 15: [231] Formatted | Luyuan Zhang | 1/23/20 11:49:00 PM |
|---|---|---|

Font: (Default) +Body (Times New Roman)

| Page 15: [232] Formatted | Luyuan Zhang | 1/23/20 11:49:00 PM |
|---|---|---|

Font: (Default) +Body (Times New Roman)

| Page 15: [233] Formatted | Luyuan Zhang | 1/23/20 11:49:00 PM |
|---|---|---|

Font: (Default) +Body (Times New Roman)

| Page 15: [234] Formatted | Luyuan Zhang | 1/23/20 11:49:00 PM |
|---|---|---|

Font: (Default) +Body (Times New Roman)

| Page 15: [235] Formatted | Luyuan Zhang | 1/23/20 11:49:00 PM |
|---|---|---|

Font: (Default) +Body (Times New Roman)

| Page 15: [236] Formatted | Luyuan Zhang | 1/23/20 11:49:00 PM |
|---|---|---|

Font: (Default) +Body (Times New Roman)

| Page 15: [237] Formatted | Luyuan Zhang | 1/23/20 11:49:00 PM |
|---|---|---|

Font: (Default) +Body (Times New Roman)

| Page 15: [238] Formatted | Luyuan Zhang | 1/23/20 11:49:00 PM |
|---|---|---|

Font: (Default) +Body (Times New Roman)

| Page 15: [239] Formatted | Luyuan Zhang | 1/23/20 11:49:00 PM |
|---|---|---|

| Page 15: [240] Formatted | Luyuan Zhang | 1/23/20 11:49:00 PM |
|---|---|---|

Font: (Default) +Body (Times New Roman)

| Page 15: [241] Formatted | Luyuan Zhang | 1/23/20 11:49:00 PM |
|---|---|---|

Font: (Default) +Body (Times New Roman)

| Page 15: [242] Formatted | Luyuan Zhang | 1/23/20 11:49:00 PM |
|---|---|---|

Font: (Default) +Body (Times New Roman)

| Page 15: [243] Formatted | Luyuan Zhang | 1/23/20 11:49:00 PM |
|---|---|---|

Font: (Default) +Body (Times New Roman)

| Page 15: [244] Formatted | Luyuan Zhang | 1/23/20 11:49:00 PM |
|---|---|---|

Font: (Default) +Body (Times New Roman)

| Page 15: [245] Formatted | Luyuan Zhang | 1/23/20 11:49:00 PM |
|---|---|---|

Font: (Default) +Body (Times New Roman)

| Page 15: [246] Formatted | Luyuan Zhang | 1/23/20 11:49:00 PM |
|---|---|---|

Font: (Default) +Body (Times New Roman)

| Page 15: [247] Formatted | Luyuan Zhang | 1/23/20 11:49:00 PM |
|---|---|---|

Font: (Default) +Body (Times New Roman)

| Page 15: [248] Formatted | Luyuan Zhang | 1/23/20 11:49:00 PM |
|---|---|---|

Font: (Default) +Body (Times New Roman)

| Page 15: [249] Formatted | Luyuan Zhang | 1/23/20 11:49:00 PM |
|---|---|---|

Font: (Default) +Body (Times New Roman)

| Page 15: [250] Formatted | Luyuan Zhang | 1/23/20 11:49:00 PM |
|---|---|---|

Font: (Default) +Body (Times New Roman)

| Page 15: [251] Formatted | Luyuan Zhang | 1/23/20 11:49:00 PM |
|---|---|---|

Formatted

| Page 15: [251] Formatted | Luyuan Zhang | 1/23/20 11:49:00 PM |
|---|---|---|

Formatted

| Page 15: [252] Formatted | Luyuan Zhang | 1/23/20 11:49:00 PM |
|---|---|---|

Font: (Default) +Body (Times New Roman)

| Page 15: [253] Formatted | Luyuan Zhang | 1/23/20 11:49:00 PM |
|---|---|---|

Font: (Default) +Body (Times New Roman)

| Page 15: [254] Formatted | Luyuan Zhang | 1/23/20 11:49:00 PM |
|---|---|---|

| Page 15: [255] Formatted | Luyuan Zhang | 1/23/20 11:49:00 PM |
|---|---|---|

Font: (Default) +Body (Times New Roman)

| Page 15: [256] Formatted | Luyuan Zhang | 1/23/20 11:49:00 PM |
|---|---|---|

Font: (Default) +Body (Times New Roman)

| Page 15: [257] Formatted | Luyuan Zhang | 1/23/20 11:49:00 PM |
|---|---|---|

Font: (Default) +Body (Times New Roman)

| Page 15: [258] Formatted | Luyuan Zhang | 1/23/20 11:49:00 PM |
|---|---|---|

Font: (Default) +Body (Times New Roman)

| Page 15: [259] Formatted | Luyuan Zhang | 1/23/20 11:49:00 PM |
|---|---|---|

Font: (Default) +Body (Times New Roman)

| Page 15: [260] Formatted | Luyuan Zhang | 1/23/20 11:49:00 PM |
|---|---|---|

Font: (Default) +Body (Times New Roman)

| Page 15: [261] Formatted | Luyuan Zhang | 1/23/20 11:49:00 PM |
|---|---|---|

Font: (Default) +Body (Times New Roman)

| Page 15: [262] Formatted | Luyuan Zhang | 1/23/20 11:49:00 PM |
|---|---|---|

Font: (Default) +Body (Times New Roman)

| Page 15: [263] Formatted | Luyuan Zhang | 1/23/20 11:49:00 PM |
|---|---|---|

Font: (Default) +Body (Times New Roman)

| Page 15: [264] Formatted | Luyuan Zhang | 1/23/20 11:49:00 PM |
|---|---|---|

Font: (Default) +Body (Times New Roman)

| Page 15: [265] Formatted | Luyuan Zhang | 1/23/20 11:49:00 PM |
|---|---|---|

Font: (Default) +Body (Times New Roman)

| Page 15: [266] Formatted | Luyuan Zhang | 1/23/20 11:49:00 PM |
|---|---|---|

Font: (Default) +Body (Times New Roman)

| Page 15: [267] Formatted | Luyuan Zhang | 1/23/20 11:49:00 PM |
|---|---|---|

Font: (Default) +Body (Times New Roman)

| Page 15: [268] Formatted | Luyuan Zhang | 1/23/20 11:49:00 PM |
|---|---|---|

Font: (Default) +Body (Times New Roman)

| Page 15: [269] Formatted | Luyuan Zhang | 1/23/20 11:49:00 PM |
|---|---|---|

Font: (Default) +Body (Times New Roman)

| Page 15: [270] Formatted | Luyuan Zhang | 1/23/20 11:49:00 PM |
|---|---|---|

Font: (Default) +Body (Times New Roman)

**Supplementary Information**

[revised manuscript text omitted]

90     **Fig. S1 Relationship between [127]I and [129]I with a weak correlation (R=0.33, p<0.01) between the two iodine isotopes. This indicates the two iodine isotopes have different sources and their temporal variation patterns were affected by different factors.**

[Figure]

[Figure]

95    **Fig. S2 Relationship between iodine isotopes and total suspended particles (TSP) in Xi'an, China (n=68), suggesting significant correlation between $^{127}$I and TSP, and no correlation between $^{129}$I and TSP. The results indicate $^{127}$I was sourced from local input and $^{129}$I was transported to the studied site externally.**

**Fig. S3 Back trajectories analyisis on date of a) 18th April, 2017; b) 18th May, 2017; c) 14th July, 2017; d) 31st August, 2017; e) 6th September, 2017; f) 15th November, 2017; g) 28th December, 2017; h) 17th January, 2018.**

[Figure]

**Fig. S4 Relations between $^{127}I$ and air pollutants including PM10, PM2.5, SO₂, NO₂, CO and O₃, showing significant correlation.**

[Figure]

**Fig. S5 850 hPa water vapor transmission flow field on 2 May, 2017 (a), and 21 May, 2017 (b). Data from: https://cmdp.ncc-cma.net/Monitoring/monsoon.php?ListElem=vt85. The red dot in the figures is the sampling location, Xi'an, China.**

115 **Table S1 Mean $^{129}$I concentrations and $^{129}$I/$^{127}$I ratios in three high-level periods (HLP) and two low-level periods (LLP)**

| No | Type | Start date | Stop date | N($^{129}$I) / ($10^5$ m$^{-3}$) | | $^{129}$I/$^{127}$I number ratio / ($\times 10^{-10}$) | | Monsoon stage |
|---|---|---|---|---|---|---|---|---|
| | | | | Average | RSD | Average | RSD | |
| 1 | HLP 1 | 28 Mar, 2017 | 22 May, 2017 | 2.37 | 91% | 101 | 89% | WM and onset of SM |
| 2 | LLP 1 | 23 May, 2017 | 25 Jul, 2017 | 0.49 | 60% | 28.5 | 65% | Active of SM |
| 3 | HLP 2 | 4 Aug, 2017 | 12 Sep, 2017 | 1.98 | 109% | 155. | 141% | Break of SM |
| 4 | LLP 2 | 21 Sep, 2017 | 11 Oct, 2017 | 0.66 | 44% | 40.1 | 44% | Revival of SM |
| 5 | HLP 3 | 13 Oct, 2017 | 20 Mar, 2018 | 2.41 | 44% | 67.9 | 83% | SM retreat and WM advance then active |

×

---

## Author Response (AR4)

**Response to Editor**

Dear Editor Dr. Jan Kaiser,

Thanks for your great effort on our manuscript.
According to the constructive comments from the two anonymous reviewers, we have carefully revised our manuscript, improved the language with the help of native English speaker and also corrected some writing errors.

Below are our major modifications in this manuscript.

1. Three figures (Figure 3, 4 and 6) have been moved from the Supplementary Information file to the manuscript to make our conclusions more convincing. The font of Figure 5 (original Figure 3) have been adjusted larger.
2. As commented by the reviewer, since iodine isotopes data did not distribute normally, in Table 1 we use Spearman correlation factors instead of Pearson correlation factors to indicate the relationship. Despite such a change, the correlation using Spearman shows almost same as Pearson, which does not affect our discussion and not change the final conclusions.
3. The statistical data, such as average values, standard deviations and correlation factors, have been carefully checked and recalculated. And we found a calculation mistake of average $^{129}I/^{127}I$ atomic ratios, which should be $(92.7\pm124) \times10^{-10}$, not $(101\pm124) \times10^{-10}$. The reason for this mistake is that four data in spring 2018 were input into wrong columns. We apologize for our carelessness. All data have been corrected throughout the whole manuscript. The data correction did not change our conclusion.
4. The point of view about iodine's role on the formation of urban fine particles have been deleted. Although we found significantly positive correlation between iodine and air pollutants (except ozone), there still lack of direct evidence to prove that iodine contributes to the primary particle formation.

Best regards

Luyuan Zhang
1/1/2020

*Response to Interactive comment from Anonymous Referee #1*

The paper acp-2019-818-manuscript-version1 entitled "Temporal variation of 129I and 127I in aerosols from Xi'an, China: influence of East Asian monsoon and heavy haze events" provides interesting data for the distribution of iodine isotopes in the aerosols of part of China which are missing from the international data base. The paper has the potential for publication after revision as given below.

We are very grateful for the reviewer's positive and constructive suggestions and comments that make our manuscript better. According to the reviewer's comments, the main modifications are made in the revised version:
1) The language is further polished, including revising the grammar mistakes and shortening the long sentences;
2) The Figs 1, 3, S1, S2, S7 and S8 are reorganized and adjusted to make the discussion part more easily understood and convincing.
   The comments are responded item by item as below.

1. The paper needs some linguistic revision as there are many grammatic mistakes.

**Response**: Sorry for such basic mistakes. We have checked throughout the context and revised all the linguistic mistakes.

2. Lines 44-45 "As a consequence of these point sources of 129I, the distribution of 129I is rather uneven (Snyder et al., 2010)? Where?

 **Response**: These point sources of $^{129}$I have been listed in the reference of Snyder et al., 2010, including the principal nuclear reprocessing plants in Russia, UK, France, USA, Pakistan, China, Israel, India, South Africa and Argentina, nuclear accidents in Chernobyl, Former Soviet and Fukushima, Japan. For brief introduction, only references are cited here, not listing all these specific sources. This sentence has been revised to be "As a consequence of $^{129}$I releases from NFRPs, nuclear accidents and nuclear weapon testing sites, the global distribution of $^{129}$I is rather uneven (Snyder et al., 2010; Xu et al., 2015)."

3. The paragraph between lines 50 and 55 is long and difficult to follow and could be rewritten to focus on aims of the study.

**Response**: Line 50 has been separated into two parts, and revised to be "And those previous studies present the time series of $^{129}$I in aerosols in monthly resolution for the purpose of nuclear environmental monitoring. Such a low time-resolution is not sufficient to understand the source, transport and temporal variation pattern and its influencing factor of $^{129}$I."
       Line 51-55 paragraph has been reorganized into three sentences as below.
       "Here, we present a day-resolution temporal variation of $^{129}$I and $^{127}$I in aerosols during 2017/2018 from a typical monsoonal zone, Xi'an city in the Guanzhong Basin of northwest of China, to make attempts to investigate the level, sources and temporal change characteristics of $^{127}$I and $^{129}$I. This study will help to establish a background value of $^{129}$I/$^{127}$I ratio serving the nuclear environmental safety monitoring. The possible influencing factors

on temporal variation of iodine isotopes are also explored, including meteorological parameters, East Asian monsoon (EAM) and heavy haze events."

4. It is not clear how much time is the "a day-resolution" sampling reflects in term of iodine residence time in the atmosphere?

Response: Iodine residence time is closely related to its species and the associated states with particles, generally ranging from a few seconds to a few days for gaseous iodine species (Saiz-Lopez et al., 2012). For particle-associated iodine, its residence time is much dependent on particle size, varying from 0.1 day to 10 days (< 1 day for particles large than ~ 1 μm, and > 1 day for particles smaller than ~ 1 μm) (Moyers and Duce, 1972). In this study, total suspended particles were sampled in a 24 h resolution, reflecting a relatively equilibrated state for gaseous-particle converted iodine and large particle-associated iodine with shorter residence time (< 1 day), and a more variable information of iodine in small particles. Such a time-resolution can benefit greatly our future work on speciation analysis of atmospheric iodine to dive deep into atmosphere iodine processes.

Referece:

Moyers, J.L., Duce, R.A., 1972. Gaseous and particulate bromine in the marine atmosphere. J. Geophys. Res. 77, 5330–5338. https://doi.org/10.1029/jc077i027p05330

Saiz-Lopez, A., Gómez Martín, J.C., Plane, J.M.C., Saunders, R.W., Baker, A.R., Von Glasow, R., Carpenter, L.J., McFiggans, G., 2012. Atmospheric chemistry of iodine. Chem. Rev. 112, 1773–1804.

5. Figures 1 a and b can be combined in one figure.

Response: Figure 1 has been reorganized. Figure 1b as an inset has been combined with 1a.

6. The results part needs further additions from the supplementary data including Figs. S1 and S2.

Response: The results including Figs. S1 and S2 in the supplementary data have been added into the manuscript.

7. Connection of iodine chemical forms (I-127 and I-129) from the sources and in the atmosphere may elucidate some of the inconclusive correlations and relationship to spatial and temporal atmospheric transport on short and long distances.

Response: We strongly agree with this comment. The short- and long-range transport of airborne iodine is strongly related with its chemical forms. The present study focuses on aerosol iodine, and the gaseous iodine ([127]I and [129]I) sample collection and analysis are under way, while no data has been available at present. We do expect that further work on iodine chemical forms in air would give further understanding on the relationship.

8. More elaboration of weathering of basement rocks as a source of I-129 will be interesting.

Response: In line 226-236, more explanation has been added as below. "Weathering of bed rock is not also a major source of airborne [129]I, since weathering just contributes 5% of stable

iodine, and [129]I in bed rock can be considered even lower the nature-produced [129]I level because of the continuous decay."

9. Addition of Figure S7 and S8 to the discussion section will enhance the understanding of the atmospheric transport pathways of the isotopes.

**Response**: For improving the understanding, Figure S8 and SI-4 discussion part have been moved to the manuscript. Considering too many figures and each importance, Figure S7 is still kept in the Supplementary Information.

10. More details on the paragraph in lines 100-104 can add clarity to general statement with respect to 127I distribution in China.

**Response**: Line 100-104 have been added more discussion of aerosol iodine in China. "Whereas, a similar range of TSP [127]I was observed to be 4.5-22 ng m$^{-3}$ at coastal urban, Shanghai, China, and iodine concentration were lowest in summer and an increase occurred in fall and winter (Gao et al., 2010). Iodine associated with PM10 and PM2.5 were found to be 3.0-115 ng m$^{-3}$ and 4-18 ng m$^{-3}$, respectively, in urban and island sites of Shanghai, slightly lower than TSP iodine (Cheng et al., 2017; Gao et al., 2010). The marine aerosol iodine offshore China was found below 8.6 ng m$^{-3}$ during the XueLong cruise from July to September 2008 (Xu et al., 2010). These results suggest a relatively high aerosol [127]I level in both inland and coastal urbans in China."

Referece:

Cheng, N., Duan, L., Xiu, G., Zhao, M., Qian, G., 2017. Comparison of atmospheric PM2.5-bounded mercury species and their correlation with bromine and iodine at coastal urban and island sites in the eastern China. Atmos. Res. 183, 17–25. https://doi.org/10.1016/j.atmosres.2016.08.009

Gao, Y., Sun, M., Wu, X., Liu, Y., Guo, Y., Wu, J., 2010. Concentration characteristics of bromine and iodine in aerosols in Shanghai, China. Atmos. Environ. 44, 4298–4302. https://doi.org/10.1016/j.atmosenv.2010.05.047

Xu, S., Xie, Z., Li, B., Liu, W., Sun, L., Kang, H., Yang, H., Zhang, P., 2010. Iodine speciation in marine aerosols along a 15000-km round-trip cruise path from Shanghai, China, to the Arctic Ocean. Environ. Chem. 7, 406–412.

11. The anthropogenic source for I-127 is mainly related to coal consumption (local source) whereas the I-129 source is mainly related to far away transport. It will be good to provide some details of how these isotopes are associated in the atmosphere with respect to airmasses altitude, chemistry and residence time of the isotope.

**Response**: The mechanism for iodine association with particles with many uncertainties, generally has two pathways, iodine compounds as primary nuclei during fine particle formation, and adsorption onto naturally occurring particles (Garland, 1967; Saiz-Lopez et al., 2012). In section 4.2.2, association of [127]I and [129]I with particles has been elucidated in respect of nucleation process and residence time. However, some interpretations need more evidence. Chemical processes could definitely affect iodine species in aerosols, such as

inorganic iodide and iodate, the former of which has been found in a large proportion of inorganic iodine likely because of the presence of reductant $SO_3$ (Zhang et al., 2016). As this paper focuses on the temporal variation of total iodine isotopes and there is no more data to suggest that chemical process would significantly affect the total iodine change, therefore, it is hard to discuss the influence of chemical processes on iodine bound to particles. Our future work on aerosol iodine species will put more efforts on this point. In section 4.2.3, "Furthermore, the back trajectory analysis also showed that the low $^{129}I$ level on April 18 can be partially attributed to an $^{129}I$-poor low-altitude air mass (< 900m) (Fig.S3a), since either they might be formed in $^{129}I$-poor inland areas, not from the $^{129}I$-rich European area, or long-range transported $^{129}I$ in low-altitude air mass could be easily lost by the topographic countercheck (Dong et al., 2018)." has been modified to explain the influence of airmass altitude.

Referece:

Dong, Z., Shao, Y., Qin, D., Zhang, L., Hou, X., Wei, T., Kang, S., Qin, X., 2018. Insight Into Radio-Isotope 129I Deposition in Fresh Snow at a Remote Glacier Basin of Northeast Tibetan Plateau, China. Geophys. Res. Lett. 0. https://doi.org/10.1029/2018GL078480

Garland, J.A., 1967. The adsorption of iodine by atmospheric particles. J. Nucl. Energy 21, 687–700.

Zhang, L., Hou, X., Xu, S., 2016. Speciation of 127I and 129I in atmospheric aerosols at Risø, Denmark: Insight into sources of iodine isotopes and their species transformations. Atmos. Chem. Phys. 16, 1971–1985.

12. May be good to make the text in Figure 3 in larger font.

**Response**: The font in Figure 3 has been adjusted larger.

13. Although the authors pointed out the possible use of air masses transport to predict iodine sources and impact on future iodine distribution, it is still not clear how the iodine data enhance our understanding of the climate or atmospheric circulation.

**Response**: Here we use a set of day-resolution iodine isotopes data to establish the crucial linkage with East Asia monsoon system and meteorological conditions. On this basis, long-term observation of iodine isotopes would refresh the understanding of climate change, not just one-year short-term iodine data presented in this paper. Moreover, high time-resolution iodine data in combination with modelling should be expected. Before make the application on climate and atmospheric circulation, plenty of questions have to be well answered, for instance, how airborne iodine species like and interact with each other.

**Response to Interactive comment from Anonymous Referee #2**

This manuscript reports the concentrations and ratios of 129-I and 127-I in aerosol samples collected over a period of approximately one year at Xi'an in China. The data are interpreted in terms of the dominant sources and transport pathways of these iso- topes to the site, and the discussion considers the influence of the fluctuating modes of the East Asian Monsoon on the observed record. The subject matter is highly relevant for Atmospheric Chemistry and Physics and the authors have a track record of producing high quality data from the demanding measurements employed. However, the manuscript suffers from a number of shortcomings, including factual errors and subjective and unsupported interpretations. I think that this will be an excellent contribution once these have been addressed. There are many minor errors in the English used, but the meaning of the manuscript is still clear.

We thank the admirable reviewer for the positive evaluation and providing us these constructive comments. The reviewer has a very deep understanding and rich experience on iodine study area. It is also our honor to have such valuable suggestions and comments, which significantly improve the quality of this manuscript. We are in complete agreement that some interpretations are not fully supported by the current evidence, for instance, the associated mechanisms of locally released $^{127}$I and externally input $^{129}$I with particles in urban atmosphere, whether they are mainly involved into primary particle formation or scavenged by existing particles. To answer this, more research is needed in the future.

Following the detailed comments, we have carefully checked throughout the content, made all English corrections and revised the manuscript. Below are our responses the comments item by item.

**Major comments**

1. There are numerous instances of inconsistent units being used for iodine concentrations for the Xi'an site in the Results section (line 77 onwards). In the text, the units are frequently given as micrograms per cubic metre, while in the figures and supplementary material the units are nanograms per cubic metre. Since the numbers in both cases are the same, one of the units must be incorrect. I assume that the units should actually be nanograms per cubic metre, but please check and correct.

**Response**: Sorry for the basic mistakes on the incorrect unit of $^{127}$I concentrations. As the reviewer commented, $^{127}$I concentrations in aerosols should be nanograms per cubic metre, not microgram per cubic metre. All the unit mistakes have been carefully checked and revised.

2. On line 91 the authors state that "A weak correlation between 129I and 127I was found with a Pearson correlation coefficient of 0.34 (p=0.01) for the whole year data, while no significant correlation between the two iodine isotopes in each season at the level of 0.05 (Table 1 and Fig. S3)." This does not agree with the statement made in the caption to Fig S3: "Relationship between 127I and 129I, showing no significant correlation (R=0.265) between the two iodine isotopes". Why do these statements not agree? Since it is apparent from Fig S3 that the dataset is not normally distributed, I would suggest the authors use a non-parametric regression method (such as Spearman's Rank Correlation) instead of Pearson's for

all regression analysis in the manuscript. This will give far more robust results. Perhaps Figure S3 might be more informative if plotted with different symbols for the time periods of interest.

**Response**: According to the reviewer's comment, we use Spearman coefficient to discuss the correlation. Although $^{127}I$ and $^{129}I$ data are not normally distributed, Pearson and Spearman coefficients are typically identical. The inconsistency between Table 1 and Figure 3 is resulted from numbers of data used for calculation. The Pearson coefficient of 0.26 is used for all the 68 data points.

It is a good idea to replot Figure S3 using different symbols. We have tried in this way as shown below. The Figure 1 below is plotted with different symbols and fitting trends for the four seasons, clearly showing the concentration distribution in different seasons. While no more information could be obtained because the previous Figure S1 and S2 (now move to the context as Figure 3 and 4, respectively) have clearly showed the information. Therefore, Figure S3 is kept as before only with a small revision by changing the Pearson correlation coefficient to Spearman coefficient.

[Figure]

Figure 1. Relation of $^{127}I$ and $^{129}I$ indicated by different symbols and trend lines for four seasons

3. I am not quite sure how the authors have used the values published in Saiz-Lopez et al., 2012 to compare to the results obtained at Xi'an. In Table 5 of Saiz-Lopez, aerosol iodine concentrations of up to 25 ng m-3 and >3.3 ng m-3 are quoted for open ocean and continental sites respectively. These do not seem to relate to the values for "terrestrial" (1 ng m-3) and "marine" (<10 ng m-3) air quoted in lines 100 & 101. The higher values in Saiz-Lopez et al. also do not give strong support to the statement in the last sentence of this paragraph (lines 102-103).

**Response**: We agree with the reviewer that this statement and citation of Saiz-Lopez is vague. Therefore, we revise lines 100-104, and also add more data in China. This paragraph was modified as below.

"The level of $^{127}I$ concentrations, in particular in winter, is much higher than those in continental sites (below 0.61 ng m$^{-3}$ in South Pole and 2.7-3.3 ng m$^{-3}$ in the Eastern

Transvaal), and comparable to those in coastal and ocean sites (typically below 20 ng m$^{-3}$, and up to 24 ng m$^{-3}$ in tropic marine aerosols) (Saiz-Lopez et al., 2012). A similar range of TSP $^{127}$I was observed to be 4.5-22 ng m$^{-3}$ at coastal urban, Shanghai, China, showing lowest in summer and an increase occurred in fall and winter (Gao et al., 2010). Iodine associated with PM10 and PM2.5 were found to be 3.0-115 ng m$^{-3}$ and 4-18 ng m$^{-3}$, respectively, in urban and island sites of Shanghai, slightly lower than TSP iodine (Cheng et al., 2017; Gao et al., 2010). The maximum of marine aerosol iodine offshore China was found below 8.6 ng m$^{-3}$ during the XueLong cruise from July to September 2008 (Xu et al., 2010). These results suggest a relatively high aerosol $^{127}$I level in both inland and coastal urbans in China."

4. The statement about natural sources of iodine (lines 104-105) comes from a rather old source (Fuge & Johnson, 1986). While it is true that sea spray contributes iodine to the atmosphere, we now know that gas-phase emissions of iodine from the ocean are a much stronger source (see, for example, Carpenter et al., 2013 – which the authors cite later in the manuscript). Thus the study of He (2012) which apparently used sodium concentrations to estimate the seaspray contribution of iodine to precipitation at Zhouzhi county almost certainly greatly underestimated the "direct contribution of ocean". (The citation of He 2012 in the reference list does not give sufficient information for the source to be found).

**Response**: Accept. The reference "Carpenter et al., 2013" is added as "Natural iodine is from marine emission through sea spray, weathering of base rock and continental release through vegetation and suspended soil particles (Carpenter et al., 2013; Fuge and Johnson, 1986)."

Because of direct emission of gaseous iodine from sea surface, we agree that marine iodine contribution in the reference of He (2012) would be underestimated when using Na$^+$ as reference element for calculation. In this reference, spatial distribution of iodine in rainwater and surface freshwater were also reported. Despite being underestimated, sea source contribution of iodine showed a decline trend with increasing distance from the sea until 100 km, over which no significant change of marine contribution could be found. Our study site, Xi'an, is an inland city about 900 km from the nearest sea. It is therefore not likely that marine source (including sea spray, direct volatilization and gaseous emission) is the major contribution of iodine.

5. In lines 114 - 118, the authors attempt to balance estimated emissions of iodine from terrestrial soil and vegetation (from Sive et al., 2007) against an estimate of iodine deposition flux. There is insufficient detail given of how this deposition flux calculation was done, but it appears to be based on "dust fall". Better explanation is required if this calculation is to be understood. Does "dust" here refer to mineral dust? If so, why should its deposition be specifically associated with the deposition of iodine? How exactly was the calculation done? The value given for the terrestrial emission flux (2.27 ug m-2 d-1) does not seem to agree with the value given by Sive et al. (2.7 ug m-2 d-1). How reliable is the comparison likely to be when the emission flux estimate is derived entirely from observations in North America, where vegetation types and land surfaces are different from the study region here?

**Response**: As commented by the reviewer, the dry deposition flux of $^{127}$I is not specific enough. In this manuscript, the dry deposition flux of $^{127}$I is calculated by $^{127}$I mass concentration in total suspension particles multiplying the average dust fall flux. Since it is hard to know the deposition velocity of total suspension particles, we use dustfall flux for approximate calculation. Dust in this manuscript refer to natural dust, not but including mineral dust in air.

The dustfall is collected by wet method, i.e. a 20 cm in diameter ×30 cm height container with enough deionized water. There are 14 sampling sites in Xi'an. The natural dustfall ranges within 4.5-47.8 t (km$^{-1}$ 30 d$^{-1}$) with an annual mean of 13.2 t (km$^{-1}$ 30 d$^{-1}$). According to the reference Yang et al., 2017 listed below, the annual dustfall flux in 2014 at Qujiang District, about 2km from our sampling site was 11.76±3.65 t (km$^{-1}$ 30 d$^{-1}$). The uncertainty for the dustfall flux is 31%, and iodine concentration uncertainty is within 5%, resulting in a total uncertainty of 32%. We have given a more detailed description about this calculation in the context.

Reference: Yang Wenjuan, Chen Ying, Zhao Jianqiang, et al. Spatial and temporal variation of atmospheric deposition pollution in Xi'an City. Environmental Science & Technology (in Chinese),2017,40(3):10-14.

In reference Sive et al., 2007, 2.7 µg m$^{-2}$ d$^{-1}$ and 2.27 µg m$^{-2}$ d$^{-1}$ were presented in abstract and Section 5, respectively. The value of 2.7 µg m$^{-2}$ d$^{-1}$ just occurred in Abstract, lacking of calculation details. The average terrestrial emission flux (2.27 µg m$^{-2}$ d$^{-1}$) was estimated, on a global basis, over an active season of 240 days, together with biome areas for temperate forest and wood lands (28.5 ×10$^{12}$ m$^2$) and temperate grasslands (31.9×10$^{12}$ m$^2$). Therefore, we cite the value of 2.27 µg m$^{-2}$ d$^{-1}$ because of its clear mathematical description.

The terrestrial emission flux by Sive et al., 2007 should be much higher than that in urban environment, since a part of the urban land is covered by houses and roads without iodine emission.

6. Have the authors considered the influence of seasonal changes in boundary layer height on aerosol iodine concentrations? These could potentially be significant, and could cause changes in surface level concentrations even when emission fluxes are constant.

**Response**: This is a very good point to consider the boundary layer height, which is closely related with air pollution, and can indicate the vertical dispersion scale of air pollutants by thermal turbulent mixing. Not only the boundary layer height (BLH), but also the atmospheric stability (AS) could directly affect the concentration and time-space distribution of pollutants. And they might be important factors to control the variation of iodine isotopes. To be honest, at present, we have no idea about the impact. In future, we would like to make further investigation for 3-4 years to evaluate, to what extent the BLH and AS have influence on variation of iodine isotopes.

7. While I understand that the authors' estimate of the potential contribution of coal combustion to aerosol iodine loading at their study site is only intended as a first-order estimate, I do not think that they have sufficient information to attempt it. The assumption that surface iodine emissions are mixed through the entire troposphere (i.e. to 10 km) is certainly not realistic, since only a small proportion of emissions are likely to leave the boundary layer (~1 km). This implies an order of magnitude greater iodine concentration derived from coal, which does not appear to be plausible.

**Response**: We agree that this calculation of aerosol iodine from coal iodine is not plausible, so the following statement has been deleted. "The area of the Guanzhong Basin is 3.6×10$^4$ m$^2$, and the height of troposphere is taking as 10 km. Then, $^{127}$I concentration in the air is about 250 ng m$^{-3}$. The particle-associated iodine accounts for approximately 10%-20% (Hasegawa

et al., 2017). Thus, $^{127}I$ in aerosols can be estimated to be about 25-50 ng m$^{-3}$. The estimated value is comparable with the $^{127}I$ peak values in winter, but about ten times higher than the less polluted aerosol $^{127}I$ concentrations (1.21-9.01 ng m$^{-3}$).”

8. Lines 221 – 222: “Two severe dust storm events occurred in Xi'an in 17-18 April and 4-6 May, 2017, as shown by the peaks of air quality index (AQI) of 268 and 306, respectively (Fig. 2e).” There is only one peak in AQI visible in Fig 2e in this time period. Please explain or amend.

**Response**: Thanks for pointing out this flaw. It is right that only the first dust storm in 17-18 April have been shown in Figure 2e, because no sample was analysed during the second sand storm in 4-6 May. Thus, we give the AQI values for the two events. Below is Figure 2 for the daily measurement of AQI, from which we can see two peaks of the dust storm events. After careful thinking and for simplification, we decide to use AQI data on the days with iodine isotopes values. Thus, we have revised the statement as “Two severe dust storm events occurred in Xi'an in 17-18 April and 4-6 May, 2017, as indicated by the peaks of air quality index (AQI) of 268 and 306, respectively.”

[Figure]

Figure 2. Temporal variation of $^{127}I$ and AQI during March 2017 to March 2018, showing the correlation of $^{127}I$ and AQI.

9. Please give further explanation of the significance of the “low-altitude air mass” mentioned on line 232.

**Response**: Further explanation has been added as below. “Furthermore, the back trajectory analysis also showed that the low $^{129}I$ level on April 18 can be partially attributed to an $^{129}I$-poor low-altitude air mass (< 900m) (Fig.S3a). This is because either the low-altitude air mass might be formed in $^{129}I$-poor inland areas, not from the $^{129}I$-rich European area, or long-range transported $^{129}I$ in low-altitude air mass could be easily lost by the topographic countercheck (Dong et al., 2018).”

10. On lines 250 – 254 (and later in the manuscript) the authors discuss the possibility that the aerosol iodine they observed might have formed through primary nucleation. While there are relationships between iodine concentration and those of other species associated with nucleation (e.g. SO2, Fig S6), it is also apparent that the concentration of SO2 is three orders of magnitude greater than that of aerosol iodine. There is no evidence available in this dataset that would make it possible to determine whether iodine is incorporated into aerosol in Xi'an via primary formation or secondary uptake onto existing particles. I therefore suggest that

discussion of the iodine aerosol formation mechanism can only be speculation, and it would be better to remove it entirely.

**Response**: We agree that the mechanism of iodine association with particle is not well understood on the basis of our data. Therefore, these corresponding statements in Section 4.2.2 has been removed as below.

"Typically, new particle formation occurs in two distinct stages, i.e., nucleation to form a critical nucleus and subsequent growth of the freshly nucleated particle to a larger size (Zhang et al., 2015). It is widely accepted that iodine is involved into the formation of fine particles, and increasing investigations have been carried out in coastal and open sea areas (Saiz-Lopez et al., 2012). However, in megacities with severe air pollution, the role of iodine on formation and development of heavy haze events is far not understood. Iodine-mediated particles were suggested to be formed from highly concentrated, localized pockets of iodine oxides as primary nucleation, and to rapidly grow by uptake of $H_2SO_4$, $H_2O$, $NO_2$, short chain dicarboxylic acids, gaseous iodine and other gaseous species (Saiz-Lopez et al., 2012). Winter urban air in Xi'an provides two requirements of sufficiently high iodine concentrations and the presence of high levels of aerosol nucleation precursors, such as $SO_2$, $NH_3$, amines, and anthropogenic VOCs."

"In spring and summer, iodine is probably associated with primary matters and secondary organic aerosols due to low level of air iodine and greatly increased artificial and biogenic VOCs (Feng et al., 2016). In fall and winter when the key aerosol nucleation precursors are noticeably elevated, the significantly positive correlation between [127]I and these precursors indicates that locally emitted iodine is likely involved into formation of secondary inorganic aerosols, while externally input [129]I may not occur in the nucleation of secondary inorganic aerosols."

"The minimum in ozone concentrations on 15 November and 14 December, 2017 may support iodine-containing aerosol nucleation process, in which ozone acted as oxidant and reactant to form iodine oxidizes, and aggregated into high valence iodine oxidizes (Saiz-Lopez et al., 2012). This study suggests iodine is closely related to aerosol formations, and high level of iodine likely facilitates the growth of fine particles along with major aerosol precursors particularly during haze episodes."

11. In section 4.2.3 the authors make a convincing case for the influence of interactions with the East Asian Monsoon on long-range 129-I transport to the study site. I am not familiar with the EAWM index mentioned on line 303, but I wonder whether it is possible to make more use of this when exploring the variations in iodine isotope concentrations and their ratios during the study. Can it be plotted on Fig 2? The "z-score" approach discussed on lines 333 – 336 would be more convincing if it could be combined with some quantitative indicator of EAM strength.

**Response**: When we prepare this manuscript, we have actually plotted the EAWM and EASM indexes with [129]I variation as shown by Figure 3 below. It is quite interesting that the fluctuation of [129]I concentrations have some close relation with these indexes. Whereas, this is our first try to link the monsoon strength with [129]I variation, so that we could not understand it deeper at present. We also expect to do further work on this.

[Figure]

Figure 3. Variation of $^{129}$I and EAWM (top) and EASM (bottom) indexes during 2017-2018

The "z-score" method gives clear indications for different monsoon stages, so Figure S8 has been moved into the manuscript as Figure 6, together with the statements.

12. On lines 301 – 302 it is stated that "In addition, the 129I level in March 2018 was much less than that in March 2017". This is certainly true, and in fact the 129I concentration in March 2018 is very similar to that in the two LLP periods. Why did the authors choose to include these samples in the HLP period?

**Response**: We can find that the fluctuation of $^{129}$I is very large during the HLP periods, but with low concentrations down to the level same as LLP periods. In March 2018, only four data are available. Considering this period is under control of EAWM, these low values were likely as a consequence of fluctuation, and therefore categorized into the HLP period.

13. The statement on line 338 that the iodine isotope ratio shows "relatively weak fluctuation" seems rather subjective, and quite surprising given the relative standard deviation quoted for the parameter of >120%. There are strong variations in the ratio during the HLP 2 period, which do not appear to be consistent with the statement on line 339 about background levels.

**Response**: Agree. The subjective statement "Both two iodine isotopes show apparently temporal changes in northwestern China, while $^{129}$I/$^{127}$I ratios show relatively weak fluctuation (Fig.2c)." has been deleted.

14. Minor comments Line 61: replace "combing" with "combined"? Line 75: replace "ration" with "ratio" Line 178-179: Toyama et al. is cited both at the beginning and end of this sentence, but with different years. Please correct. Line 211: I think the correct units for ozone concentration here should be ppbv, not pptv.

**Response**: Line 61, "combing" has been revised to be "combined";
Line 75, "ration" has been revised to be "ratios";

Line 178-179: "Toyama et al. (2012)" at the beginning of this sentence has been revised to "Toyama et al. (2013)", and the citation at the end of the sentence has been deleted.

Line 211: We appreciate the reviewer for this unit mistake. After carefully checking the cited reference, ozone concentration here has been to revised to be ppbv.

[revised manuscript text omitted]

Font: (Default) Times New Roman

| Page 8: [7] Formatted | Luyuan Zhang | 12/16/19 4:00:00 PM |
|---|---|---|

Font: (Default) Times New Roman

| Page 8: [8] Deleted | Luyuan Zhang | 12/16/19 2:24:00 PM |
|---|---|---|

| Page 8: [8] Deleted | Luyuan Zhang | 12/16/19 2:24:00 PM |
|---|---|---|

| Page 8: [8] Deleted | Luyuan Zhang | 12/16/19 2:24:00 PM |
|---|---|---|

| Page 8: [8] Deleted | Luyuan Zhang | 12/16/19 2:24:00 PM |
|---|---|---|

| Page 8: [8] Deleted | Luyuan Zhang | 12/16/19 2:24:00 PM |
|---|---|---|

| Page 8: [8] Deleted | Luyuan Zhang | 12/16/19 2:24:00 PM |
|---|---|---|

| Page 8: [9] Deleted | Luyuan Zhang | 12/16/19 8:27:00 AM |
|---|---|---|

| Page 8: [9] Deleted | Luyuan Zhang | 12/16/19 8:27:00 AM |
|---|---|---|

| Page 8: [9] Deleted | Luyuan Zhang | 12/16/19 8:27:00 AM |
|---|---|---|

| Page 8: [9] Deleted | Luyuan Zhang | 12/16/19 8:27:00 AM |
|---|---|---|

| Page 8: [9] Deleted | Luyuan Zhang | 12/16/19 8:27:00 AM |
|---|---|---|

| Page 8: [9] Deleted | Luyuan Zhang | 12/16/19 8:27:00 AM |
|---|---|---|

| Page 8: [9] Deleted | Luyuan Zhang | 12/16/19 8:27:00 AM |
|---|---|---|

| Page 8: [10] Formatted | Luyuan Zhang | 1/9/20 11:10:00 AM |
|---|---|---|

Font: (Default) Times New Roman

| Page 8: [10] Formatted | Luyuan Zhang | 1/9/20 11:10:00 AM |
|---|---|---|

Font: (Default) Times New Roman

| Page 13: [11] Deleted | Luyuan Zhang | 12/16/19 6:28:00 PM |
|---|---|---|

| Page 13: [11] Deleted | Luyuan Zhang | 12/16/19 6:28:00 PM |
|---|---|---|

| Page 13: [12] Formatted | Luyuan Zhang | 1/7/20 3:20:00 PM |
|---|---|---|

Font color: Text 1

| Page 13: [12] Formatted | Luyuan Zhang | 1/7/20 3:20:00 PM |
|---|---|---|

Font color: Text 1

| Page 13: [12] Formatted | Luyuan Zhang | 1/7/20 3:20:00 PM |
|---|---|---|

Font color: Text 1

| Page 13: [13] Formatted | Luyuan Zhang | 1/7/20 3:20:00 PM |
|---|---|---|

Font color: Text 1

| Page 13: [13] Formatted | Luyuan Zhang | 1/7/20 3:20:00 PM |
|---|---|---|

Font color: Text 1

| Page 13: [13] Formatted | Luyuan Zhang | 1/7/20 3:20:00 PM |
|---|---|---|

Font color: Text 1

| Page 13: [14] Deleted | Luyuan Zhang | 1/7/20 2:09:00 PM |
|---|---|---|

| Page 13: [14] Deleted | Luyuan Zhang | 1/7/20 2:09:00 PM |
|---|---|---|

| Page 13: [15] Deleted | Luyuan Zhang | 1/7/20 2:11:00 PM |
|---|---|---|

| Page 13: [15] Deleted | Luyuan Zhang | 1/7/20 2:11:00 PM |
| --- | --- | --- |

| Page 13: [15] Deleted | Luyuan Zhang | 1/7/20 2:11:00 PM |
| --- | --- | --- |

| Page 13: [16] Deleted | Luyuan Zhang | 12/20/19 4:08:00 PM |
| --- | --- | --- |

| Page 13: [16] Deleted | Luyuan Zhang | 12/20/19 4:08:00 PM |
| --- | --- | --- |

| Page 13: [17] Deleted | Luyuan Zhang | 12/19/19 5:15:00 PM |
| --- | --- | --- |

| Page 13: [17] Deleted | Luyuan Zhang | 12/19/19 5:15:00 PM |
| --- | --- | --- |

| Page 13: [18] Deleted | Luyuan Zhang | 12/18/19 4:41:00 PM |
| --- | --- | --- |

| Page 13: [19] Formatted | Luyuan Zhang | 12/18/19 4:30:00 PM |
| --- | --- | --- |

Font: 7 pt

| Page 13: [19] Formatted | Luyuan Zhang | 12/18/19 4:30:00 PM |
| --- | --- | --- |

Font: 7 pt

| Page 13: [19] Formatted | Luyuan Zhang | 12/18/19 4:30:00 PM |
| --- | --- | --- |

Font: 7 pt

| Page 13: [19] Formatted | Luyuan Zhang | 12/18/19 4:30:00 PM |
| --- | --- | --- |

Font: 7 pt

| Page 13: [19] Formatted | Luyuan Zhang | 12/18/19 4:30:00 PM |
| --- | --- | --- |

Font: 7 pt

| Page 13: [19] Formatted | Luyuan Zhang | 12/18/19 4:30:00 PM |
| --- | --- | --- |

Font: 7 pt

| Page 13: [19] Formatted | Luyuan Zhang | 12/18/19 4:30:00 PM |
| --- | --- | --- |

Font: 7 pt

| Page 13: [19] Formatted | Luyuan Zhang | 12/18/19 4:30:00 PM |
| --- | --- | --- |

Font: 7 pt

| Page 13: [19] Formatted | Luyuan Zhang | 12/18/19 4:30:00 PM |
| --- | --- | --- |

Font: 7 pt

Font: 7 pt

| Page 13: [19] Formatted | Luyuan Zhang | 12/18/19 4:30:00 PM |

Font: 7 pt

| Page 13: [19] Formatted | Luyuan Zhang | 12/18/19 4:30:00 PM |

Font: 7 pt

| Page 13: [19] Formatted | Luyuan Zhang | 12/18/19 4:30:00 PM |

Font: 7 pt

| Page 13: [19] Formatted | Luyuan Zhang | 12/18/19 4:30:00 PM |

Font: 7 pt

| Page 13: [19] Formatted | Luyuan Zhang | 12/18/19 4:30:00 PM |

Font: 7 pt

| Page 13: [19] Formatted | Luyuan Zhang | 12/18/19 4:30:00 PM |

Font: 7 pt

[revised manuscript text omitted]

$$y = 0.0974x + 1.3604$$

[Figure]

Fig. S1 Relationship between $^{127}$I and $^{129}$I with a weak correlation (R=0.33, p<0.01) between the two iodine isotopes. This indicates the two iodine isotopes have different sources and their temporal variation patterns were affected by different factors.

215

[Figure]

Fig. S2 Relationship between iodine isotopes and total suspended particles (TSP) in Xi'an, China (n=68), suggesting significant correlation between [127]I and TSP, and no correlation between [129]I and TSP. The results indicate [127]I was sourced from local input and [129]I was transported to the studied site externally.

225

[Figure]

230 **Fig. S3 Back trajectories analyisis on date of a) 18th April, 2017; b) 18th May, 2017; c) 14th July, 2017; d) 31st August, 2017; e) 6th September, 2017; f) 15th November, 2017; g) 28th December, 2017; h) 17th January, 2018.**

[Figure]

235

Fig. S4 Relations between $^{127}$I and air pollutants including PM10, PM2.5, SO₂, NO₂, CO and O₃, showing significant correlation.

[Figure]

240

**Fig. S5 850 hPa water vapor transmission flow field on 2 May, 2017 (a), and 21 May, 2017 (b). Data from: https://cmdp.ncc-cma.net/Monitoring/monsoon.php?ListElem=vt85. The red dot in the figures is the sampling location, Xi'an, China.**

¶

[Figure]

**Fig. S8 Two-dimension graph of z-score normalized $^{129}I$ concentrations and $^{129}I/^{127}I$ ratios, suggesting the refined features of East Asia summer (onset, active, break and revival in yellow diamond, green triangle, red circle and pink circle, respectively) and winter monsoons (WM, black dot) (a). The colored symbols clearly demonstrate a detailed cycle of onset-active-break-revival for the summer monsoon with $Z_{129I}\leq$-0.5 and $Z_{Ratio}\leq$0, as illustrated in the blue oval area (b).¶**

Table S1 Mean $^{129}$I concentrations and $^{129}$I/$^{127}$I ratios in three high-level periods (HLP) and two low-level periods (LLP)

| No | Type | Start date | Stop date | $^{129}$I, × 10$^5$ atoms/m$^3$ | | $^{129}$I/$^{127}$I atomic ratio, ×10$^{-10}$ | | Monsoon stage |
|---|---|---|---|---|---|---|---|---|
| | | | | Average | RSD | Average | RSD | |
| 1 | HLP 1 | 28 Mar, 2017 | 22 May, 2017 | 2.37 | 91% | 101 | 89% | WM and onset of SM |
| 2 | LLP 1 | 23 May, 2017 | 25 Jul, 2017 | 0.49 | 60% | 28.5 | 65% | Active of SM |
| 3 | HLP 2 | 4 Aug, 2017 | 12 Sep, 2017 | 1.98 | 109% | 155 | 141% | Break of SM |
| 4 | LLP 2 | 21 Sep, 2017 | 11 Oct, 2017 | 0.66 | 44% | 40.1 | 44% | Revival of SM |
| 5 | HLP 3 | 13 Oct, 2017 | 20 Mar, 2018 | 2.41 | 44% | 67.9 | 83% | SM retreat and WM advance then active |

260

No ... [4]

| Page 3: [1] Deleted | Luyuan Zhang | 12/14/19 4:14:00 PM |
|---|---|---|
| Page 3: [2] Deleted | Luyuan Zhang | 12/12/19 6:13:00 PM |
| Page 3: [3] Deleted | Luyuan Zhang | 12/12/19 6:13:00 PM |
| Page 9: [4] Deleted | Luyuan Zhang | 12/12/19 6:01:00 PM |

**Response to Editor**

Dear Editor Dr. Jan Kaiser,

We are sincerely grateful for your kind help and efforts on our manuscript.

The manuscript and Supplementary Information have been revised based on your suggestions and comments, including the format, expression, usage of SI, the incompletely responded comments from Referee #2, etc. Below are our responses item by item.

Best regards and happy Chinese New Year!

Luyuan Zhang

Jan 21, 2020

ACP requires use of the International System of Units (SI). Therefore, please add the word "concentration" after the symbols 127I and 129I where you refer to concentrations (starting with ll. 15, 19 and 19 in the abstract, but at many places elsewhere, too). Chemical element symbols on their own do not indicate the physical quantity. Please also change the axis labels to include quantity symbols and units in quantity algebra notation, e.g. $\gamma$(127I)/(ng m-3), for the mass concentration of iodine-127 [I have chosen the symbol $\gamma$ rather than c to distinguish mass from molar concentrations], and N(129I)/(m-3), for the number concentration of iodine-129. Note that the word "atoms" should not form part of the unit. These symbols could also be used in the text for clarity and brevity, e.g. l. 78, "$\gamma$(127I), N(129I) and 129I/127I number ratios".

**Response**: All the expressions for $^{127}$I, $^{129}$I and the ratios in the manuscript and Supplementary Information have been revised according to the comment. And the quantity symbols and units in quantity algebra notation have also been revised in Figs. 2, 3, 4, and 5 in the updated manuscript and Figs. S1, S2, S4 and Table S1 in the Supplementary Information.

Referee #2 comment #6 on the relevance of boundary layer height for observed concentrations has not been addressed: Variations may occur even when emissions are constant. Your response should be reflected by appropriate additions to the manuscript.

**Response**: In order to address the possible influence of atmospheric boundary layer height (ABLH), a paragraph has been added into Section 4.2.1 (Line248-252). Here, we point out that the influences of ABLH should be different for locally input $^{127}$I and externally introduced $^{129}$I. The paragraph has been copied as below.

"In addition to atmospheric reflected by precipitation, wind speed and temperature, atmospheric boundary layer height determines vertical dispersion scale of air pollutions by thermal turbulent mixing, which might be a factor for variation of iodine isotopes. Since $^{127}$I is locally input and $^{129}$I is remotely transported from Europe, the influence of boundary layer height might be different for the two iodine isotopes. It will be further explored with longer temporal variation of iodine isotopes in the future."

Referee #2 comment #8: Again, your reply and revised figure is not included in the new manuscript. Also, there is a problem with your dates. The AQI value shows peaks in middle of March and early April, not middle of April and early May.

**Response**: With regards to "The AQI value shows peaks in middle of March and early April, not middle of April and early May", we carefully checked the raw data and figure, and find all of them are correct. The reason for the date is that the tick label for each month is not the 1st of the month, but 26th of the month. So, we change the tick label as the 1st of each month, as shown in Figure below.

Furthermore, based on the referee's comment, Fig.2 in the new manuscript has been modified and added the AQI peak on 4-6 May, 2017 in Fig. 2e.

[Figure]

Figure. Temporal variation of $^{127}I$ and AQI during March 2017 to March 2018, showing the correlation of $^{127}I$ and AQI.

l. 12: Please give an institutional email address in addition to the private one.

Response: The institutional email address "zhangly@ieecas.cn" has been added into the tittle pages of manuscript and Supplementary Information.

l. 32: Replace "Whereas, " with "In contrast, "

Response: "Whereas, " has been replaced by "In contrast, " in Line 32.

l. 126: Please add gaseous emissions to this list.

Response: "and gaseous emissions from seas" has been added after "sea spray" in Line 126.

l. 237 Please use SI units for mole fractions (nmol mol-1), not ppbv.

Response: "ppbv" has been revised to be " nmol mol$^{-1}$" in Line 237.

l. 364: This should be "influence of weather", not climate (as per the subsequent sentence of the same paragraph).

Response: Yes, we agree with the editor's opinion. Thus, "influence of climate" has been revised to be " influence of weather" in Line 364.

[revised manuscript text omitted]

**Response to Editor**

Comments to the Author:

Many thanks for revising the paper so quickly.

I am happy with the corrections made except formatting of axis labels and the AQI data in Fig. 2e. Please follow the links to SI brochure and IUPAC Green Book given in the manuscript preparation guidelines:

https://www.atmospheric-chemistry-and-physics.net/for_authors/manuscript_preparation.html

Specifically, section 5.4.1 of the SI Brochure and section 1.1 pf the IUPAC Green Book give examples for the correct formating of axis labels. The unit is separated from the quantity symbol by a division symbol, e.g. N/(10^5 m–3). There should be no commas after the quantity symbol or multiplication symbols before the unit.

Table 1 and Fig. S4 still need quantity symbols for the concentrations, e.g. $\gamma(O3)/(\mu g\ m\text{-}3)$, and other variables.

Forgive me if I am wrong, but the AQI data in Fig. 2e still seem to have much lower time resolution than the data in the figure provided in the response to the reviewers (cyan shaded area).

Dear Editor Dr. Jan Kaiser,

We sincerely thank for your great efforts with the strong responsibility. The recommended files help us a lot on the usage of SI.

According to your comments and the SI related documents, we carefully revised the manuscript and Supplementary Information, mainly including axis labels and table labels in Figs. 2-5 and Table 1 in MS, as well as Figs. 1,2,4 and Table S1 in SI.

As you mentioned, the time resolution of AQI data in Fig. 2e is not so high as that in our reply because we originally would like to show the data on those days when iodine isotopes were measured. Now in this version, we use the daily resolution of AQI data to reproduce Fig. 2e to give a clearer view of air pollution states during our analysis period.

Best regards and happy Chinese New Year!

Luyuan Zhang

Jan 24, 2020

[revised manuscript text omitted]

Not Superscript/ Subscript

| Page 15: [1] Formatted | Luyuan Zhang | 1/23/20 11:49:00 PM |

Not Superscript/ Subscript

| Page 15: [1] Formatted | Luyuan Zhang | 1/23/20 11:49:00 PM |

Not Superscript/ Subscript

| Page 15: [1] Formatted | Luyuan Zhang | 1/23/20 11:49:00 PM |

Not Superscript/ Subscript

| Page 15: [2] Formatted | Luyuan Zhang | 1/23/20 11:49:00 PM |

Formatted

| Page 15: [2] Formatted | Luyuan Zhang | 1/23/20 11:49:00 PM |

Formatted

| Page 15: [2] Formatted | Luyuan Zhang | 1/23/20 11:49:00 PM |

Formatted

| Page 15: [2] Formatted | Luyuan Zhang | 1/23/20 11:49:00 PM |

Formatted

| Page 15: [3] Formatted | Luyuan Zhang | 1/23/20 11:49:00 PM |

Formatted

| Page 15: [4] Formatted | Luyuan Zhang | 1/23/20 11:49:00 PM |

Formatted

| Page 15: [4] Formatted | Luyuan Zhang | 1/23/20 11:49:00 PM |

Formatted

| Page 15: [5] Formatted | Luyuan Zhang | 1/23/20 11:49:00 PM |

Formatted

| Page 15: [6] Formatted | Luyuan Zhang | 1/23/20 11:49:00 PM |

Formatted

| Page 15: [6] Formatted | Luyuan Zhang | 1/23/20 11:49:00 PM |

Formatted

| Page 15: [7] Formatted | Luyuan Zhang | 1/23/20 11:49:00 PM |

Formatted

| Page 15: [8] Formatted | Luyuan Zhang | 1/23/20 11:49:00 PM |

| Page 15: [8] Formatted | Luyuan Zhang | 1/23/20 11:49:00 PM |
|---|---|---|

Formatted

| Page 15: [9] Formatted | Luyuan Zhang | 1/23/20 11:49:00 PM |
|---|---|---|

Formatted

| Page 15: [10] Formatted | Luyuan Zhang | 1/23/20 11:49:00 PM |
|---|---|---|

Formatted

| Page 15: [10] Formatted | Luyuan Zhang | 1/23/20 11:49:00 PM |
|---|---|---|

Formatted

| Page 15: [11] Formatted | Luyuan Zhang | 1/23/20 11:49:00 PM |
|---|---|---|

Formatted

| Page 15: [11] Formatted | Luyuan Zhang | 1/23/20 11:49:00 PM |
|---|---|---|

Formatted

| Page 15: [12] Formatted | Luyuan Zhang | 1/23/20 11:49:00 PM |
|---|---|---|

Font: (Default) +Body (Times New Roman)

| Page 15: [13] Formatted | Luyuan Zhang | 1/23/20 11:49:00 PM |
|---|---|---|

Font: (Default) +Body (Times New Roman)

| Page 15: [14] Formatted | Luyuan Zhang | 1/23/20 11:49:00 PM |
|---|---|---|

Font: (Default) +Body (Times New Roman)

| Page 15: [15] Formatted | Luyuan Zhang | 1/23/20 11:49:00 PM |
|---|---|---|

Font: (Default) +Body (Times New Roman)

| Page 15: [16] Formatted | Luyuan Zhang | 1/23/20 11:49:00 PM |
|---|---|---|

Font: (Default) +Body (Times New Roman)

| Page 15: [17] Formatted | Luyuan Zhang | 1/23/20 11:49:00 PM |
|---|---|---|

Font: (Default) +Body (Times New Roman)

| Page 15: [18] Formatted | Luyuan Zhang | 1/23/20 11:49:00 PM |
|---|---|---|

Font: (Default) +Body (Times New Roman)

| Page 15: [19] Formatted | Luyuan Zhang | 1/23/20 11:49:00 PM |
|---|---|---|

Font: (Default) +Body (Times New Roman)

| Page 15: [20] Formatted | Luyuan Zhang | 1/23/20 11:49:00 PM |
|---|---|---|

Font: (Default) +Body (Times New Roman)

| Page 15: [21] Formatted | Luyuan Zhang | 1/23/20 11:49:00 PM |
|---|---|---|

| Page 15: [22] Formatted | Luyuan Zhang | 1/23/20 11:49:00 PM |
|---|---|---|

Formatted

| Page 15: [22] Formatted | Luyuan Zhang | 1/23/20 11:49:00 PM |
|---|---|---|

Formatted

| Page 15: [23] Formatted | Luyuan Zhang | 1/23/20 11:49:00 PM |
|---|---|---|

Font: (Default) +Body (Times New Roman)

| Page 15: [24] Formatted | Luyuan Zhang | 1/23/20 11:49:00 PM |
|---|---|---|

Font: (Default) +Body (Times New Roman)

| Page 15: [25] Formatted | Luyuan Zhang | 1/23/20 11:49:00 PM |
|---|---|---|

Font: (Default) +Body (Times New Roman)

| Page 15: [26] Formatted | Luyuan Zhang | 1/23/20 11:49:00 PM |
|---|---|---|

Font: (Default) +Body (Times New Roman)

| Page 15: [27] Formatted | Luyuan Zhang | 1/23/20 11:49:00 PM |
|---|---|---|

Font: (Default) +Body (Times New Roman)

| Page 15: [28] Formatted | Luyuan Zhang | 1/23/20 11:49:00 PM |
|---|---|---|

Font: (Default) +Body (Times New Roman)

| Page 15: [29] Formatted | Luyuan Zhang | 1/23/20 11:49:00 PM |
|---|---|---|

Font: (Default) +Body (Times New Roman)

| Page 15: [30] Formatted | Luyuan Zhang | 1/23/20 11:49:00 PM |
|---|---|---|

Font: (Default) +Body (Times New Roman)

| Page 15: [31] Formatted | Luyuan Zhang | 1/23/20 11:49:00 PM |
|---|---|---|

Font: (Default) +Body (Times New Roman)

| Page 15: [32] Formatted | Luyuan Zhang | 1/23/20 11:49:00 PM |
|---|---|---|

Font: (Default) +Body (Times New Roman)

| Page 15: [33] Formatted | Luyuan Zhang | 1/23/20 11:49:00 PM |
|---|---|---|

Font: (Default) +Body (Times New Roman)

| Page 15: [34] Formatted | Luyuan Zhang | 1/23/20 11:49:00 PM |
|---|---|---|

Font: (Default) +Body (Times New Roman)

| Page 15: [35] Formatted | Luyuan Zhang | 1/23/20 11:49:00 PM |
|---|---|---|

Font: (Default) +Body (Times New Roman)

| Page 15: [36] Formatted | Luyuan Zhang | 1/23/20 11:49:00 PM |
|---|---|---|

**Page 15: [37] Formatted** | **Luyuan Zhang** | **1/23/20 11:49:00 PM**

Font: (Default) +Body (Times New Roman)

**Page 15: [38] Formatted** | **Luyuan Zhang** | **1/23/20 11:49:00 PM**

Font: (Default) +Body (Times New Roman)

**Page 15: [39] Formatted** | **Luyuan Zhang** | **1/23/20 11:49:00 PM**

Font: (Default) +Body (Times New Roman)

**Page 15: [40] Formatted** | **Luyuan Zhang** | **1/23/20 11:49:00 PM**

Font: (Default) +Body (Times New Roman)

**Page 15: [41] Formatted** | **Luyuan Zhang** | **1/23/20 11:49:00 PM**

Font: (Default) +Body (Times New Roman)

**Page 15: [42] Formatted** | **Luyuan Zhang** | **1/23/20 11:49:00 PM**

Font: (Default) +Body (Times New Roman)

**Page 15: [43] Formatted** | **Luyuan Zhang** | **1/23/20 11:49:00 PM**

Formatted

**Page 15: [43] Formatted** | **Luyuan Zhang** | **1/23/20 11:49:00 PM**

Formatted

**Page 15: [44] Formatted** | **Luyuan Zhang** | **1/23/20 11:49:00 PM**

Font: (Default) +Body (Times New Roman)

**Page 15: [45] Formatted** | **Luyuan Zhang** | **1/23/20 11:49:00 PM**

Font: (Default) +Body (Times New Roman)

**Page 15: [46] Formatted** | **Luyuan Zhang** | **1/23/20 11:49:00 PM**

Font: (Default) +Body (Times New Roman)

**Page 15: [47] Formatted** | **Luyuan Zhang** | **1/23/20 11:49:00 PM**

Font: (Default) +Body (Times New Roman)

**Page 15: [48] Formatted** | **Luyuan Zhang** | **1/23/20 11:49:00 PM**

Font: (Default) +Body (Times New Roman)

**Page 15: [49] Formatted** | **Luyuan Zhang** | **1/23/20 11:49:00 PM**

Font: (Default) +Body (Times New Roman)

**Page 15: [50] Formatted** | **Luyuan Zhang** | **1/23/20 11:49:00 PM**

Font: (Default) +Body (Times New Roman)

**Page 15: [51] Formatted** | **Luyuan Zhang** | **1/23/20 11:49:00 PM**

| Page 15: [52] Formatted | Luyuan Zhang | 1/23/20 11:49:00 PM |
|---|---|---|

Font: (Default) +Body (Times New Roman)

| Page 15: [53] Formatted | Luyuan Zhang | 1/23/20 11:49:00 PM |
|---|---|---|

Font: (Default) +Body (Times New Roman)

| Page 15: [54] Formatted | Luyuan Zhang | 1/23/20 11:49:00 PM |
|---|---|---|

Font: (Default) +Body (Times New Roman)

| Page 15: [55] Formatted | Luyuan Zhang | 1/23/20 11:49:00 PM |
|---|---|---|

Font: (Default) +Body (Times New Roman)

| Page 15: [56] Formatted | Luyuan Zhang | 1/23/20 11:49:00 PM |
|---|---|---|

Font: (Default) +Body (Times New Roman)

| Page 15: [57] Formatted | Luyuan Zhang | 1/23/20 11:49:00 PM |
|---|---|---|

Font: (Default) +Body (Times New Roman)

| Page 15: [58] Formatted | Luyuan Zhang | 1/23/20 11:49:00 PM |
|---|---|---|

Font: (Default) +Body (Times New Roman)

| Page 15: [59] Formatted | Luyuan Zhang | 1/23/20 11:49:00 PM |
|---|---|---|

Font: (Default) +Body (Times New Roman)

| Page 15: [60] Formatted | Luyuan Zhang | 1/23/20 11:49:00 PM |
|---|---|---|

Font: (Default) +Body (Times New Roman)

| Page 15: [61] Formatted | Luyuan Zhang | 1/23/20 11:49:00 PM |
|---|---|---|

Font: (Default) +Body (Times New Roman)

| Page 15: [62] Formatted | Luyuan Zhang | 1/23/20 11:49:00 PM |
|---|---|---|

Font: (Default) +Body (Times New Roman)

| Page 15: [63] Formatted | Luyuan Zhang | 1/23/20 11:49:00 PM |
|---|---|---|

Font: (Default) +Body (Times New Roman)

| Page 15: [64] Formatted | Luyuan Zhang | 1/23/20 11:49:00 PM |
|---|---|---|

Formatted

| Page 15: [65] Formatted | Luyuan Zhang | 1/23/20 11:49:00 PM |
|---|---|---|

Font: (Default) +Body (Times New Roman)

| Page 15: [66] Formatted | Luyuan Zhang | 1/23/20 11:49:00 PM |
|---|---|---|

Font: (Default) +Body (Times New Roman)

| Page 15: [67] Formatted | Luyuan Zhang | 1/23/20 11:49:00 PM |
|---|---|---|

**Page 15: [68] Formatted**          Luyuan Zhang          1/23/20 11:49:00 PM

Font: (Default) +Body (Times New Roman)

**Page 15: [69] Formatted**          Luyuan Zhang          1/23/20 11:49:00 PM

Font: (Default) +Body (Times New Roman)

**Page 15: [70] Formatted**          Luyuan Zhang          1/23/20 11:49:00 PM

Font: (Default) +Body (Times New Roman)

**Page 15: [71] Formatted**          Luyuan Zhang          1/23/20 11:49:00 PM

Font: (Default) +Body (Times New Roman)

**Page 15: [72] Formatted**          Luyuan Zhang          1/23/20 11:49:00 PM

Font: (Default) +Body (Times New Roman)

**Page 15: [73] Formatted**          Luyuan Zhang          1/23/20 11:49:00 PM

Font: (Default) +Body (Times New Roman)

**Page 15: [74] Formatted**          Luyuan Zhang          1/23/20 11:49:00 PM

Font: (Default) +Body (Times New Roman)

**Page 15: [75] Formatted**          Luyuan Zhang          1/23/20 11:49:00 PM

Font: (Default) +Body (Times New Roman)

**Page 15: [76] Formatted**          Luyuan Zhang          1/23/20 11:49:00 PM

Font: (Default) +Body (Times New Roman)

**Page 15: [77] Formatted**          Luyuan Zhang          1/23/20 11:49:00 PM

Font: (Default) +Body (Times New Roman)

**Page 15: [78] Formatted**          Luyuan Zhang          1/23/20 11:49:00 PM

Font: (Default) +Body (Times New Roman)

**Page 15: [79] Formatted**          Luyuan Zhang          1/23/20 11:49:00 PM

Font: (Default) +Body (Times New Roman)

**Page 15: [80] Formatted**          Luyuan Zhang          1/23/20 11:49:00 PM

Font: (Default) +Body (Times New Roman)

**Page 15: [81] Formatted**          Luyuan Zhang          1/23/20 11:49:00 PM

Font: (Default) +Body (Times New Roman)

**Page 15: [82] Formatted**          Luyuan Zhang          1/23/20 11:49:00 PM

Font: (Default) +Body (Times New Roman)

**Page 15: [83] Formatted**          Luyuan Zhang          1/23/20 11:49:00 PM

| | | |
|---|---|---|
| **Page 15: [84] Formatted** | **Luyuan Zhang** | **1/23/20 11:49:00 PM** |

Font: (Default) +Body (Times New Roman)

| | | |
|---|---|---|
| **Page 15: [85] Formatted** | **Luyuan Zhang** | **1/23/20 11:49:00 PM** |

Superscript

| | | |
|---|---|---|
| **Page 15: [86] Formatted** | **Luyuan Zhang** | **1/23/20 11:49:00 PM** |

Font: (Default) +Body (Times New Roman)

| | | |
|---|---|---|
| **Page 15: [87] Formatted** | **Luyuan Zhang** | **1/23/20 11:49:00 PM** |

Font: (Default) +Body (Times New Roman)

| | | |
|---|---|---|
| **Page 15: [88] Formatted** | **Luyuan Zhang** | **1/23/20 11:49:00 PM** |

Font: (Default) +Body (Times New Roman)

| | | |
|---|---|---|
| **Page 15: [89] Formatted** | **Luyuan Zhang** | **1/23/20 11:49:00 PM** |

Font: (Default) +Body (Times New Roman)

| | | |
|---|---|---|
| **Page 15: [90] Formatted** | **Luyuan Zhang** | **1/23/20 11:49:00 PM** |

Font: (Default) +Body (Times New Roman)

| | | |
|---|---|---|
| **Page 15: [91] Formatted** | **Luyuan Zhang** | **1/23/20 11:49:00 PM** |

Font: (Default) +Body (Times New Roman)

| | | |
|---|---|---|
| **Page 15: [92] Formatted** | **Luyuan Zhang** | **1/23/20 11:49:00 PM** |

Font: (Default) +Body (Times New Roman)

| | | |
|---|---|---|
| **Page 15: [93] Formatted** | **Luyuan Zhang** | **1/23/20 11:49:00 PM** |

Font: (Default) +Body (Times New Roman)

| | | |
|---|---|---|
| **Page 15: [94] Formatted** | **Luyuan Zhang** | **1/23/20 11:49:00 PM** |

Font: (Default) +Body (Times New Roman)

| | | |
|---|---|---|
| **Page 15: [95] Formatted** | **Luyuan Zhang** | **1/23/20 11:49:00 PM** |

Font: (Default) +Body (Times New Roman)

| | | |
|---|---|---|
| **Page 15: [96] Formatted** | **Luyuan Zhang** | **1/23/20 11:49:00 PM** |

Font: (Default) +Body (Times New Roman)

| | | |
|---|---|---|
| **Page 15: [97] Formatted** | **Luyuan Zhang** | **1/23/20 11:49:00 PM** |

Font: (Default) +Body (Times New Roman)

| | | |
|---|---|---|
| **Page 15: [98] Formatted** | **Luyuan Zhang** | **1/23/20 11:49:00 PM** |

Font: (Default) +Body (Times New Roman)

| | | |
|---|---|---|
| **Page 15: [99] Formatted** | **Luyuan Zhang** | **1/23/20 11:49:00 PM** |

**Page 15: [100] Formatted** | **Luyuan Zhang** | **1/23/20 11:49:00 PM**

Font: (Default) +Body (Times New Roman)

**Page 15: [101] Formatted** | **Luyuan Zhang** | **1/23/20 11:49:00 PM**

Font: (Default) +Body (Times New Roman)

**Page 15: [102] Formatted** | **Luyuan Zhang** | **1/23/20 11:49:00 PM**

Font: (Default) +Body (Times New Roman)

**Page 15: [103] Formatted** | **Luyuan Zhang** | **1/23/20 11:49:00 PM**

Font: (Default) +Body (Times New Roman)

**Page 15: [104] Formatted** | **Luyuan Zhang** | **1/23/20 11:49:00 PM**

Font: (Default) +Body (Times New Roman)

**Page 15: [105] Formatted** | **Luyuan Zhang** | **1/23/20 11:49:00 PM**

Font: (Default) +Body (Times New Roman)

**Page 15: [106] Formatted** | **Luyuan Zhang** | **1/23/20 11:49:00 PM**

Font: (Default) +Body (Times New Roman)

**Page 15: [107] Formatted** | **Luyuan Zhang** | **1/23/20 11:49:00 PM**

Font: (Default) +Body (Times New Roman)

**Page 15: [108] Formatted** | **Luyuan Zhang** | **1/23/20 11:49:00 PM**

Font: (Default) +Body (Times New Roman)

**Page 15: [109] Formatted** | **Luyuan Zhang** | **1/23/20 11:49:00 PM**

Font: (Default) +Body (Times New Roman)

**Page 15: [110] Formatted** | **Luyuan Zhang** | **1/23/20 11:49:00 PM**

Font: (Default) +Body (Times New Roman)

**Page 15: [111] Formatted** | **Luyuan Zhang** | **1/23/20 11:49:00 PM**

Font: (Default) +Body (Times New Roman)

**Page 15: [112] Formatted** | **Luyuan Zhang** | **1/23/20 11:49:00 PM**

Font: (Default) +Body (Times New Roman)

**Page 15: [113] Formatted** | **Luyuan Zhang** | **1/23/20 11:49:00 PM**

Font: (Default) +Body (Times New Roman)

**Page 15: [114] Formatted** | **Luyuan Zhang** | **1/23/20 11:49:00 PM**

Font: (Default) +Body (Times New Roman)

**Page 15: [115] Formatted** | **Luyuan Zhang** | **1/23/20 11:49:00 PM**

**Page 15: [116] Formatted** | **Luyuan Zhang** | **1/23/20 11:49:00 PM**

Font: (Default) +Body (Times New Roman)

**Page 15: [117] Formatted** | **Luyuan Zhang** | **1/23/20 11:49:00 PM**

Font: (Default) +Body (Times New Roman)

**Page 15: [118] Formatted** | **Luyuan Zhang** | **1/23/20 11:49:00 PM**

Font: (Default) +Body (Times New Roman)

**Page 15: [119] Formatted** | **Luyuan Zhang** | **1/23/20 11:49:00 PM**

Font: (Default) +Body (Times New Roman)

**Page 15: [120] Formatted** | **Luyuan Zhang** | **1/23/20 11:49:00 PM**

Font: (Default) +Body (Times New Roman)

**Page 15: [121] Formatted** | **Luyuan Zhang** | **1/23/20 11:49:00 PM**

Font: (Default) +Body (Times New Roman)

**Page 15: [122] Formatted** | **Luyuan Zhang** | **1/23/20 11:49:00 PM**

Font: (Default) +Body (Times New Roman)

**Page 15: [123] Formatted** | **Luyuan Zhang** | **1/23/20 11:49:00 PM**

Font: (Default) +Body (Times New Roman)

**Page 15: [124] Formatted** | **Luyuan Zhang** | **1/23/20 11:49:00 PM**

Font: (Default) +Body (Times New Roman)

**Page 15: [125] Formatted** | **Luyuan Zhang** | **1/23/20 11:49:00 PM**

Font: (Default) +Body (Times New Roman)

**Page 15: [126] Formatted** | **Luyuan Zhang** | **1/23/20 11:49:00 PM**

Font: (Default) +Body (Times New Roman)

**Page 15: [127] Formatted** | **Luyuan Zhang** | **1/23/20 11:49:00 PM**

Font: (Default) +Body (Times New Roman)

**Page 15: [128] Formatted** | **Luyuan Zhang** | **1/23/20 11:49:00 PM**

Font: (Default) +Body (Times New Roman)

**Page 15: [129] Formatted** | **Luyuan Zhang** | **1/23/20 11:49:00 PM**

Font: (Default) +Body (Times New Roman)

**Page 15: [130] Formatted** | **Luyuan Zhang** | **1/23/20 11:49:00 PM**

Font: (Default) +Body (Times New Roman)

**Page 15: [131] Formatted** | **Luyuan Zhang** | **1/23/20 11:49:00 PM**

**Page 15: [132] Formatted** | **Luyuan Zhang** | **1/23/20 11:49:00 PM**

Font: (Default) +Body (Times New Roman)

**Page 15: [133] Formatted** | **Luyuan Zhang** | **1/23/20 11:49:00 PM**

Font: (Default) +Body (Times New Roman)

**Page 15: [134] Formatted** | **Luyuan Zhang** | **1/23/20 11:49:00 PM**

Font: (Default) +Body (Times New Roman)

**Page 15: [135] Formatted** | **Luyuan Zhang** | **1/23/20 11:49:00 PM**

Font: (Default) +Body (Times New Roman)

**Page 15: [136] Formatted** | **Luyuan Zhang** | **1/23/20 11:49:00 PM**

Font: (Default) +Body (Times New Roman)

**Page 15: [137] Formatted** | **Luyuan Zhang** | **1/23/20 11:49:00 PM**

Font: (Default) +Body (Times New Roman)

**Page 15: [138] Formatted** | **Luyuan Zhang** | **1/23/20 11:49:00 PM**

Font: (Default) +Body (Times New Roman)

**Page 15: [139] Formatted** | **Luyuan Zhang** | **1/23/20 11:49:00 PM**

Font: (Default) +Body (Times New Roman)

**Page 15: [140] Formatted** | **Luyuan Zhang** | **1/23/20 11:49:00 PM**

Font: (Default) +Body (Times New Roman)

**Page 15: [141] Formatted** | **Luyuan Zhang** | **1/23/20 11:49:00 PM**

Font: (Default) +Body (Times New Roman)

**Page 15: [142] Formatted** | **Luyuan Zhang** | **1/23/20 11:49:00 PM**

Font: (Default) +Body (Times New Roman)

**Page 15: [143] Formatted** | **Luyuan Zhang** | **1/23/20 11:49:00 PM**

Font: (Default) +Body (Times New Roman)

**Page 15: [144] Formatted** | **Luyuan Zhang** | **1/23/20 11:49:00 PM**

Font: (Default) +Body (Times New Roman)

**Page 15: [145] Formatted** | **Luyuan Zhang** | **1/23/20 11:49:00 PM**

Font: (Default) +Body (Times New Roman)

**Page 15: [146] Formatted** | **Luyuan Zhang** | **1/23/20 11:49:00 PM**

Formatted

**Page 15: [147] Formatted** | **Luyuan Zhang** | **1/23/20 11:49:00 PM**

**Page 15: [148] Formatted**      **Luyuan Zhang**      **1/23/20 11:49:00 PM**

Font: (Default) +Body (Times New Roman)

**Page 15: [149] Formatted**      **Luyuan Zhang**      **1/23/20 11:49:00 PM**

Font: (Default) +Body (Times New Roman)

**Page 15: [150] Formatted**      **Luyuan Zhang**      **1/23/20 11:49:00 PM**

Font: (Default) +Body (Times New Roman)

**Page 15: [151] Formatted**      **Luyuan Zhang**      **1/23/20 11:49:00 PM**

Font: (Default) +Body (Times New Roman)

**Page 15: [152] Formatted**      **Luyuan Zhang**      **1/23/20 11:49:00 PM**

Font: (Default) +Body (Times New Roman)

**Page 15: [153] Formatted**      **Luyuan Zhang**      **1/23/20 11:49:00 PM**

Font: (Default) +Body (Times New Roman)

**Page 15: [154] Formatted**      **Luyuan Zhang**      **1/23/20 11:49:00 PM**

Font: (Default) +Body (Times New Roman)

**Page 15: [155] Formatted**      **Luyuan Zhang**      **1/23/20 11:49:00 PM**

Font: (Default) +Body (Times New Roman)

**Page 15: [156] Formatted**      **Luyuan Zhang**      **1/23/20 11:49:00 PM**

Font: (Default) +Body (Times New Roman)

**Page 15: [157] Formatted**      **Luyuan Zhang**      **1/23/20 11:49:00 PM**

Font: (Default) +Body (Times New Roman)

**Page 15: [158] Formatted**      **Luyuan Zhang**      **1/23/20 11:49:00 PM**

Font: (Default) +Body (Times New Roman)

**Page 15: [159] Formatted**      **Luyuan Zhang**      **1/23/20 11:49:00 PM**

Font: (Default) +Body (Times New Roman)

**Page 15: [160] Formatted**      **Luyuan Zhang**      **1/23/20 11:49:00 PM**

Font: (Default) +Body (Times New Roman)

**Page 15: [161] Formatted**      **Luyuan Zhang**      **1/23/20 11:49:00 PM**

Font: (Default) +Body (Times New Roman)

**Page 15: [162] Formatted**      **Luyuan Zhang**      **1/23/20 11:49:00 PM**

Font: (Default) +Body (Times New Roman)

**Page 15: [163] Formatted**      **Luyuan Zhang**      **1/23/20 11:49:00 PM**

**Page 15: [164] Formatted**            **Luyuan Zhang**            **1/23/20 11:49:00 PM**

Font: (Default) +Body (Times New Roman)

**Page 15: [165] Formatted**            **Luyuan Zhang**            **1/23/20 11:49:00 PM**

Font: (Default) +Body (Times New Roman)

**Page 15: [166] Formatted**            **Luyuan Zhang**            **1/23/20 11:49:00 PM**

Font: (Default) +Body (Times New Roman)

**Page 15: [167] Formatted**            **Luyuan Zhang**            **1/23/20 11:49:00 PM**

Formatted

**Page 15: [167] Formatted**            **Luyuan Zhang**            **1/23/20 11:49:00 PM**

Formatted

**Page 15: [168] Formatted**            **Luyuan Zhang**            **1/23/20 11:49:00 PM**

Font: (Default) +Body (Times New Roman)

**Page 15: [169] Formatted**            **Luyuan Zhang**            **1/23/20 11:49:00 PM**

Font: (Default) +Body (Times New Roman)

**Page 15: [170] Formatted**            **Luyuan Zhang**            **1/23/20 11:49:00 PM**

Font: (Default) +Body (Times New Roman)

**Page 15: [171] Formatted**            **Luyuan Zhang**            **1/23/20 11:49:00 PM**

Font: (Default) +Body (Times New Roman)

**Page 15: [172] Formatted**            **Luyuan Zhang**            **1/23/20 11:49:00 PM**

Font: (Default) +Body (Times New Roman)

**Page 15: [173] Formatted**            **Luyuan Zhang**            **1/23/20 11:49:00 PM**

Font: (Default) +Body (Times New Roman)

**Page 15: [174] Formatted**            **Luyuan Zhang**            **1/23/20 11:49:00 PM**

Font: (Default) +Body (Times New Roman)

**Page 15: [175] Formatted**            **Luyuan Zhang**            **1/23/20 11:49:00 PM**

Font: (Default) +Body (Times New Roman)

**Page 15: [176] Formatted**            **Luyuan Zhang**            **1/23/20 11:49:00 PM**

Font: (Default) +Body (Times New Roman)

**Page 15: [177] Formatted**            **Luyuan Zhang**            **1/23/20 11:49:00 PM**

Font: (Default) +Body (Times New Roman)

**Page 15: [178] Formatted**            **Luyuan Zhang**            **1/23/20 11:49:00 PM**

| Page 15: [179] Formatted | Luyuan Zhang | 1/23/20 11:49:00 PM |
|---|---|---|

Font: (Default) +Body (Times New Roman)

| Page 15: [180] Formatted | Luyuan Zhang | 1/23/20 11:49:00 PM |
|---|---|---|

Font: (Default) +Body (Times New Roman)

| Page 15: [181] Formatted | Luyuan Zhang | 1/23/20 11:49:00 PM |
|---|---|---|

Font: (Default) +Body (Times New Roman)

| Page 15: [182] Formatted | Luyuan Zhang | 1/23/20 11:49:00 PM |
|---|---|---|

Font: (Default) +Body (Times New Roman)

| Page 15: [183] Formatted | Luyuan Zhang | 1/23/20 11:49:00 PM |
|---|---|---|

Font: (Default) +Body (Times New Roman)

| Page 15: [184] Formatted | Luyuan Zhang | 1/23/20 11:49:00 PM |
|---|---|---|

Font: (Default) +Body (Times New Roman)

| Page 15: [185] Formatted | Luyuan Zhang | 1/23/20 11:49:00 PM |
|---|---|---|

Font: (Default) +Body (Times New Roman)

| Page 15: [186] Formatted | Luyuan Zhang | 1/23/20 11:49:00 PM |
|---|---|---|

Font: (Default) +Body (Times New Roman)

| Page 15: [187] Formatted | Luyuan Zhang | 1/23/20 11:49:00 PM |
|---|---|---|

Font: (Default) +Body (Times New Roman)

| Page 15: [188] Formatted | Luyuan Zhang | 1/23/20 11:49:00 PM |
|---|---|---|

Formatted

| Page 15: [188] Formatted | Luyuan Zhang | 1/23/20 11:49:00 PM |
|---|---|---|

Formatted

| Page 15: [189] Formatted | Luyuan Zhang | 1/23/20 11:49:00 PM |
|---|---|---|

Font: (Default) +Body (Times New Roman)

| Page 15: [190] Formatted | Luyuan Zhang | 1/23/20 11:49:00 PM |
|---|---|---|

Font: (Default) +Body (Times New Roman)

| Page 15: [191] Formatted | Luyuan Zhang | 1/23/20 11:49:00 PM |
|---|---|---|

Font: (Default) +Body (Times New Roman)

| Page 15: [192] Formatted | Luyuan Zhang | 1/23/20 11:49:00 PM |
|---|---|---|

Font: (Default) +Body (Times New Roman)

| Page 15: [193] Formatted | Luyuan Zhang | 1/23/20 11:49:00 PM |
|---|---|---|

**Page 15: [194] Formatted** | **Luyuan Zhang** | **1/23/20 11:49:00 PM**

Font: (Default) +Body (Times New Roman)

**Page 15: [195] Formatted** | **Luyuan Zhang** | **1/23/20 11:49:00 PM**

Font: (Default) +Body (Times New Roman)

**Page 15: [196] Formatted** | **Luyuan Zhang** | **1/23/20 11:49:00 PM**

Font: (Default) +Body (Times New Roman)

**Page 15: [197] Formatted** | **Luyuan Zhang** | **1/23/20 11:49:00 PM**

Font: (Default) +Body (Times New Roman)

**Page 15: [198] Formatted** | **Luyuan Zhang** | **1/23/20 11:49:00 PM**

Font: (Default) +Body (Times New Roman)

**Page 15: [199] Formatted** | **Luyuan Zhang** | **1/23/20 11:49:00 PM**

Font: (Default) +Body (Times New Roman)

**Page 15: [200] Formatted** | **Luyuan Zhang** | **1/23/20 11:49:00 PM**

Font: (Default) +Body (Times New Roman)

**Page 15: [201] Formatted** | **Luyuan Zhang** | **1/23/20 11:49:00 PM**

Font: (Default) +Body (Times New Roman)

**Page 15: [202] Formatted** | **Luyuan Zhang** | **1/23/20 11:49:00 PM**

Font: (Default) +Body (Times New Roman)

**Page 15: [203] Formatted** | **Luyuan Zhang** | **1/23/20 11:49:00 PM**

Font: (Default) +Body (Times New Roman)

**Page 15: [204] Formatted** | **Luyuan Zhang** | **1/23/20 11:49:00 PM**

Font: (Default) +Body (Times New Roman)

**Page 15: [205] Formatted** | **Luyuan Zhang** | **1/23/20 11:49:00 PM**

Font: (Default) +Body (Times New Roman)

**Page 15: [206] Formatted** | **Luyuan Zhang** | **1/23/20 11:49:00 PM**

Font: (Default) +Body (Times New Roman)

**Page 15: [207] Formatted** | **Luyuan Zhang** | **1/23/20 11:49:00 PM**

Font: (Default) +Body (Times New Roman)

**Page 15: [208] Formatted** | **Luyuan Zhang** | **1/23/20 11:49:00 PM**

Font: (Default) +Body (Times New Roman)

**Page 15: [209] Formatted** | **Luyuan Zhang** | **1/23/20 11:49:00 PM**

| Page 15: [209] Formatted | Luyuan Zhang | 1/23/20 11:49:00 PM |
|---|---|---|

Formatted

| Page 15: [210] Formatted | Luyuan Zhang | 1/23/20 11:49:00 PM |
|---|---|---|

Font: (Default) +Body (Times New Roman)

| Page 15: [211] Formatted | Luyuan Zhang | 1/23/20 11:49:00 PM |
|---|---|---|

Font: (Default) +Body (Times New Roman)

| Page 15: [212] Formatted | Luyuan Zhang | 1/23/20 11:49:00 PM |
|---|---|---|

Font: (Default) +Body (Times New Roman)

| Page 15: [213] Formatted | Luyuan Zhang | 1/23/20 11:49:00 PM |
|---|---|---|

Font: (Default) +Body (Times New Roman)

| Page 15: [214] Formatted | Luyuan Zhang | 1/23/20 11:49:00 PM |
|---|---|---|

Font: (Default) +Body (Times New Roman)

| Page 15: [215] Formatted | Luyuan Zhang | 1/23/20 11:49:00 PM |
|---|---|---|

Font: (Default) +Body (Times New Roman)

| Page 15: [216] Formatted | Luyuan Zhang | 1/23/20 11:49:00 PM |
|---|---|---|

Font: (Default) +Body (Times New Roman)

| Page 15: [217] Formatted | Luyuan Zhang | 1/23/20 11:49:00 PM |
|---|---|---|

Font: (Default) +Body (Times New Roman)

| Page 15: [218] Formatted | Luyuan Zhang | 1/23/20 11:49:00 PM |
|---|---|---|

Font: (Default) +Body (Times New Roman)

| Page 15: [219] Formatted | Luyuan Zhang | 1/23/20 11:49:00 PM |
|---|---|---|

Font: (Default) +Body (Times New Roman)

| Page 15: [220] Formatted | Luyuan Zhang | 1/23/20 11:49:00 PM |
|---|---|---|

Font: (Default) +Body (Times New Roman)

| Page 15: [221] Formatted | Luyuan Zhang | 1/23/20 11:49:00 PM |
|---|---|---|

Font: (Default) +Body (Times New Roman)

| Page 15: [222] Formatted | Luyuan Zhang | 1/23/20 11:49:00 PM |
|---|---|---|

Font: (Default) +Body (Times New Roman)

| Page 15: [223] Formatted | Luyuan Zhang | 1/23/20 11:49:00 PM |
|---|---|---|

Font: (Default) +Body (Times New Roman)

| Page 15: [224] Formatted | Luyuan Zhang | 1/23/20 11:49:00 PM |
|---|---|---|

| Page 15: [225] Formatted | Luyuan Zhang | 1/23/20 11:49:00 PM |
|---|---|---|

Font: (Default) +Body (Times New Roman)

| Page 15: [226] Formatted | Luyuan Zhang | 1/23/20 11:49:00 PM |
|---|---|---|

Font: (Default) +Body (Times New Roman)

| Page 15: [227] Formatted | Luyuan Zhang | 1/23/20 11:49:00 PM |
|---|---|---|

Font: (Default) +Body (Times New Roman)

| Page 15: [228] Formatted | Luyuan Zhang | 1/23/20 11:49:00 PM |
|---|---|---|

Font: (Default) +Body (Times New Roman)

| Page 15: [229] Formatted | Luyuan Zhang | 1/23/20 11:49:00 PM |
|---|---|---|

Font: (Default) +Body (Times New Roman)

| Page 15: [230] Formatted | Luyuan Zhang | 1/23/20 11:49:00 PM |
|---|---|---|

Formatted

| Page 15: [230] Formatted | Luyuan Zhang | 1/23/20 11:49:00 PM |
|---|---|---|

Formatted

| Page 15: [231] Formatted | Luyuan Zhang | 1/23/20 11:49:00 PM |
|---|---|---|

Font: (Default) +Body (Times New Roman)

| Page 15: [232] Formatted | Luyuan Zhang | 1/23/20 11:49:00 PM |
|---|---|---|

Font: (Default) +Body (Times New Roman)

| Page 15: [233] Formatted | Luyuan Zhang | 1/23/20 11:49:00 PM |
|---|---|---|

Font: (Default) +Body (Times New Roman)

| Page 15: [234] Formatted | Luyuan Zhang | 1/23/20 11:49:00 PM |
|---|---|---|

Font: (Default) +Body (Times New Roman)

| Page 15: [235] Formatted | Luyuan Zhang | 1/23/20 11:49:00 PM |
|---|---|---|

Font: (Default) +Body (Times New Roman)

| Page 15: [236] Formatted | Luyuan Zhang | 1/23/20 11:49:00 PM |
|---|---|---|

Font: (Default) +Body (Times New Roman)

| Page 15: [237] Formatted | Luyuan Zhang | 1/23/20 11:49:00 PM |
|---|---|---|

Font: (Default) +Body (Times New Roman)

| Page 15: [238] Formatted | Luyuan Zhang | 1/23/20 11:49:00 PM |
|---|---|---|

Font: (Default) +Body (Times New Roman)

| Page 15: [239] Formatted | Luyuan Zhang | 1/23/20 11:49:00 PM |
|---|---|---|

**Page 15: [240] Formatted** | **Luyuan Zhang** | **1/23/20 11:49:00 PM**

Font: (Default) +Body (Times New Roman)

**Page 15: [241] Formatted** | **Luyuan Zhang** | **1/23/20 11:49:00 PM**

Font: (Default) +Body (Times New Roman)

**Page 15: [242] Formatted** | **Luyuan Zhang** | **1/23/20 11:49:00 PM**

Font: (Default) +Body (Times New Roman)

**Page 15: [243] Formatted** | **Luyuan Zhang** | **1/23/20 11:49:00 PM**

Font: (Default) +Body (Times New Roman)

**Page 15: [244] Formatted** | **Luyuan Zhang** | **1/23/20 11:49:00 PM**

Font: (Default) +Body (Times New Roman)

**Page 15: [245] Formatted** | **Luyuan Zhang** | **1/23/20 11:49:00 PM**

Font: (Default) +Body (Times New Roman)

**Page 15: [246] Formatted** | **Luyuan Zhang** | **1/23/20 11:49:00 PM**

Font: (Default) +Body (Times New Roman)

**Page 15: [247] Formatted** | **Luyuan Zhang** | **1/23/20 11:49:00 PM**

Font: (Default) +Body (Times New Roman)

**Page 15: [248] Formatted** | **Luyuan Zhang** | **1/23/20 11:49:00 PM**

Font: (Default) +Body (Times New Roman)

**Page 15: [249] Formatted** | **Luyuan Zhang** | **1/23/20 11:49:00 PM**

Font: (Default) +Body (Times New Roman)

**Page 15: [250] Formatted** | **Luyuan Zhang** | **1/23/20 11:49:00 PM**

Font: (Default) +Body (Times New Roman)

**Page 15: [251] Formatted** | **Luyuan Zhang** | **1/23/20 11:49:00 PM**

Formatted

**Page 15: [251] Formatted** | **Luyuan Zhang** | **1/23/20 11:49:00 PM**

Formatted

**Page 15: [252] Formatted** | **Luyuan Zhang** | **1/23/20 11:49:00 PM**

Font: (Default) +Body (Times New Roman)

**Page 15: [253] Formatted** | **Luyuan Zhang** | **1/23/20 11:49:00 PM**

Font: (Default) +Body (Times New Roman)

**Page 15: [254] Formatted** | **Luyuan Zhang** | **1/23/20 11:49:00 PM**

**Page 15: [255] Formatted** | **Luyuan Zhang** | **1/23/20 11:49:00 PM**

Font: (Default) +Body (Times New Roman)

**Page 15: [256] Formatted** | **Luyuan Zhang** | **1/23/20 11:49:00 PM**

Font: (Default) +Body (Times New Roman)

**Page 15: [257] Formatted** | **Luyuan Zhang** | **1/23/20 11:49:00 PM**

Font: (Default) +Body (Times New Roman)

**Page 15: [258] Formatted** | **Luyuan Zhang** | **1/23/20 11:49:00 PM**

Font: (Default) +Body (Times New Roman)

**Page 15: [259] Formatted** | **Luyuan Zhang** | **1/23/20 11:49:00 PM**

Font: (Default) +Body (Times New Roman)

**Page 15: [260] Formatted** | **Luyuan Zhang** | **1/23/20 11:49:00 PM**

Font: (Default) +Body (Times New Roman)

**Page 15: [261] Formatted** | **Luyuan Zhang** | **1/23/20 11:49:00 PM**

Font: (Default) +Body (Times New Roman)

**Page 15: [262] Formatted** | **Luyuan Zhang** | **1/23/20 11:49:00 PM**

Font: (Default) +Body (Times New Roman)

**Page 15: [263] Formatted** | **Luyuan Zhang** | **1/23/20 11:49:00 PM**

Font: (Default) +Body (Times New Roman)

**Page 15: [264] Formatted** | **Luyuan Zhang** | **1/23/20 11:49:00 PM**

Font: (Default) +Body (Times New Roman)

**Page 15: [265] Formatted** | **Luyuan Zhang** | **1/23/20 11:49:00 PM**

Font: (Default) +Body (Times New Roman)

**Page 15: [266] Formatted** | **Luyuan Zhang** | **1/23/20 11:49:00 PM**

Font: (Default) +Body (Times New Roman)

**Page 15: [267] Formatted** | **Luyuan Zhang** | **1/23/20 11:49:00 PM**

Font: (Default) +Body (Times New Roman)

**Page 15: [268] Formatted** | **Luyuan Zhang** | **1/23/20 11:49:00 PM**

Font: (Default) +Body (Times New Roman)

**Page 15: [269] Formatted** | **Luyuan Zhang** | **1/23/20 11:49:00 PM**

Font: (Default) +Body (Times New Roman)

**Page 15: [270] Formatted** | **Luyuan Zhang** | **1/23/20 11:49:00 PM**

Font: (Default) +Body (Times New Roman)

**Supplementary Information**

[revised manuscript text omitted]

$$y = 0.0974x + 1.3604$$

[Figure]

[Figure]

90  **Fig. S1 Relationship between $^{127}$I and $^{129}$I with a weak correlation (R=0.33, p<0.01) between the two iodine isotopes. This indicates the two iodine isotopes have different sources and their temporal variation patterns were affected by different factors.**

[Figure]

[Figure]

95  **Fig. S2 Relationship between iodine isotopes and total suspended particles (TSP) in Xi'an, China (n=68), suggesting significant correlation between $^{127}$I and TSP, and no correlation between $^{129}$I and TSP. The results indicate $^{127}$I was sourced from local input and $^{129}$I was transported to the studied site externally.**

**Fig. S3 Back trajectories analyisis on date of a) 18th April, 2017; b) 18th May, 2017; c) 14th July, 2017; d) 31st August, 2017; e) 6th September, 2017; f) 15th November, 2017; g) 28th December, 2017; h) 17th January, 2018.**

[Figure]

**Fig. S4 Relations between $^{127}$I and air pollutants including PM10, PM2.5, SO₂, NO₂, CO and O₃, showing significant correlation.**

[Figure]

**Fig. S5** 850 hPa water vapor transmission flow field on 2 May, 2017 (a), and 21 May, 2017 (b). Data from: https://cmdp.ncc-cma.net/Monitoring/monsoon.php?ListElem=vt85. The red dot in the figures is the sampling location, Xi'an, China.

115      **Table S1 Mean $^{129}$I concentrations and $^{129}$I/$^{127}$I ratios in three high-level periods (HLP) and two low-level periods (LLP)**

| No | Type | Start date | Stop date | N($^{129}$I) / ($10^5$ m$^{-3}$) | | $^{129}$I/$^{127}$I number ratio / ($\times 10^{-10}$) | | Monsoon stage |
|----|------|------------|-----------|---------|-----|---------|-----|---------------|
| | | | | Average | RSD | Average | RSD | |
| 1 | HLP 1 | 28 Mar, 2017 | 22 May, 2017 | 2.37 | 91% | 101 | 89% | WM and onset of SM |
| 2 | LLP 1 | 23 May, 2017 | 25 Jul, 2017 | 0.49 | 60% | 28.5 | 65% | Active of SM |
| 3 | HLP 2 | 4 Aug, 2017 | 12 Sep, 2017 | 1.98 | 109% | 155. | 141% | Break of SM |
| 4 | LLP 2 | 21 Sep, 2017 | 11 Oct, 2017 | 0.66 | 44% | 40.1 | 44% | Revival of SM |
| 5 | HLP 3 | 13 Oct, 2017 | 20 Mar, 2018 | 2.41 | 44% | 67.9 | 83% | SM retreat and WM advance then active |